# Versatile nanobody-based approach to image, track and reconstitute functional Neurexin-1 in vivo

Rosario Vicidomini [1,5], Saumitra Dey Choudhury [1,4,5], Tae Hee Han [1], Tho Huu Nguyen[1], Peter Nguyen[1], Felipe Opazo [2,3] & Mihaela Serpe [1] ✉

Neurexins are key adhesion proteins that coordinate extracellular and intracellular synaptic components. Nonetheless, the low abundance of these multidomain proteins has complicated any localization and structure-function studies. Here we combine an ALFA tag (AT)/nanobody (NbALFA) tool with classic genetics, cell biology and electrophysiology to examine the distribution and function of the *Drosophila* Nrx-1 in vivo. We generate full-length and ΔPDZ ALFA-tagged Nrx-1 variants and find that the PDZ binding motif is key to Nrx-1 surface expression. A PDZ binding motif provided in trans, via genetically encoded cytosolic NbALFA-PDZ chimera, fully restores the synaptic localization and function of Nrx$^{ΔPDZ-AT}$. Using cytosolic NbALFA-mScarlet intrabody, we achieve compartment-specific detection of endogenous Nrx-1, track live Nrx-1 transport along the motor neuron axons, and demonstrate that Nrx-1 co-migrates with Rab2-positive vesicles. Our findings illustrate the versatility of the ALFA system and pave the way towards dissecting functional domains of complex proteins in vivo.

Neurexins are transmembrane proteins that coordinate synapse organization through trans-synaptic interactions with a variety of postsynaptic binding partners, including neuroligins, leucine-rich repeat transmembrane neuronal proteins, α-dystroglycan, latrophilins, and cerebellins [reviewed by ref. 1,2]. Neurexins also act cell-autonomously to mediate presynaptic active zone assembly and function[3,4]; this activity relies on the short intracellular domain of neurexins, which interacts with synaptic vesicle exocytosis proteins, e.g., synaptotagmin, and PDZ-domain proteins such as CASK and Mints[5–7]. Mutations in genes encoding neurexins have been implicated in various neurodevelopmental and neuropsychiatric disorders, including schizophrenia, autism, and Tourette syndrome[8]. Thus, understanding how neurexins localize and function at the synapse is particularly relevant. This understanding has been very difficult because of the low abundance of neurexins and their >1500

mammalian isoforms[1,2]. Studies using transgenic mice that express selective tagged isoforms did not generate definitive conclusions due to heterologous promoters and potential overexpression artifacts[9,10]. A role for the PDZ binding motif in targeting neurexins to synapses has been derived from studies in tissue culture[11,12]. However, the cellular and physiological importance of the PDZ binding motif in vivo remains unclear.

The *Drosophila* genome has only one neurexin gene, *Nrx-1*, which encodes for a protein similar to mammalian α-neurexins[13]. As in mammals, *Drosophila* Nrx-1 localizes and functions predominantly at the presynaptic terminals of neurons, although a postsynaptic pool has also been reported[14]. Nrx-1 accumulates at the active zones of the glutamatergic neuromuscular junction (NMJ). Loss of Nrx-1 induces partial developmental lethality; the adult escapers are sluggish and have severe locomotor impairments[13]. *Nrx-1$^{null}$* larvae have smaller

[1]Section on Cellular Communication, Eunice Kennedy Shriver National Institute of Child Health and Human Development, NIH, Bethesda, MD, USA. [2]Department of Neuro and Sensory Physiology, University Medical Center Göttingen, Göttingen, Germany. [3]NanoTag Biotechnologies GmbH, Göttingen, Germany. [4]Present address: Centralized Core Research Facility-Microscopy, All India Institute of Medical Sciences, New Delhi, Delhi, India. [5]These authors contributed equally: Rosario Vicidomini, Saumitra Dey Choudhury. ✉e-mail: mihaela.serpe@nih.gov

NMJs, with disorganized pre- and postsynaptic structures and significantly reduced basal neurotransmitter release[13,14]. These deficits are caused not only by the loss of neurexin-mediated trans-synaptic complexes but also by the loss of intracellular interactions implicating the C-terminal PDZ binding motif, including binding to scaffolds like Scribble, Syd-1, and Spinophilin, which promote active zone assembly and synapse maturation[3,15,16]. Nrx-1[null] phenotypes are partly rescued by neuronal overexpression of various Nrx-1 transgenes, including C-terminal tagged Nrx-1 or truncation with no intracellular domain[13,17]. Likewise, most of the cytoplasmic tails of mammalian neurexins are dispensable for neurexin-neuroligin interactions and the formation of heterologous synapses in cultured cells[11,12,18]. Since neurexins are relatively low-abundance proteins, the individual contributions of functional domains are likely obscured in overexpression settings.

Here, we use the Drosophila model system to investigate the in vivo significance of the C-terminal PDZ domain-binding sequence for the Nrx-1 function. To conduct structure-function studies and accomplish reliable detection of Nrx-1 variants in different tissues, we employed a recently described cell biology tool, the ALFA system[19]. This system is composed of a rationally designed epitope tag of only 14 amino acids, the ALFA tag (AT), with no homology in the main animal models, and a nanobody (NbALFA) that binds to ALFA-tagged proteins with low picomolar affinities. These features together with the ability of NbALFA to function as intrabody (recognizing ALFA-tagged proteins when expressed in living cells) have prompted the development of a variety of in vitro cell biology applications, from super-resolution to live detection of tagged proteins[19].

Guided by phylogenetic analysis and secondary structure prediction, we generated ALFA-tagged Nrx-1 variants, including an endogenously tagged Nrx-1[AT] allele, which is indistinguishable from the wild-type control, and a Nrx[ΔPDZ-AT] allele that resembles the Nrx-1[null] mutant. We found that the PDZ binding motif is key to Nrx-1 in vivo surface expression and synaptic localization. A PDZ binding motif provided in trans, via genetically encoded cytosolic NbALFA-PDZ chimera, fully restored the synaptic localization and function of Nrx[ΔPDZ-AT]. Using a genetically encoded cytosolic NbALFA-mScarlet chimera, we conducted compartment-specific detection and live imaging along the motor neuron axons. We found that Nrx-1 co-migrates with Rab2-positive vesicles, which transport several critical presynaptic components, suggesting that coordinated Rab2-mediated transport ensures that optimal levels of Nrx-1 and other synaptic components traffic and are stabilized at synaptic sites.

## Results

### ALFA tag site selection and in vivo detection of Nrx-1 variants

Mammals have three Nrx genes, each of which uses different promoters to generate longer α− and shorter β− and γ− forms. In conjunction with alternative splicing, these three genes generate >1500 isoforms[2,20]. Drosophila Nrx-1 codes for only one protein with the topology of α-neurexins (Fig. 1a): Extracellularly, Nrx-1 contains six laminin-Nx-sex-hormone-binding (LNS) domains with interspersed epidermal growth factor (EGF)-like repeats followed by a glycosylated stalk; the transmembrane domain is followed by a cytoplasmic domain ending with a C-terminal PDZ binding motif[13]. Similar to mammalian neurexins, the extracellular domains of Drosophila Nrx-1 engage in trans-synaptic interactions with postsynaptic partners, including at least two neuroligins, Nlg1 and Nlg2[21–23], to coordinate pre-and post-synaptic development. In vertebrates, these interactions depend on the LNS6 domain and a heparan sulfate chain attached to the stalk[11,24,25]. Inside the cell, Nrx-1 engages in a variety of interactions that impact the assembly and function of active zones. For example, Nrx-1 interacts with the N-Ethylmaleimide-sensitive Factor (NSF), facilitating NSF recruitment at synaptic terminals and promoting SNARE complex disassembly[26]. In the visual system, Nrx-1 maintains columnar organization by binding to and promoting Ephrin clustering[27].

The C-terminal PDZ binding motif has been implicated in a variety of interactions critical for the function of the glutamatergic NMJ. Like mammalian neurexins, Drosophila Nrx-1 recruits the calcium-/calmodulin-dependent serine protein kinase (CASK) at synaptic sites where CASK phosphorylates Nrx-1, modulating its activities[28,29]. In addition, Nrx-1 binds to the PDZ domain of Scribble forming a Nrx-1–Scribble–Pix complex that activates Rac1 and subsequently stimulates assembly of presynaptic F-actin and clustering of synaptic vesicles[15]. Finally, a pair of PDZ domain-containing scaffolding proteins, Syd-1 and Spinophilin, compete for binding to the C-terminal region of Nrx-1: Syd-1 immobilizes Nrx-1 at the active zone and together promote active zone assembly and postsynaptic maturation[3], whereas Spinophilin limits Nrx-1/Syd-1 activities and controls active zone number and function[16].

To visualize different Nrx-1 variants and conduct structure-function studies, we searched for a detection strategy that is independent of protein length and post-translational modifications. We chose the ALFA system because of the small size and picomolar binding affinities between the ALFA tag (AT) and its single-domain nanobody (NbALFA)[19]. Since an additional alpha helix could interfere with the proper folding of extracellular domains or disrupt the rigidity of the stalk, we searched for a suitable AT insertion site within the intracellular domain of Nrx-1. Phylogenetic analysis of intracellular domains of Nrx-1 from various insects showed blocks of highly conserved regions immediately after the transmembrane domain and at the C-terminus (Fig. 1b). In between these conserved regions, the dipteran Nrx-1 proteins have variable insertions of G-rich sequences. We chose to insert the AT within the first G-rich sequence; this site lies within an unstructured region at the beginning of the Ephrin binding region. This AT insertion apparently preserves the overall topology of the entire Nrx1 intracellular domain (Supplementary Fig. 1). Importantly, the tag is flanked by prolines to isolate its alpha-helical secondary structure, which is necessary for the recognition by the NbALFA[19].

To test whether the AT insertion interferes with Nrx-1 activities, we generated tagged and untagged Nrx-1 transgenes integrated at the same chromosomal location (vk37)[30]. A previously described Nrx-1-GFP line, with a C-terminal GFP tag following the PDZ binding motif, was used for comparison[14]; a diagram of the transgenes utilized in this study is shown in Fig. 1a. We expressed these transgenes in motor neurons and compared the distribution of Nrx-1 variants in ventral nerve cord (VNC) and at synaptic terminals (Fig. 1c–j). As expected, the anti-Nrx-1 antibodies decorated the neuropils of control but not Nrx-1[null] mutants. Strong overexpression of untagged Nrx-1 in the motor neurons (with the BG380-Gal4 driver, referred to as MN-Gal4) induced some ectopic expression in lateral and posterior cells of larval VNC and accumulation along the motor neuron axons (Fig. 1e, arrows, and Supplementary Fig. 2). This suggests that excess Nrx-1 traffics to the axonal terminals. Indeed, neuronal overexpression of Nrx-1 produced strong signals at synaptic terminals (Fig. 1f), whereas endogenous Nrx-1 remained undetectable (see below). Similarly, Nrx-1-AT over-expression induced Nrx-1 accumulation along the axons and at synaptic terminals (Fig. 1g, h). Ectopic AT-positive signals were also present in a few lateral cells, but they appeared punctate in contrast with the more diffuse Nrx-1-labeled aggregates distributed throughout the cell soma. These sharp puncta likely reflect the accumulation of excess Nrx-1-AT in the ER and/or Golgi. The level of detail that the ALFA system captured in one single immunohistochemistry step was remarkable and was not matched by the custom anti-Nrx-1 antibodies or by using the Nrx-1-GFP transgene (Fig. 1i, j). Instead, Nrx-1-GFP overexpression triggered a more diffuse accumulation of GFP-positive signals in the neuropil and at synaptic terminals. Excess Nrx-1-GFP also formed aggregates in selective motor neurons (Fig. 1i), albeit the GFP signals mostly followed the normal Nrx-1 expression pattern. Thus, all tested transgenes recapitulate the

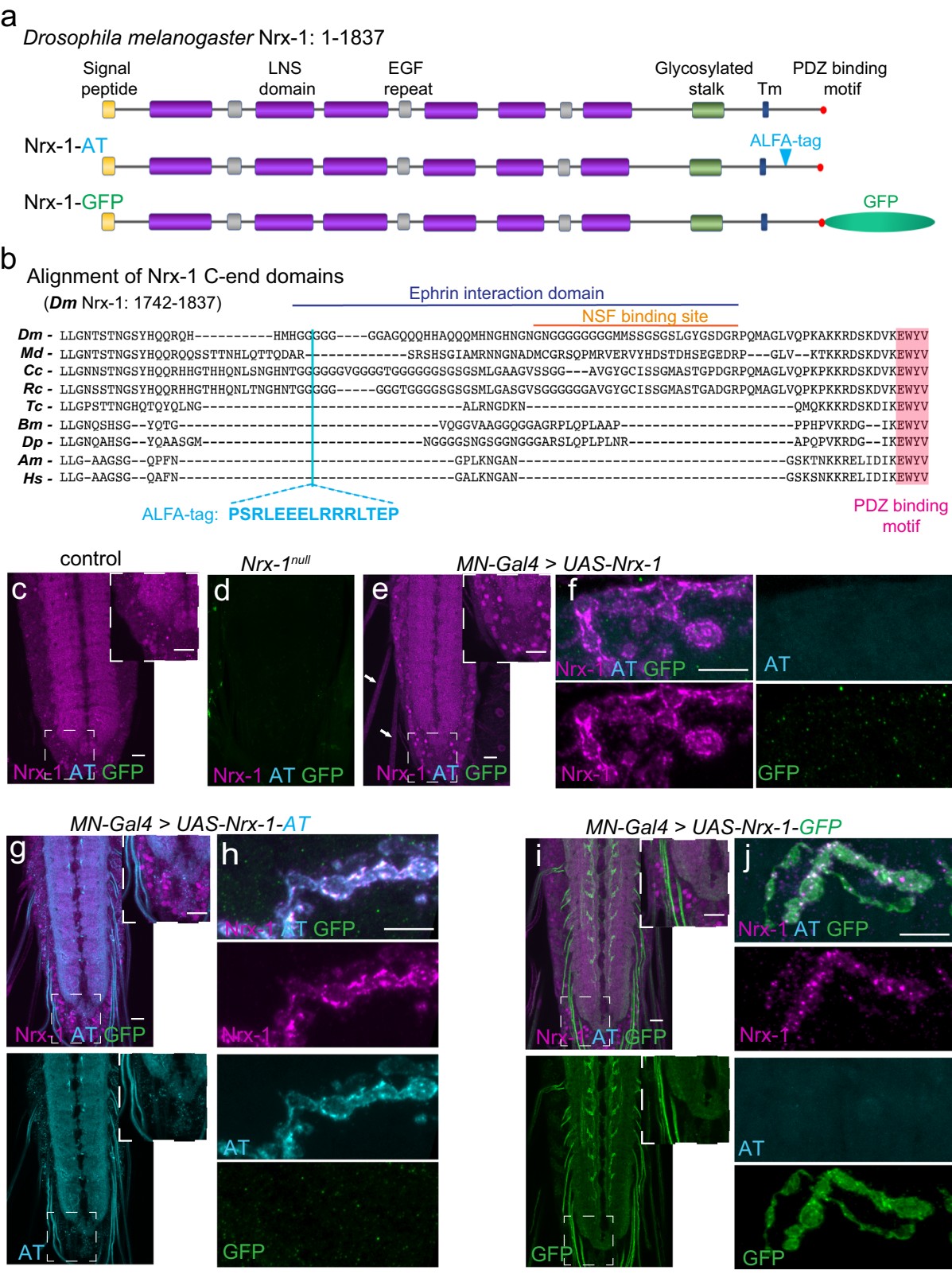

normal Nrx-1 distribution and induce different degrees of Nrx-1 accumulation when overexpressed.

## Insertion of ALFA tag preserves the in vivo Nrx-1 function

We next examined whether Nrx-1-AT can rescue the functional deficits of $Nrx-1^{null}$ mutant NMJ. As previously reported[13,14], the number of synaptic boutons was significantly reduced in the absence of Nrx-1

(by 38% at NMJ6/7, $p < 0.0001$ and by 51% at NMJ4, $p < 0.0001$) compared to control NMJs (Fig. 2a, b, quantified in 2f-h). The $Nrx-1^{null}$ NMJs appeared simpler, more compact and had almost a four-fold reduction in the overall length (to 27%, $p < 0.0001$) relative to the control. Neuronal overexpression of any of the three $Nrx-1$ transgenes restored both the NMJ length and the number of synaptic boutons at $Nrx-1^{null}$ NMJs (Fig. 2c–h). Furthermore, excess of any Nrx-

**Fig. 1 | ALFA tag site selection and in vivo detection. a** Domain organization of the *Drosophila* Nrx-1. The transgenes utilized in this study encode the predominant isoform (Nrx-1-RA) and the corresponding ALFA-tagged and GFP-tagged variants. **b** Alignment of Nrx-1 intracellular domains from various insect species indicates several blocks of conservation, including the C-terminal PDZ binding motif and the *Diptera*-specific G-rich expansions. The ALFA-tag insertion site is marked, together with other interaction domains of *Drosophila* Nrx-1: The Ephrin binding domain (residues 1760–1813) and the NSF binding site (residues 1788–1813). *(Dm- Drosophila melanogaster, Md- Musca domestica, Cc- Ceratitis capita, Rc- Rhagoletis cerasi, Tc- Tribolium castaneum, Bm- Bombyx mori, Dp- Danaus Plexippus, Am- Apis mellifera, Hs- Harpegnathos saltator).* **c**–**j** Representative confocal images of third instar

larvae VNC (**c**, **d**, **e**, **g** and **i**) and NMJs (**f**, **h**, and **j**) of the indicated genotypes labeled for Nrx-1 (magenta), GFP (green), and ALFA tag (cyan) showing the distribution of endogenous Nrx-1 and neuronally expressed Nrx-1 variants. These experiments were repeated three times with similar results. When overexpressed in the motor neurons, Nrx-1 decorates the presynaptic membrane and accumulates in distinct puncta at synaptic terminals. These puncta coincide with the ALFA-positive signals. In contrast, the GFP-positive signals are more diffuse and separated from the Nrx-1-positive puncta. The GFP and ALFA tag signals/channels are shown for all genotypes to illustrate the background signal. Scale bars: 10 μm. Genotypes: control (*w^1118^*), MN > UAS-Nrx-1 *^{tag/no tag}* (BG380-Gal4/+; UAS-Nrx-1 *^{tag/no tag}*/+), Nrx-1*^{null}* (Nrx*^{273}*/Df).

1 variant induced a small but significant increase in bouton number compared to the control (*w^1118^*), consistent with previous findings that Nrx-1 promotes NMJ growth in a dose-dependent manner[13]. Thus, when provided in motor neurons, each Nrx-1 variant (non-tagged, AT-tagged, and GFP-tagged) rescued the morphological deficits of *Nrx-1^null^* NMJ. This conclusion agrees with previous observations that the C-terminal part of Nrx-1 is dispensable for NMJ growth: neuronal overexpression of a Nrx-1 variant lacking most of its intracellular domain (83 out of 121 residues) rescued the growth of *Nrx-1^null^* NMJs[17].

Previous work has demonstrated that loss of *Nrx-1* results in disorganized *Drosophila* NMJ synapses, including increased number of presynaptic active zones per bouton and enlarged postsynaptic receptor fields[13]. Such structural changes cause synaptic transmission defects: the frequency and amplitude of miniature excitatory junction potentials (mEJPs) (which represent the postsynaptic response to the spontaneous release of single synaptic vesicles) are increased compared to control animals, whereas the evoked excitatory junction potentials (EJPs) are reduced, diminishing the locomotor activities of *Nrx-1^null^* adults and larvae[13,17]. We recorded from muscle 6 (abdominal segments 3 and 4) from *Nrx-1^null^* third instar larvae and recapitulated these electrophysiological deficits (Fig. 2I, j) (Supplementary Table 1). Specifically, in the absence of Nrx-1, the EJPs decreased by 36% (from $40.26 \pm 1.39$ mV in control to $25.75 \pm 1.33$ mV in *Nrx-1^null^*, $p < 0.0001$), the mEJP frequency increased by 50% (from $2.21 \pm 0.19$ to $3.33 \pm 0.22$ Hz, $p = 0.0409$) and the mEJP amplitude increased from $1.01 \pm 0.05$ to $1.73 \pm 0.15$ mV ($p = 0.0035$). Consequently, the quantal content (QC) was reduced by 42% (from $37.46 \pm 2.78$ to $15.65 \pm 1.40$, $p < 0.0001$). Neuronal overexpression of *Nrx-1-AT* and non-tagged transgenes completely rescued the EJP amplitude to control levels (Fig. 2k, l, quantified in Fig. 2n–q). These results resemble the full rescue of EJP achieved using a randomly inserted non-tagged *Nrx-1* transgene[13]. In contrast, Nrx-1-GFP did not rescue the EJP amplitude, which remained at $30.25 \pm 1.79$ mV, a level more similar to the *Nrx-1^null^* mutants ($25.75 \pm 1.33$ mV, $p = 0.1519$) and significantly different from the control ($40.26 \pm 1.39$ mV, $p < 0.0001$). Since the *Nrx-1-GFP* transgene effectively rescued the morphological defects of *Nrx-1^null^* mutants, the net levels of synaptic Nrx-1-GFP cannot explain its inability to rescue the NMJ function. Instead, these results suggest that the C-terminal GFP tag alters Nrx-1 activities required for normal synaptic transmission.

Furthermore, both *Nrx-1-AT* and non-tagged transgenes rescued the mEJPs amplitude and the quantal content at *Nrx-1^null^* NMJs (Fig. 2n, q). This result was never achieved before; instead, neuronal or ubiquitous expression of *Nrx-1* could not rescue the mEJP amplitude or the complexity of synaptic specializations in *Nrx-1^null^* mutants[13]. In addition, the Nrx-1-GFP rescued the mEJP amplitude but not frequency, which remained similar with the *Nrx-1^null^* ($3.34 \pm 0.46$ vs $3.33 \pm 0.22$ Hz, $p = 0.0382$) (Fig. 2m–p). Together our data indicate that neuronal expression of Nrx-1-GFP can rescue the morphology but not the synaptic transmission at *Nrx-1^null^* NMJs. In contrast, insertion of the compact AT in a non-conserved cytoplasmic region has a minimal

fingerprint and it preserved both NMJ morphology and synaptic transmission functions of Nrx-1.

## One-step detection of endogenously tagged Nrx-1^AT^

A critical question is how sensitive the ALFA system is in detecting endogenous synaptic proteins. Our in vivo rescue experiments used moderate to high levels of Nrx-1 expression. However, synaptic components, including Nrx-1, are generally low-abundant proteins. Is the ALFA system sensitive enough to detect endogenous levels of Nrx-1 within the dense synaptic sites? To answer this question, we started by editing the endogenous *Nrx-1* locus using CRISPR/Cas-9 methodology (see "Methods"). The resulting *Nrx-1^AT^* edited flies are homozygous, viable, and fertile. More importantly, using NMJ electrophysiological recordings, we could not detect any significant difference between the control (*w^1118^*) and the edited *Nrx-1^AT^* third instar larvae (compare Fig. 3a, b, quantified in c-e). The normal EJPs and quantal content at the *Nrx-1^AT^* NMJs indicate that endogenously tagged Nrx-1^AT^ is expressed at appropriate levels and functions normally at these synapses.

Editing the *Nrx-1* locus also preserved the normal subcellular localization of Nrx-1 (Fig. 3 f–i). First, both the wild-type Nrx-1 and the endogenously tagged Nrx-1^AT^ concentrated at the nerve cord neuropil, a synapse-rich region comprised of dendrites, axons, and glial cell processes; this was illustrated by staining of fixed specimens with anti-Nrx-1 antibodies (compare Fig. 3f, h) and by the co-localization of the AT signals with the Nrx-1 immunoreactivities. Secondly, the AT signals formed puncta that localized to the HRP-marked synaptic boutons (Fig. 3i). This presynaptic distribution matches that of endogenous Nrx-1 immunoreactivities at synaptic terminals (ref. 13 and see below). Notably, under comparable confocal settings, we could capture endogenous Nrx-1^AT^ at larval NMJs using the ALFA system but not with the Nrx-1 C-terminal antibodies (Supplementary Fig. 3).

The proper functioning of edited *Nrx-1^AT^* was further supported by the normal morphology of the third instar larval NMJs (Fig. 3i and Supplementary Fig. 4). As previously observed (ref. 13 and Fig. 2), the number of boutons at larval NMJ is proportional with Nrx-1 activity and varies from $55.40 \pm 2.60$ in *Nrx-1^null^* mutants to $100.90 \pm 1.25$ in strong overexpression settings. The number of boutons at muscle 6/7 in *Nrx-1^AT^* third instar larvae was slightly increased compared to control NMJs ($97.30 \pm 1.73$ in *Nrx-1^AT^* vs $89.80 \pm 0.95$, $p = 0.0180$) and significantly different from *Nrx-1^null^* mutants (Fig. 2 and Supplementary Fig. 4). This result further confirms that editing does not disrupt Nrx-1 function; rather, the AT insertion appears to slightly increase the ability of Nrx-1^AT^ to promote NMJ growth. Together these data indicate that the AT insertion does not disrupt the functionality of tagged Nrx-1.

Importantly, the ALFA system enabled the detection of endogenous Nrx-1^AT^ in only one immunohistochemistry step using the monovalent binder NbALFA conjugated to two fluorophores (FluoTag-X2 anti-ALFA). Indeed, in fixed preparations, the AT signals were unambiguously detected at both *Nrx-1^AT^* larval ventral cords and synaptic terminals (Fig. 3h, i). The AT signals were clear, and the signal-to-noise ratio was exceptional for a single-step staining. The NMJ

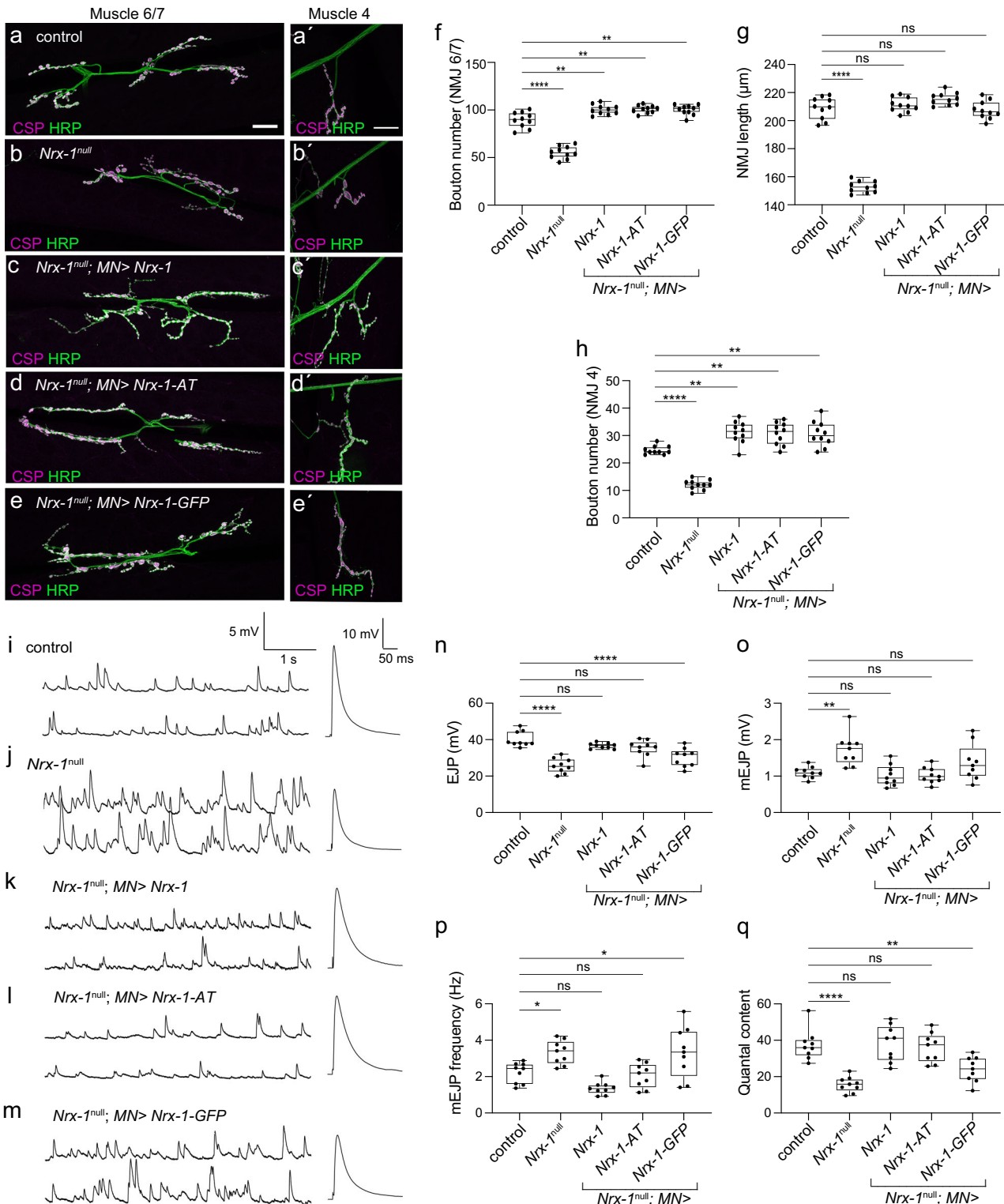

**Fig. 2 | Neuronal expression of untagged and ALFA-tagged transgenes, but not Nrx-1-GFP, rescue the functional deficits at Nrx-1^null NMJs. a–e** Representative confocal images of NMJ 6/7 and NMJ4 from third instar larvae of the indicated genotypes stained for Horseradish peroxidase, HRP (green), which labels neuronal surface, and Cysteine-string protein, CSP (magenta), which labels presynaptic vesicles. These experiments were repeated three times with similar results. Scale bars, 20 μm. **f–h** Quantification of bouton number (**f** and **h**) and NMJ length (**g**) in the indicated genotypes; *n* = 10 for each genotype. **i–m** Representative traces of miniature junctional potentials (mEJPs) and evoked junctional potentials (EJPs) in control, *Nrx-1^null* mutant, and neuronally rescued larvae as indicated. EJPs are

reduced at *Nrx-1^null* mutant NMJs and are restored upon neuronal expression of *Nrx-1* or *Nrx-1-AT* but not GFP-tagged transgenes. **n–q** Quantification of mEJPs amplitude, mEJPs frequency, EJPs amplitude, and Quantal Content of the indicated genotypes; *n* = 9 for each genotype. Data are represented as mean ± SEM (one-way ANOVA with Tukey's multiple comparisons); ****$p < 0.0001$, **$p < 0.005$, ns, $p > 0.05$. The boxes expand from the first to the third quartile, and the whiskers from minimum to maximum values; the center lines mark the mean values. Source data are provided as a Source Data file. Genotypes: control (*w^1118*), *Nrx-1^null* (*Nrx^273/Df*), *Nrx-1^null; MN> Nrx-1^tag/no tag* (*BG380-Gal4/+; UAS-Nrx-1^tag/no tag/+; Nrx^273/Df*).

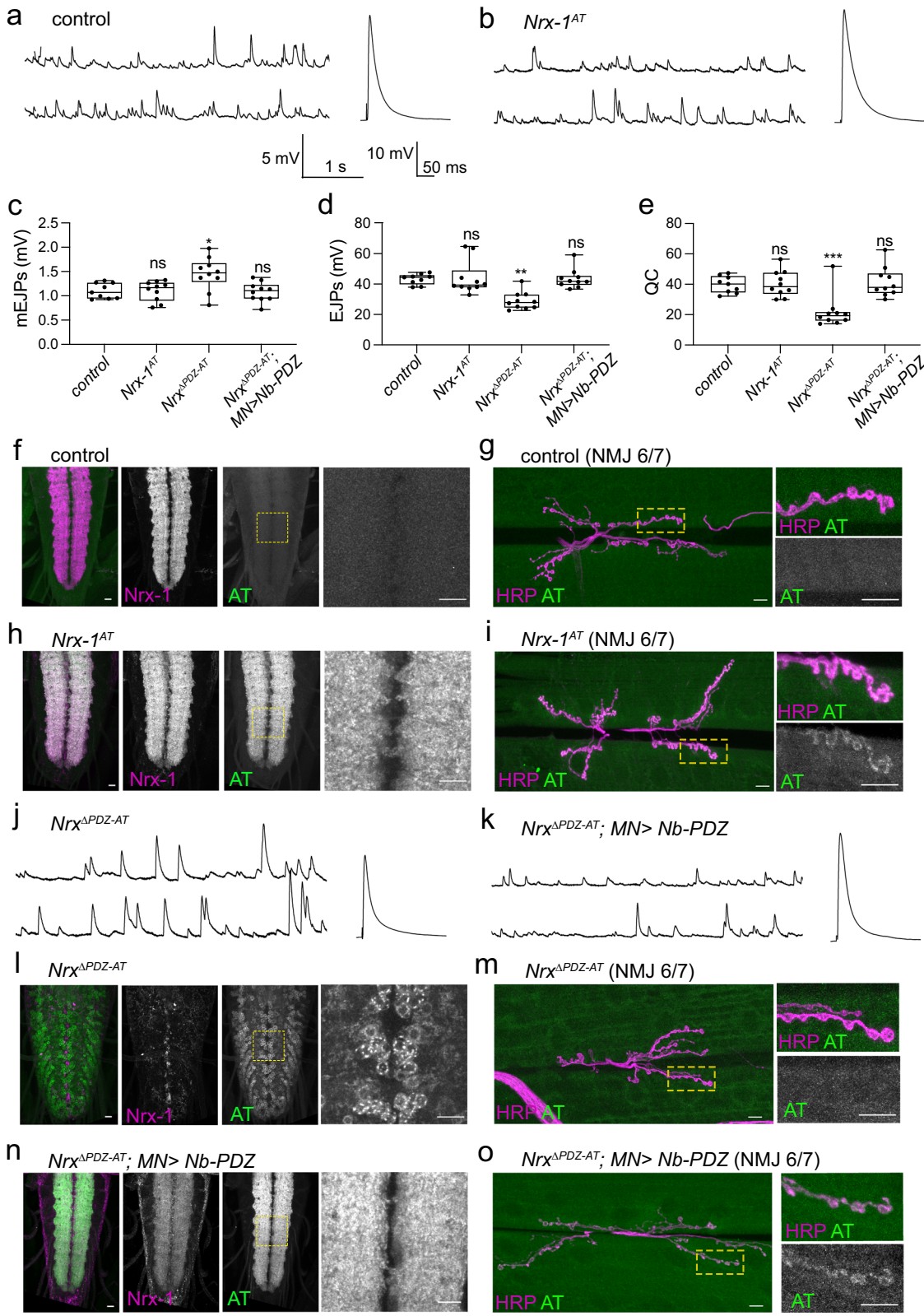

signals were, however, dim, likely reflecting the low abundance of synaptic Nrx-1 and the absence of any additional amplification steps during immunostaining. The AT signals were completely absent in untagged control specimens; also, we found that NbALFA coupled with a variety of fluorophores elicits no background in fixed *Drosophila* tissues. Together these data confirm the expectation that the ALFA system ensures high affinity binding and virtually no background in animal tissues.

### Functional analysis of the Nrx-1 PDZ binding motif

Previous ex vivo studies implicated the C-terminal PDZ binding motif of neurexin in the transport to neuron surface and then to target

**Fig. 3 | Detection and in vivo reconstitution of endogenously edited ALFA-tagged Nrx-1. a, b** Representative traces of mEJPs and EJPs in control and endogenously tagged $Nrx-1^{AT}$ third instar larvae. **c–e** Quantification of mEJP amplitude, EJP amplitude, and Quantal Content in the indicated genotypes; $n = 9$ for the control and 10 for all the other genotypes. **f, i** Representative confocal images of VNCs or muscle 6/7 NMJs of third instar larvae of the indicated genotypes labeled for ALFA tag (green) and Nrx-1 or HRP (magenta). The images were acquired with the same confocal microscopy settings and scaled equally for direct visual comparison. These experiments were repeated three times with similar results; $n = 15$ for (**f–j**) and 5 for (**i**). The edited Nrx-1$^{AT}$ shows the expected Nrx-1 accumulation in neurites. **j, k** Representative traces of mEJPs and EJPs recorded in $Nrx^{\Delta PDZ-AT}$ third instar larvae alone (**j**) or in the presence of neuronally expressed Nb-PDZ chimera (**k**). Deletion of the PDZ binding motif of Nrx-1 causes loss-of-function phenotypes that resemble the $Nrx-1^{null}$ mutant deficits. These phenotypes are completely rescued by neuronal

expression of the Nb-PDZ chimera. **l–o** Representative confocal images of VNCs or muscle 6/7 NMJs of third instar larvae of the indicated genotypes labeled for ALFA tag (green) and Nrx-1 or HRP (magenta). These experiments were repeated three times with similar results; $n = 15$ for (**l–n**) and 5 for (**o**). The absence of a PDZ binding motif shifts the distribution of Nrx$^{\Delta PDZ-AT}$ from the neurites to the motor neuron soma, more specifically in the ER. The addition of the PDZ binding motif in trans restores the cell surface location and normal distribution of reconstituted Nrx-1 (Nrx$^{\Delta PDZ-AT}$/Nb-PDZ) to the neuropil. Scale bars: 10 μm. Data are represented as mean ± SEM (one-way ANOVA with Tukey's multiple comparisons); $****p < 0.0001$, $**p < 0.005$, ns, $p > 0.05$. The boxes expand from the first to the third quartile, and the whiskers from minimum to maximum values; the center lines mark the mean values. Source data are provided as a Source Data file. Genotypes: control ($w^{1118}$), $Nrx^{\Delta PDZ-AT}$; $MN > Nb-PDZ$ (BG380-Gal4/+; UAS-Nb-PDZ/+; $Nrx^{\Delta PDZ-AT}$).

axonal membrane[11,12]. To directly test the relevance of the Nrx-1 PDZ binding motif, we generated an endogenously ALFA-tagged, truncated allele, $Nrx^{\Delta PDZ-AT}$, which lacks the last five residues, KEWYV (Supplementary Fig. 4 and "Methods"). These flies are developmentally lethal, indicating that the PDZ binding motif is essential for Nrx-1 functions in *Drosophila*. Electrophysiological recordings from $Nrx^{\Delta PDZ-AT}$ larval NMJs uncovered functional deficits very similar to those of $Nrx-1^{null}$ mutants (Fig. 3j, quantified in c-e). Specifically, the mean mEJP amplitude was elevated by 32% (to $1.45 \pm 0.11$ mV vs. $1.10 \pm 0.05$ mV in control, $p = 0.0146$) while the EJPs amplitude was reduced by 33% (to $28.95 \pm 1.84$ mV vs. $43.21 \pm 1.16$ mV in control, $p = 0.0008$); consequently, the quantal content was $21.79 \pm 3.48$, a drop of 45% in comparison with control ($39.81 \pm 1.85$, $p = 0.0006$). Overall, the $Nrx^{\Delta PDZ-AT}$ NMJ recordings resembled those observed in $Nrx-1^{null}$ animals (Fig. 2).

The $Nrx^{\Delta PDZ-AT}$ variant could not be detected using polyclonal anti-Nrx-1 antibodies (Fig. 3l and Supplementary Fig. 5). This result likely reflects a loss of relevant epitopes, since these antibodies were raised against a peptide which includes the PDZ binding motif. The AT signals are concentrated in the perinuclear region of the VNC neurons (Fig. 3l), a pattern reminiscent of ER or early Golgi compartments. Using Calnexin 99 A as a marker for ER[31], we confirmed that the AT signals are primarily confined to ER (Supplementary Fig. 6). The AT signals were undetectable at synaptic terminals at $Nrx^{\Delta PDZ-AT}$ larval NMJs or along the motor neuron axons (Fig. 3l, m). Instead, our data suggest that Nrx$^{\Delta PDZ-AT}$ accumulates in the early secretory compartments and cannot traffic to synaptic terminals. This distribution is consistent with the deficits captured by electrophysiological recordings (Fig. 3j) and explains the similarity between $Nrx^{\Delta PDZ-AT}$ and $Nrx-1^{null}$ phenotypes. Like $Nrx-1^{null}$, the $Nrx^{\Delta PDZ-AT}$ NMJs showed a reduced number of boutons (Supplementary Fig. 4), $60.50 \pm 1.97$ boutons at NMJ6/7 and $15.60 \pm 0.81$ at NMJ4. For comparison, control animals had $89.80 \pm 0.95$ boutons at NMJ6/7 and $25.80 \pm 1.14$ at NMJ4 (Supplementary Fig. 4), whereas $Nrx-1^{null}$ had $55.40 \pm 2.05$ and $12.10 \pm 0.57$ boutons respectively (Fig. 2). Taken together, our data demonstrate that the PDZ binding motif controls Nrx-1 surface expression and synaptic distribution.

## Providing the PDZ binding motif in trans rescues Nrx$^{\Delta PDZ}$ function

We next asked whether the ALFA system could deliver the missing PDZ binding motif in trans, facilitating the reconstitution of functional Nrx-1. To this end, we generated genetically encoded NbALFA-PDZ binding motif chimera (*UAS-Nb-PDZ*) and expressed it in the $Nrx^{\Delta PDZ-AT}$ neurons (Supplementary Fig. 4). The resulting animals were viable and fertile, suggesting that neuronal expression of Nb-PDZ restored the essential function(s) of Nrx-1. More importantly, electrophysiological recordings at NMJs with reconstituted Nrx-1 (Nrx$^{\Delta PDZ-AT}$/Nb-PDZ) indicated normal functional parameters (Fig. 3k). The EJPs, mEJPs, and QC at "reconstituted" NMJs were indistinguishable from the wild-type or $Nrx^{AT}$ NMJs (quantified in Fig. 3c–e). Neuronal overexpression of *UAS-*

$Nb-PDZ$ produced no electrophysiological deficits on its own or when overexpressed in wild-type or in the $Nrx-1-AT$ background (Supplementary Fig. 7). This indicates that excess Nb-PDZ chimera causes no detectable synaptic disruption and no apparent interference with other PDZ domain proteins. Also, excess Nb-PDZ had no detrimental effects on the full-length Nrx-1$^{AT}$ function (Supplementary Fig. 7 and Table 1), indicating that an extra PDZ binding motif causes no sequestration/ re-routing of Nrx-1$^{AT}$. These results indicate that the ALFA system could be effectively used as a split system to bring together individual protein domains and achieve functional reconstitution in vivo.

The full recovery of electrophysiological properties predicts that the reconstituted Nrx-1 (Nrx$^{\Delta PDZ-AT}$/Nb-PDZ) should accumulate at presynaptic sites where Nrx-1 functions. Indeed, we found that ALFA-mediated Nrx-1 reconstitution restored the expected Nrx-1 expression pattern (Fig. 3n, o). In larval VNC, reconstituted Nrx-1 (stained with both anti-Nrx-1 antibodies and anti-ALFA) labeled the entire neuropil; at synaptic terminals, reconstituted Nrx-1 formed puncta decorating the synaptic boutons. Nrx-1 reconstitution in trans also rescued the small size of $Nrx^{\Delta PDZ-AT}$ NMJs (Supplementary Fig. 4). The rescued larvae had $97.50 \pm 1.96$ boutons at NMJ6/7 and $30.00 \pm 0.83$ boutons at NMJ4, which is very different than $Nrx^{\Delta PDZ-AT}$ or $Nrx-1^{null}$ alleles and closer to controls ($89.80 \pm 0.95$ boutons at NMJ6/7 and $25.80 \pm 1.14$ at NMJ4). In fact, rescued larvae showed an ~ 11% increase in the number of boutons compared to control (Supplementary Fig. 4, $p = 0.0146$). This mild overexpression phenotype may reflect a small increase/stabilization of Nrx-1 at synaptic terminals.

Together our data indicate that reconstituted Nrx-1 (Nrx$^{\Delta PDZ-AT}$/Nb-PDZ) has normal localization and function. These results also revealed several aspects of Nrx-1 cell biology. First, the AT in Nrx$^{\Delta PDZ-AT}$ must be exposed to the cytoplasm and accessible to the intracellularly provided Nb-PDZ. This implies that Nrx$^{\Delta PDZ-AT}$ is fully translated, stable, and confined to the early secretory compartments where cytosol-exposed AT can bind to cytoplasmic Nb-PDZ. Thus, the PDZ binding motif enables Nrx-1 proper trafficking and surface localization. Of note, the presence of the PDZ binding motif both *in cis*, in Nrx-1(AT), or in trans, in reconstituted Nrx-1 (Nrx$^{\Delta PDZ-AT}$/Nb-PDZ), ensures that Nrx-1 traffics and functions normally at the synaptic terminals.

These results also have several technological implications. First, the ALFA system can be used as a split system to reconstitute functional proteins in trans. Secondly, the presence of NbALFA alone or bound to Nrx-1$^{AT}$ did not interfere with Nrx-1 functions, indicating that the small NbALFA preserves the function of targeted proteins. Finally, the high affinity of the ALFA system was not a barrier in the detection of reconstituted proteins in fixed tissues: We have successfully used fluorescently labeled Atto-NbALFA for detecting reconstituted Nrx-1 (Nrx$^{\Delta PDZ-AT}$/Nb-PDZ). This indicates that at least a subset of synaptic Nrx$^{\Delta PDZ-AT}$ must have dissociated from Nb-PDZ to become accessible for detection.

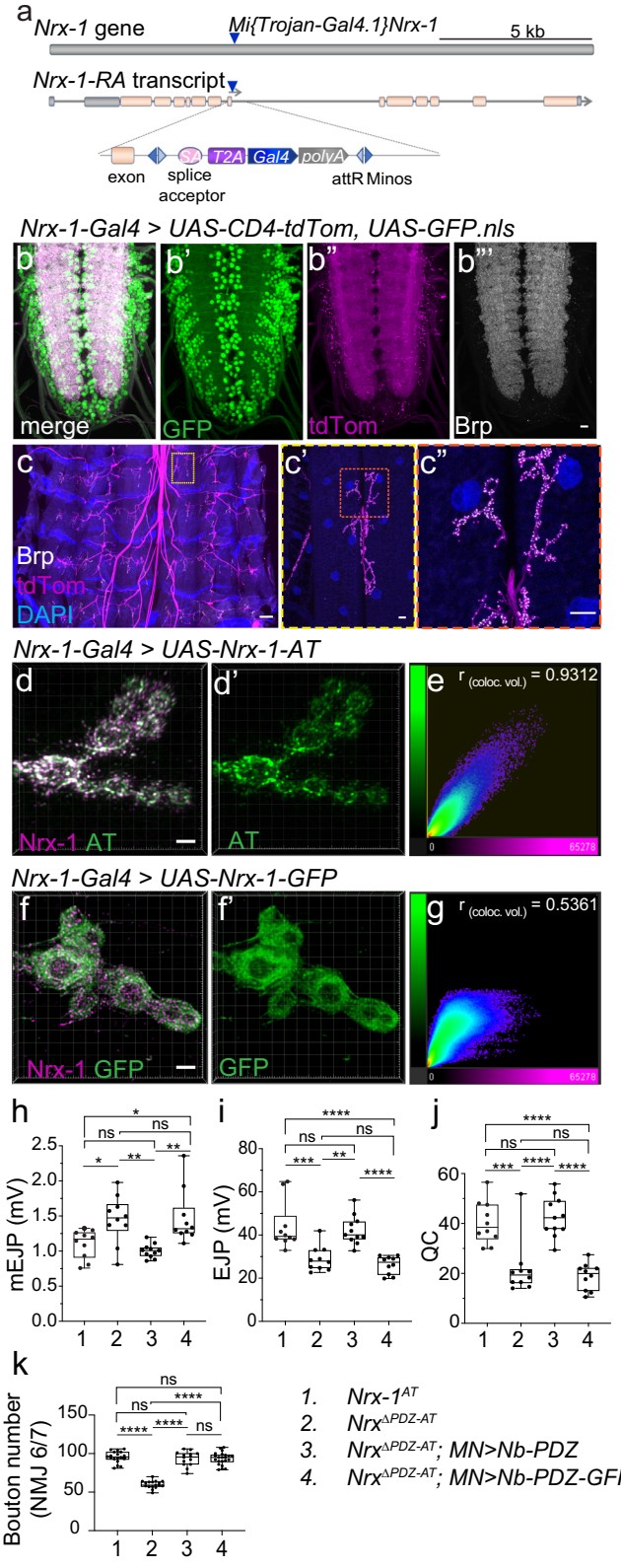

**Fig. 4 | Interfering with the PDZ binding motif disrupts Nrx-1 subcellular distribution. a** Diagram of *Nrx-1-Gal4* organization. A *MiMIC* transposon, *[MI10278]*, inserted right after exon 9 in the *Nrx-1* gene, was converted with a Trojan *T2A-Gal4* cassette inserted in the correct orientation and coding frame. The *T2A-Gal4* in-frame insertion disrupts the expression of Nrx-1 and produces Gal4 instead. **b, c** Confocal micrographs of larval ventral cords (**b**) and synaptic terminals (**c**) of third instar larvae expressing *UAS-CD4-tdTom* and *UAS-GFP.nls* under *Nrx-1-Gal4* control. Brp (white) marks the presynaptic active zones. The *Nrx-1-Gal4* recapitulates the *Nrx-1* endogenous expression pattern. **d–g** Deconvolved STED images of NMJ of third instar larvae expressing Nrx-1-AT (**d**) or Nrx-1-GFP (**f**) and stained for Nrx-1 (magenta) and either ALFA tag or GFP (green). Scatter plots of Nrx-1 and ALFA tag (**e**) or GFP (**g**) signals. Pearson's coefficient in the co-localization volume shows a very high correlation for Nrx-1 and ALFA tags, but a low correlation for Nrx-1 and GFP signals. These experiments were repeated three times with similar results. **h–j** Summary bar graphs showing the mean amplitude of mEJPs (**h**), the mean amplitude of EJPs (**i**), and the QC (**j**) for the indicated genotypes; *n* = 10 for each genotype. **k** Quantification of bouton number for the indicated genotypes; *n* = 14 or more for each genotype. Scale bars: 100 μm (**c**), 10 μm (**b'''**, **c'** and **c''**), and 2 μm (**d** and **f**). Data are represented as mean ± SEM (one-way ANOVA with Tukey's multiple comparisons); ****$p < 0.0001$, ***$p < 0.001$, **$p < 0.005$, *$p < 0.05$, ns, $p > 0.05$. The boxes expand from the first to the third quartile, and the whiskers from minimum to maximum values; the center lines mark the mean values. Source data are provided as a Source Data file.

Furthermore, neuronal expression of Nrx-1-GFP rescues fully the morphological deficits at *Nrx^null^* NMJs but partially the NMJ function [ref. 13 and Fig. 2]. This suggests that the addition of a C-terminal GFP tag may disrupt a subset of Nrx-1 functional interactions and/or interfere with its subsynaptic distribution.

We probed into these two possibilities 1) by comparing the synaptic distribution of Nrx-1-AT and Nrx-1-GFP using super-resolution microscopy and 2) by examining whether the neuronal expression of Nb-PDZ-GFP, like Nb-PDZ, could restore the function of Nrx^ΔPDZ-AT^. For the first set of experiments, we replaced the strong *BG380-Gal4* driver with the *Nrx-1-Gal4* line, which has a *T2A-Gal4* sequence inserted in a frame within the *Nrx-1* locus, *Mi[Trojan-GAL4.1]Nrx-1^MI10278-TG4.1^* [32] (Fig. 4a) and therefore expresses *Gal4* using *Nrx-1* regulatory sequences. This insertion disrupts *Nrx-1* cDNA and induces developmental lethality in homozygous animals. We confirmed the fidelity of the *Nrx-1-Gal4* expression pattern by examining the expression of two reporters, the membrane-bound tdTomato (CD4-tdTom) and the nuclear GFP.nls (Fig. 4b, c). In larval VNC, the GFP.nls reporter captured the pan-neural expression of Nrx-1. The CD4-tdTom labeled the larval VNC neuropil structures in a pattern resembling that of the active zone scaffold Bruchpilot, Brp[33]; CD4-tdTom also marked motor neuron axons and synaptic terminals, where Nrx-1 functions. We next compared the distribution of *Nrx-1-Gal4* expressed Nrx-1-AT and Nrx-1-GFP chimera at synaptic terminals using stimulated emission depletion (STED) microscopy (Fig. 4d–g). Scatter plots of Nrx-1 and AT or GFP signals revealed co-localization with a Pearson's coefficient of 0.93 for Nrx-1 and AT, within a distance of 15.80 ± 9.60 nm, nearing the resolution limit achievable by STED microscopy (Supplementary Fig. 8). In contrast, Nrx-1-GFP appeared diffuse and co-localized only modestly with the Nrx-1 signals (Pearson's coefficient of 0.53). For this experiment, we used heterozygous (*Nrx-1-Gal4/+*) larvae to eliminate one copy of *Nrx-1*; consequently, the Nrx-1 immunoreactivities are primarily due to the added transgenes, with a relatively minor contribution from the endogenous Nrx-1. The GFP spreading may reflect a higher mobility and/or flexibility of presynaptic Nrx-1-GFP; alternatively, the GFP and Nrx-1 signals may represent distinct pools of Nrx-1-GFP proteins distributed to different subcellular locations. These results indicate that interfering with the PDZ binding motif of Nrx-1 has a limited (if any) effect on Nrx-1 trafficking to the synaptic terminals.

To test whether the C-terminal GFP interferes with Nrx-1 synaptic functions, we generated a pair of *UAS-Nb-PDZ* and *UAS-Nb-PDZ-GFP*

## Interfering with the PDZ binding motif alters Nrx-1 synaptic function

The difference between removing or interfering with the PDZ binding motif of Nrx-1 is puzzling: Removing the PDZ binding motif renders Nrx^ΔPDZ^ unable to traffic to the cell surface (Fig. 3), whereas the addition of a C-terminal GFP tag that hinders this PDZ binding motif allows for significant synaptic targeting of Nrx-GFP variants [ref. 13 and Fig. 1].

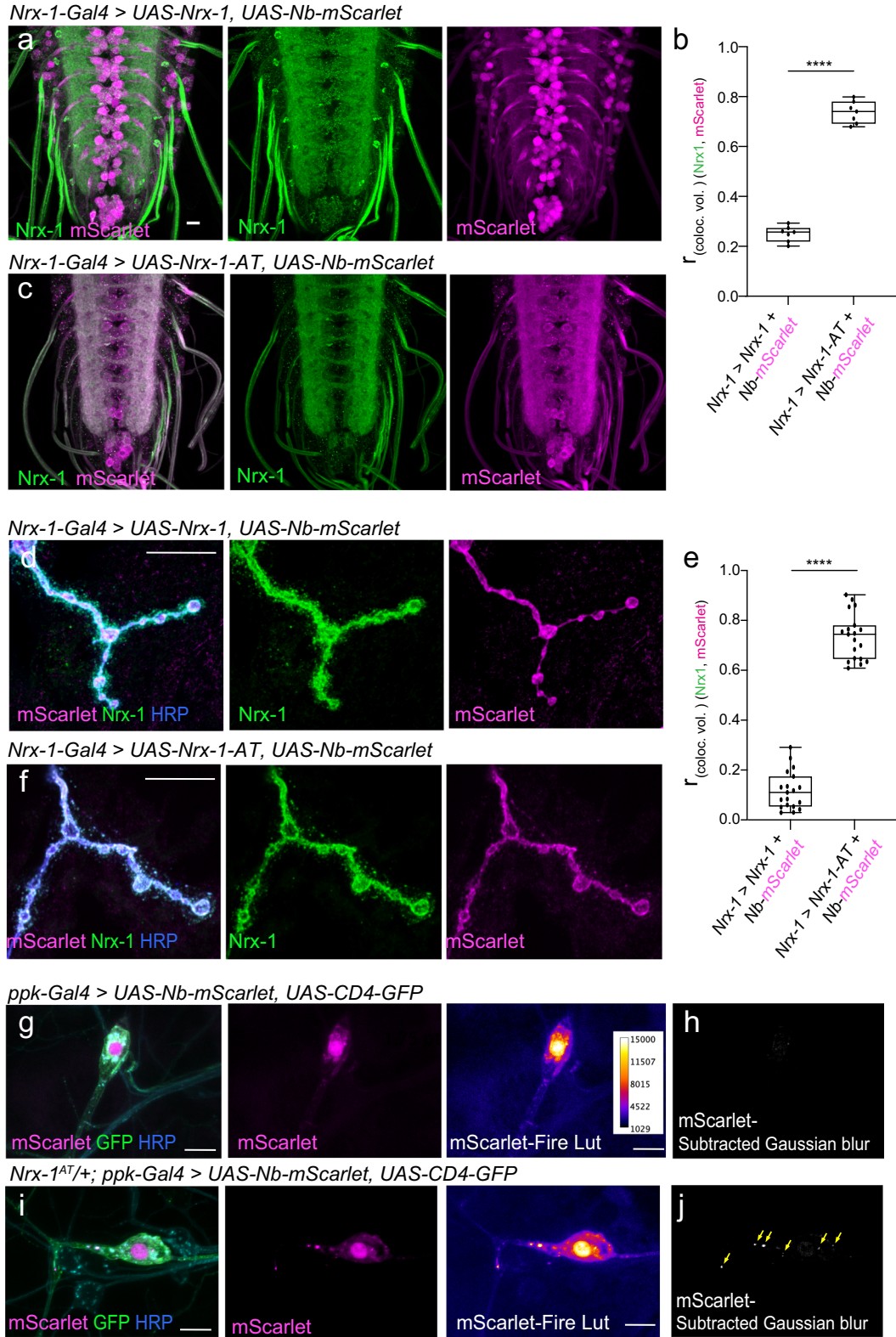

transgenes integrated at the same chromosomal location (vk37)[30] and repeated the *Nrx^{ΔPDZ-AT}* reconstitution experiments from above (Fig. 3). Once again, expression of Nb-PDZ in the *Nrx^{ΔPDZ-AT}* motor neurons rescued the physiological properties of reconstituted Nrx-1 (Nrx^{ΔPDZ-AT}/Nb-PDZ) to normal levels (Fig. 4h–J and Supplementary Table 1). By contrast, the electrophysiology recordings at *Nrx^{ΔPDZ-AT}* and *Nrx^{ΔPDZ-AT}; MN > Nb-PDZ-GFP* NMJs were indistinguishable, indicating that neuronal expression of Nb-PDZ-GFP cannot alleviate the functional deficits at

*Nrx^{ΔPDZ-AT}* NMJs. Neuronal expression of both Nb-PDZ and Nb-PDZ-GFP changed the distribution of AT signals from the ER/early Golgi compartments in *Nrx* to the neuropil (Supplementary Fig. 9). Thus, both transgenes containing the PDZ binding motif restored the cell surface distribution of reconstituted Nrx-1 combinations (Nrx^{ΔPDZ-AT}/Nb-PDZ or Nrx^{ΔPDZ-AT}/Nb-PDZ-GFP), indicating that the addition of the C-terminal GFP does not perturb Nrx-1 surface expression. Furthermore, neuronal expression of either transgene rescued the number of boutons at

**Fig. 5 | Compartment specific detection using genetically encoded cytosolic Nb-mScarlet. a–e** Representative confocal images of VNCs (**a**, **c**) and NMJs (**d**, **f**) from third instar larvae expressing cytosolic Nb-mScarlet together with either untagged Nrx-1 or Nrx-1-AT, as indicated. All transgenes were expressed under the control of *Nrx-1-Gal4*, and the specimens were fixed and labeled for Nrx-1 (green) and mScarlet (magenta). These experiments were repeated three times with similar results; *n* = 7 (**a**–**c**) or 14 (**d**–**f**) per genotype. Both Nrx-1 and Nrx-1-AT distribute to synaptic terminals. To quantify the co-localization between Nrx-1 and mScarlet channels, Pearson's coefficient was calculated within an HRP-selected mask (**b**, **e**). In the absence of an ALFA tag, Nb-mScarlet does not co-localize with Nrx-1 and instead localizes to the neuron soma (see also Supplementary Fig. 10) and throughout the synaptic boutons. In larvae expressing Nrx-1-AT, Nb-mScarlet mirrors the Nrx-1 distribution. (**g**–**j**) Representative confocal images of larval sensory neurons from control (**g**) or *Nrx-1-AT*/+ heterozygous (**i**) third instar larvae expressing cytosolic Nb-mScarlet and membrane-bound mCD4-GFP under the control of *ppk-Gal4*. These experiments were repeated three times with similar results; *n* = 7 for each genotype. The fillets were fixed and labeled for mScarlet (magenta), GFP (green), and HRP (blue). The mScarlet signals (shown in Fire Lut) were further filtered using Gaussian blur and image subtraction (**h**, **j**). Multiple mScarlet-positive vesicles, marked by yellow arrows, are detected in larvae with one copy of *Nrx-1-AT* but not in control. Additional examples are shown in Supplementary Fig. 14. Scale bars: 10 μm. Data are represented as mean ± SEM (unpaired *t* test); ****$p < 0.0001$. The boxes expand from the first to the third quartile, and the whiskers from minimum to maximum values; the center lines mark the mean values. Source data are provided as a Source Data file.

$Nrx^{\Delta PDZ-AT}$ NMJs (Fig. 4k), indicating that reconstituted Nrx-1-GFP ($Nrx^{\Delta PDZ-AT}$/Nb-PDZ-GFP) reaches the synaptic terminals and supports normal NMJ growth. Of note, the AT signals were not detectable at larval NMJs for either transgene, probably because of low expression levels. The rescued NMJ morphology and the physiological deficits observed in animals with reconstituted Nrx-1-GFP ($Nrx^{\Delta PDZ-AT}$/Nb-PDZ-GFP) mirror the phenotypes of $Nrx-1^{null}$ mutants rescued with neuronal Nrx-1-GFP (Figs. 2, 4 and ref. 13). Together, these results demonstrate that the hindrance of the PDZ binding motif impacts Nrx-1 functioning at presynaptic sites, presumably by precluding the binding of synaptic partners that anchor presynaptic Nrx-1 and modulate the active zone function.

## Compartment-specific detection via genetically encoded NbALFA

To refine our understanding of Nrx-1 trafficking and subcellular distribution, we next examined whether the ALFA system could facilitate compartment-specific detection of tagged proteins in vivo. We built genetically encoded NbALFA-mScarlet cytosolic chimera (*UAS-Nb-mScarlet* transgenes) and examined their subcellular distribution relative to ALFA-tagged and untagged Nrx-1. When expressed under the *Nrx-1-Gal4* promoter, both Nrx-1(-AT) variants distributed to presynaptic sites throughout the neuropil (Fig. 5a–c, compare with Fig. 1). In the presence of untagged Nrx-1 (Fig. 5a, b), Nb-mScarlet accumulated in the neuron soma, away from the membrane-bound Nrx-1. In contrast, the presence of Nrx-1-AT produced a striking shift in the subcellular distribution of Nb-mScarlet towards the Nrx-1-marked synaptic terminals (Fig. 5b, c). Co-localization analyses clearly captured this difference: The mScarlet signals showed strong colocalization with the Nrx-1-AT but not with untagged Nrx-1 (Supplementary Fig. 10). This result indicates that the AT/NbALFA binding re-directs the cytoplasmic Nb-mScarlet to the synaptic terminals, where Nrx-1-AT resides. Likewise, Nb-mScarlet distributed diffusely throughout synaptic boutons, filling the synaptic terminals in control NMJs (Fig. 5d). In the presence of Nrx-1-AT, Nb-mScarlet became restricted near the neuronal membranes marked by anti-HRP staining (Fig. 5f). It is important to note that these animals developed normally and produced fertile adults, suggesting that Nrx-1-AT/Nb-mScarlet association does not interfere with normal development. Intracellular recordings further confirmed that neuronal overexpression of *UAS-Nb-mScarlet* by itself or together with *UAS-Nrx-1-AT* does not alter the NMJ functionality (Supplementary Fig. 11 and Table 1). Thus, neuronal excess of Nb-mScarlet does not appear to induce any detrimental effects on protein-protein interactions or impair the NMJ development and function.

*Nrx-1-Gal4* driver is also active in dendritic arborization neurons, which are sensory neurons that cover the body wall of the *Drosophila* larva, and in chordotonal organs, which ensure proprioception and other mechanosensory functions (Supplementary Fig. 12). *Nrx-1 > GFP* is also present in several larval muscles, including thoracic muscles and body-wall muscle 31, present only in the abdominal segment A1. Postsynaptic distribution for Nrx-1 in body-wall muscles

has been previously reported in late *Drosophila* embryos[14,34]. In late larval stages, when animals stop crawling and prepare for pupariation, we observed variable expression of *Nrx-1* reporters in the body-wall muscles, except for muscle 31, which was consistently labeled (in 36 out of 36 muscles examined). The *Nrx-1-Gal4* driven expression was also evident in the enteroendocrine cells of the *Drosophila* gut epithelium [Supplementary Fig. 13 and ref. 34]. These cells are part of the endocrine system, which modulates the function of the fly intestinal tract, similar to pancreatic islets modulating the function of the vertebrate intestinal tract[35]. Neurexins and many other protein constituents of the neurotransmitter exocytotic machinery are expressed in pancreatic β cells and form structures reminiscent of the inhibitory central synapses to mediate insulin granule docking and secretion[36,37]. Expression of *Nrx-1-Gal4* driver in larval enteroendocrine cells suggests conserved molecular and functional strategies across animal endocrine systems. In addition, studies in vertebrate systems indicate that neurexins and neuroligins are present in endothelial and vascular smooth muscle cells throughout the vasculature and are dynamically regulated during vessel remodeling[38]. Likewise, *Nrx-1-Gal4* driven reporters labeled the visceral muscle along the entire intestine (Supplementary Fig. 13). This may reflect a role for Nrx-1 in the exocytosis of biologically active proteins, such as EGF-type ligands, or may be relevant in ensuring proper junctions between visceral muscle and the intestine epithelia.

Sparse Nb-mScarlet expression in $Nrx-1^{AT}$ edited animals may enable the detection of Nrx-1$^{AT}$ in a small subset of cells. To test this possibility, we expressed cytosolic Nb-mScarlet in a subset of sensory neurons and examined its distribution in control and in animals carrying a copy of $Nrx-1^{AT}$ (Fig. 5 i–l and see also Supplementary Fig. 14). In control neurons, mScarlet signals were diffuse and easily eliminated by Gaussian filtering (see "Methods"); in contrast, in the presence of $Nrx-1^{AT}$, we observed mScarlet-positive puncta in the proximity of cell bodies, suggesting that Nb-mScarlet associates with Nrx-1$^{AT}$-carrying vesicles. Together, these data highlight the tremendous potential of the ALFA system for the detection of complex proteins in specific cells and subcellular compartments.

## Live imaging of Nrx-1 axonal transport

The normal synaptic distribution of reconstituted Nrx-1 ($Nrx^{\Delta PDZ-AT}$/Nb-PDZ) complexes and the successful detection of Nrx-1$^{AT}$ in specific cells and at synaptic terminals raised the possibility that similar multicomponent complexes could be followed in vivo, en route to the synaptic terminals. We tested this possibility using the *UAS-Nb-mScarlet* transgenes described above and examining the live distribution of mScarlet along intact neuron bundles in *Drosophila* third instar larvae; these bundles contain motor neurons, projecting away from VNC, and sensory neurons, projecting towards VNC. In untagged Nrx-1 control, axons filled uniformly with mScarlet signals, with no detectable vesicles or larger packages of mobile mScarlet (Fig. 6b and Supplementary Movie 2). In animals expressing *UAS-Nrx-1-AT*, we detected mobile mScarlet-labeled vesicles (Fig. 6a and Supplementary

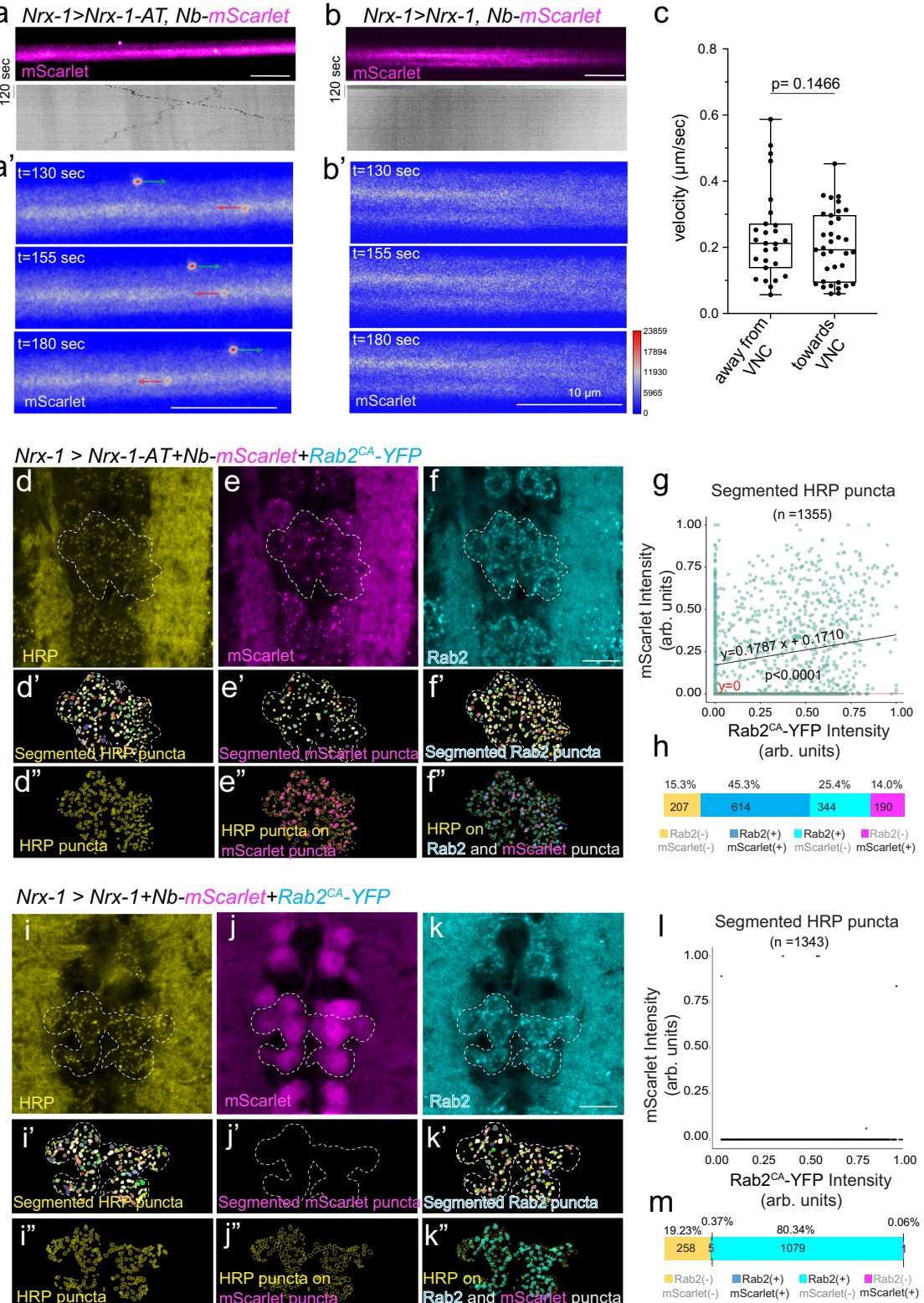

Movie 1). The estimated velocities of vesicles were $0.24 \pm 0.03\,\mu m/s$ ($n = 27$) away from the VNC and $0.20 \pm 0.10\,\mu m/s$ ($n = 35$) towards the VNC (Fig. 6c). For comparison, in primary hippocampal culture, neurexin transport vesicles move at $1.2\text{-}1.7\,\mu m/s$[39]. The difference between live *Drosophila* neurons and hippocampal primary cultures may reflect different neurexin variants/tags, cell types, different culturing conditions (*i.e.* temperature), and experimental settings, in vivo versus ex vivo.

Since neuronal overexpression induced high levels of cytosolic Nb-mScarlet, which may obscure low signal vesicles, we repeated this experiment in a setting where Gal4-induced expression was tightly controlled via Gal80$^{ts}$-mediated repression[40]. At non-permissive temperatures, ubiquitous Gal80 expression (controlled by a *tubulin* promoter) blocks transcription of any *UAS* transgenes; shifting the animals to permissive temperatures inactivates the Gal80 and allows for Gal4-regulated expression of *UAS* transgenes (Supplementary Fig. 15). In this

**Fig. 6 | Cytosolic Nb-mScarlet captures Nrx-1-AT co-localization with Rab2 vesicles in the soma and trafficking along the motor neuron axons. a, b** Live imaging of Nrx-1-AT trafficking along the motor neuron axons. Representative snapshots and kymographs from time-lapse confocal imaging of axons co-expressing *UAS-Nb-mScarlet* with either *UAS-Nrx-1-AT* (**a**) or *UAS-Nrx-1* (**b**) under the control of *Nrx-1-Gal4*. No mScarlet-positive puncta/vesicles are detectable in the absence of the ALFA tag. Vesicles move away (green) or towards the VNC (red) with similar mean velocities (**c**). N = 18 individual larvae were imaged over three different days; moving vesicles were detected and captured in two or three individual bundles per animal. Data are represented as mean ± SEM (unpaired *t* test). The boxes expand from the first to the third quartile, and the whiskers from minimum to maximum values; the center lines mark the mean values. **d–h** Representative confocal images of VNC from third instar larvae expressing *Nrx-1-AT, Nb-mScarlet,* and *Rab2$^{CA}$-YFP* under the control of *Nrx-1-Gal4*. The specimens were fixed and labeled for mScarlet (magenta), YFP (cyan), and HRP (yellow). Maximum projections of the z-planes containing the motor neurons soma are shown and the regions of interest comprising several motor neurons are marked. Vesicles identified by segmentation are shown in the lower panels and in the expanded Supplementary Fig. 17. Overlap of HRP-positive puncta on the combined mScarlet- and Rab2-positive puncta (**f″**) highlights the composition of HRP-positive puncta. **g** Scatter plot indicating mScarlet and Rab2 intensity in each HRP puncta enabled the analysis of puncta composition (**h**) 45% of the HRP puncta (614/1355 total) are positive for both mScarlet and Rab2. Linear regression and non-linear fit tests were used. The null hypothesis was tested using an F-test: F = 53.02 (Df = 1, Dfd = 1353), slope (0.1306 − 0.2269), Y-intercept (0.1507 − 0.1913) 95% confidence, *p* < 0.0001. **i–m** Confocal images and puncta analysis in VNCs from third instar larvae expressing untagged *Nrx-1, Nb-mScarlet,* and *Rab2$^{CA}$-YFP* under the control of *Nrx-1-Gal4*. The analyses followed the workflow described above and revealed only 0.37% overlap among HRP, mScarlet, and Rab2 (5 vesicles /1343 total). F-test: F = 0.501 (Dfn = 1, Dfd = 1341), slope (− 0.006971 − 0.01484), Y-intercept (− 0.006971 − 0.01484) 95% confidence, *p* = 0.4331. Scale bars: 10 μm. Source data are provided as a Source Data file.

system, the mScarlet background was significantly reduced allowing for clear detection of mobile vesicles in the proximal regions of motor neurons in animals co-expressing Nb-mScarlet and Nrx-1-AT (Supplementary Fig. 16 and Movies 3–6). In contrast, no moving particles were visible in control, untagged Nrx-1 animals.

Recent *Drosophila* studies uncovered Rab2-marked vesicles moving along the motor neuron axons anterogradely and retrogradely[41,42]. The Rab2 vesicles originate from the trans-Golgi network and enable the trafficking of presynaptic components at the fly NMJ. Likewise, mammalian neurexins comigrate with synaptic vesicle protein transport vesicles in primary hippocampal neurons[39]. The similarities between the movement of Rab2 vesicles and the Nrx-1-AT/mScarlet packages prompted us to test whether *Drosophila* Nrx-1 synaptic trafficking could be Rab2-mediated. To this end, we labeled the transport vesicles with a constitutively active (Q65L) YFP-tagged Rab2 protein[42] and compared the nature of the Rab2- and Nrx-1-AT/Nb-mScarlet-positive puncta in fixed specimens. Live imaging with three different *UAS* transgenes (*UAS-Nb-mScarlet, UAS-Rab2$^{CA}$-YFP,* and *UAS-Nrx-1-AT*) yielded very few puncta along the axons. We used anti-HRP antibodies to stain all neuronal membranes[43] and then isolated and analyzed intracellular puncta within motor neurons soma in larva expressing Nrx-1-AT (Fig. 6d–f and Supplementary Fig. 17 and "Methods"). We limited our analyses to type Ib motor neurons and excluded the type Is, which showed Nb-mScarlet expression but no detectable Rab2$^{CA}$-YFP for reasons that we do not understand. We used machine learning algorithms to isolate individual vesicles and examine the mScarlet and Rab2 intensity in each HRP puncta and found that 45% of the HRP puncta (614/1355 total) are positive for both mScarlet and Rab2 (Fig. 6f–j). Overall, 75% of the vesicles containing Nrx-1-AT/Nb-mScarlet are also Rab2-positive. The absence of Rab2 signals in the remaining 25% of the vesicles, together with the type Ib-specific expression of Rab2 noted above, suggest that additional mechanism(s) may control Nrx1 trafficking at synaptic terminals. In contrast, in larvae expressing untagged Nrx-1, Nb-mScarlet, and Rab2$^{CA}$-YFP, only 0.37% of HRP puncta were mScarlet- and Rab2-positive (5 vesicles /1343 total) (Fig. 6i–m). Together these results indicate that a large fraction of Nrx-1 co-localizes with Rab2-positive vesicles and may migrate together to synaptic terminals.

## Discussion

Here we address the in vivo role of the PDZ binding motif in the subcellular distribution and function of *Drosophila* Nrx-1. Using the ALFA system, we detected endogenous levels of Nrx-1$^{AT}$ and Nrx$^{ΔPDZ-AT}$ and demonstrated that the PDZ binding motif is key to Nrx-1 exit from ER, traffic to synapses, and function. In contrast, interfering with the PDZ binding motif primarily impacts Nrx-1 activities at synaptic terminals. Importantly, we showed that the ALFA system can operate as a split system, permitting the functional reconstitution of Nrx-1 (from Nrx$^{ΔPDZ-}$

$^{AT}$ and Nb-PDZ). We captured Nrx-1-AT/Nb-mScarlet vesicles moving along the motor neuron axons and demonstrated that 75% of the Nrx-1 transport vesicles are also Rab2 positive.

Our discoveries were aided by the small and versatile ALFA system, a system suitable for super-resolution detection, subcellular and compartment-specific localization. Since the NbALFA retains its high AT binding affinity inside the cell, the ALFA tag/nanobody pair emerges as a powerful conditional reconstitution system and split methodology. Importantly, chromophore-coupled NbALFA produces images with neglectable background staining in *Drosophila* tissues, and we anticipate a similar result for other model organisms.

Most dynamic studies to date monitor overexpressed fluorescently tagged proteins[44–46]. However, overexpression induces gain-of-function phenotypes, and the addition of large fluorescent tags often alters the function of tagged proteins. The ALFA system solves many of these problems by providing the fluorescent moiety in trans, via a genetically encoded nanobody-fluorescent protein chimera. Here we focused on Nrx-1 localization in larval neurons, but Nrx-1 is expressed in many other tissues, including muscle 31 (Supplementary Figs. 12, 13). Thus, at NMJ-31, Nrx-1 may function presynaptically, postsynaptically, or both; compartment-specific expression of Nb-mScarlet in *Nrx-1$^{AT}$* could differentiate between various Nrx-1 pools. The distribution and levels of tagged proteins could be further manipulated by anchoring the NbALFA to specific cellular structures or by controlling the NbALFA function with optogenetics or chemogenetics switches[47–49].

Notably, the ALFA tag/nanobody pair can also function as a split, conditional reconstitution system, suitable for comprehensive structure-function studies. The small size of the ALFA system components and the ability of NbALFA to bind to ALFA-tagged proteins both inside and outside the cell confers exceptional versatility to this split system. For example, by combining the *Nrx$^{ΔPDZ-AT}$* (loss-of-function) allele with binary expression systems, one could independently control Nb-PDZ expression in muscles or motor neurons and accomplish compartment-specific Nrx-1 reconstitution. Binary expression systems for *Drosophila* studies include GAL4/UAS[50], LexA/LexAop[51], and the Q-systems[52], which have been adapted for use in mammalian cells, zebrafish, worms, and mosquitos[53–56]. Multiple conditional expression systems are also available for the mouse model[57].

Using the ALFA tag/nanobody system, we demonstrate that the PDZ binding motif is required for critical steps during Nrx-1 life cycle, including (1) regulation of Nrx-1's exit from ER, (2) control of Nrx-1 trafficking and (3) function at synaptic terminals. Several lines of evidence support these conclusions. First, in the absence of the PDZ binding motif, the Nrx$^{ΔPDZ-AT}$ variant remains confined to the ER/early secretory compartments (Fig. 3). Likewise, when transfected in hippocampal neurons, mammalian neurexins lacking the PDZ binding motif cannot reach the plasma membrane and remain concentrated in the ER[11,12]. Mammalian neurexins contain a highly conserved basic

motif implicated in ER retention[11], whereas *Drosophila* Nrx-1 contains a typical di-leucine ER retention motif (reviewed in ref. 58). In addition, other sequences may be involved in trafficking of Nrx-1 since neuronal expression of a Nrx-1 variant with no intracellular residues can rescue the NMJ growth of *Nrx-1^null*, presumably by reaching the synaptic terminals and establishing productive interactions with Nlg1[17,21]. Likewise, most of the cytoplasmic tails of mammalian neurexins are dispensable for neurexin-neuroligin interactions and the formation of heterologous synapses in cultured cells[11,12,18].

Second, the role of the Nrx-1 PDZ binding motif in ensuring proper transit through the ER/Golgi network appears to be fairly flexible, as this motif can function in trans (this study) or can be replaced by a foreign PDZ binding sequence[11].

Third, interfering with the PDZ binding motif of neurexins (*i.e.* by addition of a C-terminal GFP tag) is less disruptive than removing this motif, since a fraction of neurexin-GFP variants reach the neuronal membrane (this study and[12–14]). Addition of C-terminal GFP tags to mammalian or *C. elegans* neurexins impaired the polarized distribution of neurexins, which became homogeneously distributed throughout the neuron[4,12]. In *Drosophila*, neuronal overexpression of GFP-tagged Nrx-1 generated a diffuse distribution throughout the presynaptic boutons (Fig. 1). The synaptic pools of Nrx-1-GFP or reconstituted Nrx-1-GFP (Nrx^ΔPDZ-AT^/Nb-PDZ-GFP) were sufficient to ensure normal NMJ growth but failed to rescue the electrophysiological deficits of *Nrx-1^null* NMJs [Figs. 2, 4 and ref. 13]. This indicates that the addition of a C-terminal tag obstructs Nrx-1-dependent intracellular interactions crucial for active zone assembly and/or function.

Fourth, Nrx-1 transit through the ER/Golgi complex seems to be limited since overexpressed Nrx-1 variants accumulate in the cell body within the VNC (Fig. 1). We visualized live Nrx-1 trafficking along the motor neuron axons and found that the vesicle velocities (0.2–0.3 μm/s) fall within the published range of fast vesicular transport[59]. Nrx-1 axonal trafficking seems to be mediated by Rab2, a Golgi-residing small GTPase, which enables trafficking of presynaptic components at the fly NMJ, including the active zone scaffold proteins ELKS/Bruchpilot and RIM-binding protein[33,41,60]. Like many synaptic proteins, normal NMJ function requires an optimal level of Nrx-1: Neurotransmission is reduced in larvae lacking Nrx-1, but also in animals with excess presynaptic Nrx-1[13,61]. Optimal levels may be coordinated by coupling Nrx-1 transport with that of other presynaptic components. Likewise, in primary hippocampal culture, Nrx co-migrates with synaptic vesicle protein transport vesicles[39]. During early development, mammalian neurexins traffic directly to the axons. As neurons mature, neurexins appear to follow an indirect trafficking pathway via SorCS1-mediated sorting in dendritic endosomes and transcytosis to the axonal surface[62]. The *Drosophila* genome does not contain a *SorCS1* gene, thus, Nrx-1 synaptic targeting may follow a direct pathway.

It was proposed that additional C-terminal interactions enable a postsynaptic role for Nrx-1 in recruiting selective subtypes of glutamate receptors at postsynaptic densities[14]. Our detailed characterization of the *Nrx-1* expression pattern provides additional support for Nrx-1 functioning outside the presynaptic compartment. Besides expression in the postsynaptic body-wall muscles, we uncovered *Nrx-1-Gal4* driven expression in visceral muscles surrounding the entire intestine (Supplementary Figs. 12, 13). This is reminiscent of the neurexins-neuroligins presence in vertebrate endothelial and vascular smooth muscle cells throughout the vasculature[38]. More importantly, we found expression of *Nrx-1* reporters in the enteroendocrine cells of the *Drosophila* gut epithelium. Similar to other neuroendocrine cells[36,37], fly enteroendocrine cells may use Nrx-1 to facilitate secretory granule docking and hormone release. The rich expression pattern in flies and vertebrates suggests a more general role for neurexins in the integrity of cellular junction and the exocytosis of biologically active proteins.

Localizing endogenous neurexins or other low abundant, junctional proteins and dissecting their functional domains in vivo have been technically very difficult. Our study demonstrates that the ALFA system offers effective solutions to overcome these challenges.

## Methods

### Molecular constructs and fly stocks

To generate the *UAS-Nrx-1* and *UAS-Nrx-1-AT* transgenes, the coding sequence for Nrx-1 and Nrx-1-AT were assembled from the available cDNA, LP14275 (SalI/HindIII+ HindIII/NotI fragments), combined with PCR amplified intracellular domains (digested NotI/XbaI). The primers used are as follows:

Nrxi-F: CTCAAGTCGAATGGCGATCGTGGC;

Xba-Nrxi-Rev: GGTTCCTTCACAAAGATCCTCTAGACATACCACT CCTTGACGTCCTTGG;

Nrx-AT-Rev: CAAGCGTCTACGCAATTCCTCTTCCAGGCGGCTGG GGCCACCATGCATATGGTGCTG;

Nrx-AT-For: CCTGGAAGAGGAATTGCGTAGACGCTTGACCGAAC CAGGTGGCGGAGGCGGTGGAGGTGC.

The wild-type intracellular domain was amplified using the Nrxi-F/ XbaI-Nrxi-Rev primer pair. The ALFA tag sequence was introduced via Gibson assembly using two overlapping PCR products (Nrxi-F/Nrx-AT-Rev and Nrx-AT-For/ XbaI-Nrxi-Rev). All sequences were verified by DNA sequencing. MacVector sequence analysis application (version 18.5) was used for sequence alignment and secondary structure prediction.

The modified *Nrx-1* and *Nrx-1-AT* coding sequences were next introduced downstream 20xUAS in plasmid pJFRC7-20XUAS-IVS-mCD8::GFP, a gift from Gerald Rubin (Addgene plasmid # 26220)[40]. The *UAS-Nrx-1* and *UAS-Nrx-1-AT* transgenes were generated by injection of pJFRC7-Nrx-1 and pJFRC7-Nrx-1-AT at docking site vk14 (BDSC 9733) and vk37 (BDSC 9752)[30]. Lines carrying vk37 docked transgenes were much healthier than those with transgenes docked at vk14 and were chosen for further experiments.

Subsequent analysis of *Nrx-1* mRNA expression was performed via qPCR: total RNA was extracted from three third instar larvae for each genotype using an RNAaqueous micro kit (Invitrogen) following the manufacturer's protocol. The reverse transcription was done with SuperScript III Reverse Transcriptase (Invitrogen), and qPCR was performed using PowerUp SYBR Green Master Mix on a StepOnePlus Real-Time PCR System (Applied Biosystems), and the results were analyzed using GraphPad Prism v9.4.1. The primers used for qPCR were as follows:

Nrx-1_qPCR_For: CAAGACGGAGAGCGAGAAGG;
Nrx-1_qPCR_Rev: GCCTCCTCCATTTCCATTACC;
RP49_qPCR_For: CCGCTTCAAGGGACAGTATC;
RP49_qPCR_Rev: GACAATCTCCTTGCGCTTCT.

The *UAS-Nb-mScarlet* transgenes were generated by subcloning the (EcoRI/NotI) NbALFA-mScarlet sequence from pNT1004[19] in pUAST[50] followed by germline transformation (Rainbow Transgenic Flies). To generate *UAS-Nb-PDZ*, the (BamHI/NotI) mScarlet sequence from pNT1004 was first replaced with the PDZ binding motif sequence using the following pair of annealed primers:

PDZ-For- 5'P-GATCCACTGGGGAAAACGTCGCAACCATGGTGAAG GAGTGGTATGTGTAAAGC;

PDZ-Rev-5'P-GGCCGCTTTACACATACCACTCCTTCACCATGGTTG CGACGTTTTCCCCAGTG.

The (EcoRI/NotI) Nb-PDZ coding sequence was next moved in pUAST for germline transformation as described above. Both plasmids, pUAST-Nb-mScarlet, and pUAST-Nb-PDZ, carry 20x UAS elements.

To generate pJFRC7-Nb-PDZ, the (SalI/XbaI) Nb-PDZ fragment was directly subcloned into pJFRC7-20XUAS-IVS-mCD8::GFP; to make pJFRC7-Nb-PDZ-GFP, the Nb-PDZ fragment was PCR amplified,

digested SalI/BglII, then subcloned into pJFRC7-20XUAS-IVS-mCD8::GFP. The primers used for PCR amplification:

PDZ-GFP-For: CTGCAGTCGACGGTACCGCGACCATGGGCTCTGG;
PDZ-GFP-Rev: CGGCCAGATCTACCGCCGCTACCTCCCACATACCACTCCTTCACCATGGTTGCG.

The pJFRC7-Nb-PDZ and pJFRC7-Nb-PDZ-GFP plasmids were introduced at the docking site vk37 (BDSC 9752)[30].

The *Nrx-AT* and *Nrx^{ΔPDZ-AT}* alleles were generated using classic CRISPR/Cas9 methodology as previously described[63]. Briefly, for each allele, two pairs of gRNAs together with either Nrx-AT-donor or Nrx^{ΔPDZ-AT}-donor plasmids were injected in *w^{1118}; [nos-Cas9]attP40/CyO* stock[64] followed by germline transformation (Rainbow Transgenic Flies).

To assemble the donor plasmids, first, the sequence upstream of the ALFA tag insertion site was PCR amplified (Nrx-13829-For/ Nrx-15708-Rev), digested with NotI, and subcloned in a vector backbone isolated from pJFRC7-Nrx-1-AT (SalI-filled in/NotI) to generate an intermediate plasmid. Secondly, the sequence downstream the ALFA tag insertion site was amplified by PCR (Nrx-3UTR-For/Nrx-16472-Rev) and introduced into the intermediate (cut EcoRI/XbaI) via Gibson assembly to generate the Nrx-AT-donor plasmid. Finally, a truncated C-terminal was generated by PCR (Nrx-14874F/Nrx-dPDZ-Rev) and subcloned into Nrx-AT-donor (XbaI/NotI) to generate the Nrx^{ΔPDZ-AT}-donor.

The primers used for gRNAs, PCR, and sequencing were as follows:

gRNA-Nrx-AT sense: 5′- P-CTTCGCGCCTCCGCCACCATGCATA;
gRNA-Nrx-AT antisense: 5′- P-AAACTATGCATGGTGGCGGAGGCGC;
gRNA-Nrx-Stop sense: 5′- P-CTTCGTGGTATGTGTAAGGCGGCA;
gRNA-Nrx-Stop antisense 5′- P-AAACTGCCGCCTTACACATACCAC;

Nrx-13829-For: CTCCTCCACATTTATTCCCTCGATTG;
Nrx-15708-Rev: CTATCTGCTATCGGTTATCTGCTACC;
Nrx-3UTR-For: GGACGTCAAGGAGTGGTATGTCTAGAGGCGGCATGGTCGCAGGGAAATATCG;
Nrx1-16472-Rev: CGAGGCCCTTTCGTCTTCAAGAATTCGAGTGGAAGGGACTAACTCAAAGTAG;
Nrx-14874F: CAGAATTAACTCGATCGAGGAGGAG;
Nrx-dPDZ-Rev: GGTTCCTTCACAAAGATCCTCTAGACGTCCTTGGAGTCGCG.

After injection and germline transformation, a series of dsRed marked chromosomes were isolated and molecularly characterized by PCR (QuickExtractDNA, Epicenter) and sequencing. The putative *Nrx-AT* and *Nrx^{ΔPDZ-AT}* alleles have been re-confirmed by sequence analysis. All fly lines and reagents generated in this study are available upon request from the lead contact without restriction.

Other fly stocks used in this study were as follows: *UAS-Nrx1-GFP*[14], *Mi[Trojan-GAL4.1]Nrx-1^{MI10278-TG4.1}* [32] (BDSC 67489), *Nrx-1^{273}*[13], *tub-Gal80[ts]* (BDSC 7108), *UAS-CD4-tdTom* (BDSC 35837), *UAS-GFP.nls* (BDSC 4775), *UAS-Rab2^{CA}-YFP* (BDSC 9760), *ppk-Gal4* (BDSC 32079). The *BG380-Gal4* was previously described[65]. Flies were reared on Jazz-Mix food (Fisher Scientific) at 25 °C (unless otherwise specified) and analyzed at the third instar larval stage.

## Immunohistochemistry

Wandering third instar larvae of the desired genotypes were dissected in ice-cooled $Ca^{2+}$-free HL-3 solution (70 mM NaCl, 5 mM KCl, 20 mM $MgCl_2$, 10 mM $NaHCO_3$, 5 mM trehalose, 5 mM HEPES, 115 mM sucrose)[66]. The samples were fixed in 4% paraformaldehyde (PFA) (Polysciences, Inc.) for 20 min or in Bouin's fixative (Bio-Rad) for 3 min and washed in PBS containing 0.1% − 0.5% Triton X-100 (Supplementary Table 2). Primary antibodies were used at the following dilutions: FluoTag-X2 anti-ALFA (N1502), 1:500[19], FluoTag-Q anti-RFP (N0401), 1:500, and FluoTag-X4 anti-GFP (N0304) 1:500 (NanoTag Biotechnologies), rabbit anti-Nrx-1 (C-terminal), 1:500 (a gift from Janet Richmond)[14], mouse anti-Brp (Nc82), 1:200[67] (Developmental Studies

Hybridoma Bank, DSHB), mouse anti-CSP (6D6), 1:1000[68];(DSHB), mouse anti-calnexin99A (Cnx99A 6-2-1), 1:10 (DSHB), chicken anti-GFP, 1:1,000 (Abcam) and FITC-, Alexa-405 and Alexa 647-conjugated goat anti-HRP, 1:1000 (Jackson ImmunoResearch Laboratories, Inc.). Note that the rabbit anti-Nrx-1 (C-terminal) polyclonal antibodies work best with 3 min Bouin fixation. Alexa Fluor 405-, Alexa Fluor 488-, Alexa Fluor 568-, and Alexa Fluor 647- conjugated secondary antibodies (Thermo Fisher Scientific) were used at 1:200. All samples were mounted in ProLong Gold (Thermo Fisher Scientific).

For all analyses, samples of different genotypes were processed simultaneously and imaged under identical confocal settings in the same imaging session with a laser scanning confocal microscope (CarlZeiss LSM780, 40X ApoChromat, 1.4 NA, oil immersion objective). All images were collected as 0.6 μm optical sections with a step of 0.3 μm along the z-axis. The z-stacks were analyzed with Imaris software (Bitplane) or ImageJ (NIH). Boutons were counted in preparations double labeled with anti-HRP and anti-CSP positive puncta. Morphometric quantifications such as branch length were quantified semi-automatically with Filament algorithm[69].

To quantify the co-localization between different channels, Pearson correlation coefficients were calculated within HRP-selected masks in larval VNCs and at synaptic boutons. To further examine the signal distribution at synaptic boutons, we selected multiple linear regions of interest (ROIs) and plotted normalized fluorescence intensity profiles along the ROIs length. We first assessed the overlap between different profiles. Next, we calculated and plotted the difference between normalized fluorescence intensities (ΔF) and determined the area under the curve (AUC). Low AUC values indicate highly co-localized profiles, whereas high AUC values indicate distinct patterns.

To detect the intracellular Nb-labeled particles, we used Gaussian filtering (ImageJ)[70]. In brief, we generated maximum intensity projections and then applied Gaussian blur to these projections to prepare filtered images that closely reflect the background noise. This step required image convolution with a Gaussian kernel (the radius parameter was set to 3 pixels). The Gaussian blurred images were then subtracted from the maximum intensity projections leaving behind enhanced representations of the Nb-labeled puncta and cellular structures.

Statistical analyses were performed using Prism 9 and the Student *t* test with a two-tailed distribution and a two-sample unequal variance. Error bars in all graphs indicate standard deviation ± SEM. ****$p < 0.0001$; ***$p < 0.001$; **$p < 0.01$; *$p < 0.05$; ns, $p > 0.05$.

## Super resolution microscopy and live imaging

Stimulated Emission Depletion (STED) microscopy was performed using a Leica SP8 microscope (Leica Microsystems), equipped with a white-light laser and pulsed 775-, 660- and 592- nm STED depletion lasers. A 100×/1.4-N.A. oil immersion objective lens (HCX PL APO STED white; Leica Microsystems) was used for imaging. Z stacks were collected at 0.160- μm depth intervals; images were deconvolved using the classical maximum-likelihood estimation algorithm in Huygens Professional software version 15.10.1 and further examined and reconstructed using Imaris software version 9.7.0 (Bitplane). For co-localization analyses, eight linear ROIs were randomly selected, and the luminance profiles for both channels were plotted using the Leica Application Suite (LAS X software, Leica Microsystems). The inter-peak distance of the two (ALFA and Nrx-1) channels was calculated directly using the distance function of LAS X. The results are presented as mean ± SEM.

For live imaging, third instar larva were dissected in $Ca^{2+}$-free HL-3 and immersed in HL-3 with 0.5 mM $Ca^{2+}$ to mimic physiological conditions during observation. The VNCs were imaged using a laser scanning confocal microscope (CarlZeiss LSM780, W Plan /ApoChromat 20x/1.0 DIC) and a plane time series scan mode (zoom 1x and

pinhole 600 μm). The motor neuron bundles were imaged with a 2-photon confocal microscope (CarlZeiss LSM 880, W Plan /ApoChromat 40x/1.0 DIC) and a plane time series scan mode (zoom 3x and pinhole 600 μm). In both cases, the frame size was 1024 × 1024 pixels with a scan speed of $2.6 \times 10^6$ pixels/second; each frame was captured every 5 s for a total of 10 min.

The z-stacks were analyzed with Imaris software (Bitplane) or ImageJ (NIH).

### Puncta segmentation and analysis
To analyze and quantify the puncta/vesicles, we developed an approach that incorporates machine learning algorithms, the Weka classifier, and Stardist (2D), an image segmentation plugin for ImageJ/ Fiji designed to detect subcellular structures[71]. Starting from a maximum projection of z-planes containing the motor neurons soma, regions of interest labeled for both Nb-mScarlet and Rab2[CA]-YFP were manually selected. The inverse of the selected regions was created and deleted in all channels. We then selected one channel, adjusted the threshold to capture the puncta of interest, and created a mask. For training the Weka classifier, puncta/vesicles of interest were manually selected and classified into two categories: class 1 for puncta (at least 30 counts) and class 2 for noise. The trained classifier calculated probabilities for each punctum/vesicle; a threshold of 0.6 was implemented. We next applied Stardist to identify each punctum as a single object, manually adjusting the threshold values to achieve optimal segmentation. This workflow was repeated for each channel separately.

For quantification, the HRP segmented puncta were superimposed on the Rab2 and then on the mScarlet puncta. The Rab2 and mScarlet fluorescence values in each HRP segmented puncta were measured and normalized. 1343 and 1355 HRP-positive puncta were isolated from 5 control and experimental VNCs, respectively.

For data visualization, a scatter plot analysis was performed in RStudio (v1.4.1717 on top of R v4.2.3), using the ggplot2, ggExtra, RColorBrewer, cowplot, and ggpubr libraries. The linear regression and non-linear fit test with Null hypothesis (YIntercept = 0) and Alternative hypothesis YIntercept unconstrained were obtained in GraphPad Prism v9 using the imported data. The composition of HRP-positive puncta was determined using a custom-made R code which calculates the number of HRP-positive puncta with zero and (mScarlet and/or Rab2) positive intensity values (Supplementary Information file).

### Electrophysiology
Standard larval body wall muscle preparation first developed by Jan and Jan (1976)[72] was used for electrophysiological recordings[73]. Wandering third instar larvae were dissected in physiological saline HL-3 saline[66], washed, and immersed in HL-3 containing 0.6 mM Ca[2+] using a custom microscope stage system[74]. We choose to use 0.6 mM Ca[2+] to facilitate comparison with previous data sets[13]. The nerve roots were cut near the exiting site of the ventral nerve cord so that the motor nerve could be picked up by a suction electrode. Intracellular recordings were made from muscle 6, abdominal segments 3 and 4. Data were used when the input resistance of the muscle was > 5 MΩ, and the resting membrane potential was between − 60 mV and − 80 mV. The input resistance of the recording microelectrode (backfilled with 3 M KCl) ranged from 20 to 25 MΩ. Muscle synaptic potentials were recorded using an Axon Clamp 2B amplifier (Axon Instruments) and analyzed using pClamp 10 software. Following motor nerve stimulation with a suction electrode (200 μsec, 5 V), evoked EJPs were recorded. Six to ten EJPs evoked by low frequency of stimulation (0.1 Hz) were averaged. To calculate mEJP mean amplitudes, 50–100 events from each muscle were measured and averaged using the Mini Analysis program (Synaptosoft). Minis with a slow rise and falling time arising from neighboring electrically coupled muscle cells were

excluded from analysis[75,76]. Quantal content was calculated by dividing the mean EJP by the mean mEJP after correction of EJP amplitude for non-linear summation according to previously described methods[77,78]. Corrected EJP amplitude = $E[Ln[E/(E - recorded\ EJP)]]$, where E is the difference between reversal potential and resting potential. The reversal potential used in this correction was 0 mV[78,79]. Statistical analysis was performed with Prism 9 (GraphPad Software) using a One-way ANOVA with Tukey correction for multiple comparisons. Differences were considered significant at $p < 0.05$. Data are presented as mean ± SEM.

### Reporting summary
Further information on research design is available in the Nature Portfolio Reporting Summary linked to this article.

## Data availability
Supplementary Figs 1–17, together with Supplementary Table 1, containing average values for the electrophysiological recordings, and Supplementary Table 2, including the experimental conditions for immunohistochemistry, are available in the Supplementary Information file. Any additional information required to reanalyze the data reported in this paper is available from the corresponding author upon request. Source data are provided with this paper.

## Code availability
The code facilitating the image data analysis in the current study is available in the Supplementary Information file.

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

## Acknowledgements

We are grateful to Janet Richmond for antibodies and fly strains. This work was supported by the Intramural Program of NICHD, project ZIA HD008914, awarded to M.S. F.O. acknowledges funding from the Deutsche Forschungsgemeinschaft (DFG, German Research Foundation) Collaborative Research Center 1286 on Quantitative Synaptologie (CRC/SFB1286) projects Z04.

## Author contributions

R.V., S.D.C., T.H.H., T.H.N., and P.N. performed the experiments. R.V., S.D.C., T.H.H., F.O., and M.S. analyzed the data and wrote the manuscript.

## Funding

## Competing interests

F.O. is an inventor on a pending patent by NanoTag Biotechnologies GmbH that covers the ALFA system and its use (application numbers: WO2020053239A1; US20220048947A1; EP3849996A1; CN113195516A; JP2022500076A). F.O. is a shareholder of NanoTag Biotechnologies GmbH. The remaining authors declare no competing interests.
