## [Peer Review File · Nature Communications]

Reviewers' Comments:

Reviewer #1:

Remarks to the Author:

In this paper, the authors address the function of the C-terminal PDZ domain binding motif in *Drosophila* Neurexin-1 (Nrx-1) using a novel nanobody-based strategy. Inserting the recently developed ALFA-tag in a less conserved region of the cytoplasmic domain of Nrx-1, both in expression constructs and the endogenous nrx-1 locus, the authors first test their modified proteins and show that they are fully functional and recognized by an ALFA-tag-specific nanobody. Deletion of the C-terminal PDZ domain-binding motif revealed that it is important for ER/Golgi exit, Nrx-1 trafficking and synaptic stabilization at neuromuscular junctions (NMJs). Interestingly, functional differences were observed between complete deletion of this motif or just masking it by attaching the relatively large GFP protein to the C-terminus of Nrx-1. In addition, the authors observe that the ALFA system could be used as a split system, as the PDZ domain-binding motif could be delivered in "trans" to reconstitute Nrx-1 function. Furthermore, they find that Nrx-1 is also expressed in tissues outside the nervous system albeit at low levels.

The paper is well structured and the genetic tools are extensively characterized, thereby confirming previous findings. I have therefore only a few suggestions to improve the manuscript.

Major comments:

1) Previous studies (Li et al., *Neuron* 2007 and Chen et al *PlosOne* 2010), have examined the subcellular localization of Nrx-1 at presynaptic active zones. While Li et al. focused on larval NMJs using confocal microscopy, Chen et al. examined embryonic NMJs using immunogold labeling. The specificity of ALFA-tag in combination with STED microscopy offers a unique opportunity to further narrow down the subcellular localization of Nrx-1 at active or peri-active zones (Brp, FasII). It would thus be well worth adding super-resolution data for co-localization of endogenously tagged Nrx-1-AT and Nrx-1DeltaPDZ-AT with these markers.

2) Figure 6a and 6b show representative snapshots of vesicles trafficking along axons. However, only single vesicles are shown and a quantitative analysis summarizing the average behavior is lacking. Although the authors estimate trafficking speed in the result section (lines 559-566), a more accurate quantitative analysis including n-numbers, particle sizes, speeds, types of movement would be more appropriate. Could the authors accumulate more Nb-Scarlet and Nrx-1-AT positive vesicles by squashing the nerve or expressing these proteins in the background of transport mutations like kinesin? While it is a good idea that these are Rab2-positive vesicles, co-expression of Rab2 and Nrx-1 is not shown in the paper. This is particularly important as Rab3 is transported along axons and has been shown to regulate Brp architecture at active zones (Graf et al., *Neuron* 2009).

Minor points:

a) Citations in line 66 (Taniguchi et al., 2007, Zhang et al., 2005) are not found in the reference list.

b) Figures 6C and 6D are difficult to comprehend and should be improved, as peripheral nerves are difficult to spot and unlabeled in Fig. 6D. Although the accompanying movie 3 clearly shows moving vesicles, most of them seem to be cytoplasmic particles but not axonal transport vesicles. This should be stressed in the corresponding legend. Alternatively, transported vesicles could be labeled in the movies or images and movies could be replaced entirely.

c) The results section is quite long and could be shortened. It is also repetitive at places (e.g. lines

321-327) or contains paragraphs that would better fit to the discussion (e.g. lines 401-416).

Reviewer #2:

Remarks to the Author:

Serpe and colleagues report the first endogenously tagged Nr_x-1 mutant in *Drosophila*, using the ALFA epitope along with its high-affinity nanobody to localize the protein in vivo. They introduced the tag intramolecularly in the c-terminal sequences of the molecule by gene editing to avoid interfering with known functions, for example, of the PDZ-binding motif at the c-terminal end. Comparison with available antibodies to endogenous Nr_x-1 revealed superior sensitivity of the ALFA tag/nb. They then additionally deleted the PDZ recognition motif in the ALFA-tagged flies to study the known mislocalization of the protein (Nr_x-1⊗PDZ) and functional impairments in larval stages by morphological methods and electrophysiological recordings of NMJ parameters. Exploiting the high-affinity binding to the tag, they genetically expressed the ALFA nanobody fused to the missing PDZ-recognition motif in these mutants to demonstrate that it can reconstitute the function in the PDZ-lacking flies. They also extensively compared their endogenously tagged flies to various other overexpression and null mutants of *Drosophila* Nr_x-1 that were characterized previously, including a C-terminally GFP tagged Nr_x1 OE and a full-length untagged Nr_x-1 OE constructs, as well as a newly generated Nr_x-1 ALFA OE, construct that rescued the null-mutant phenotype. Other findings include trafficking data of Nr_x-1 along neurites or expression in enteroendocrine cells, mostly confirming results from previous studies in *Drosophila* or vertebrate systems. They conclude that the ALFA tag/ nanobody technology can be successfully applied in *Drosophila* even for low-abundance molecules such as Nr_x-1.

In my view, there are two general problems with this study that need to be resolved before publication:

(1) As it stands, the manuscript is confusingly structured and written, and, at least in the opinion of this reviewer, is completely underselling its most novel aspect which is the first demonstration of endogenous Nr_x in *Drosophila*! For example, the first two figures of the manuscript start defensively with a little impressive structural prediction on the tentative tag position and generation of an OE construct with the ALFA epitope that simply confirms or repeats what other OE approaches (tagged or wild-type Nr_x-1) have shown before in *Drosophila* and mice. Only in figure 3, the novel aspect of an endogenously-tagged Nr_x-1 using the ALFA system is introduced, and its high-affinity binding to the nanobody is exploited to target Nb-fusion protein with the PDZ motif to Nr_x-1⊗PDZ. Afterward, the study returns to overexpression constructs to confirm existing data on expression patterns and Nr_x trafficking along axons.

(2) Despite the successful application of the very sensitive ALFA tag/Nb system to an endogenous, low-abundant molecule such as Nr_x-1 in *Drosophila*, the actual results reported here are largely confirmatory and mostly repeat results from earlier studies. For example, various epitope-tagged Nr_x constructs were used in flies and mice to rescue the null-mutant alleles, and the significance of the PDZ recognition/binding motif at the C terminus of Nr_xn has been characterized extensively in vertebrate and invertebrate neurons before. Similarly, the data on the endogenous expression of Nr_x-1 also do not contain substantial new information on the whereabouts of Nr_x. For this, the authors used a Minos/TrojanGal4 line available from Bloomington. The insertion truncates the Nr_x-1 mRNA and renders flies homozygous lethal. Using heterozygous flies, they find pan-neuronal as well as body wall muscle expression - essentially the same pattern as shown in embryos and larvae (Chen et al. 2010, Sun et al. 2015). Also, expression was found in a subpopulation of gut cells but they fail to acknowledge that this was reported before (Sun et al. 2015). In contrast, it would be very interesting to see the ALFA system's usability for high-resolution imaging of NMJs using the endogenous Nr_x-1-AT.

Additional major concerns:

(3) The trafficking data are preliminary at best. There is no information on how the speed and size of “vesicles” are quantified, neither on temporal resolution nor the time of a single scan. From the data presented in Figures 6 a and a’, it is simply not possible to follow the conclusions. First, mScarlet accumulations are hypothesized to be physiologically relevant vesicles but this is not shown. A co-staining for known vesicular components would be necessary to make this point. In videos 3 and 4, accumulations of mScarlet appear to be very large and move alternately backward and forwards rather than unidirectionally, raising doubts about the nature of these “vesicles”. Even if a vesicular structure was confirmed by decisive labeling, a larger number of vesicles (no information on the number of vesicles quantified is given in the current manuscript) need to be quantified to make sense of such observations. Second, anterograde or retrograde transport cannot be distinguished here, because *Nrx-1-Gal4* directed expression of *UAS-Nrx-AT* is pan-neuronal, including sensory neuron axons which project through the abdominal nerves toward the VNC. Thus, the retrograde transport of large “vesicles” could correspond to anterograde transport in sensory neuron axons. Third, the data collected with *BG380-Gal4* in Figure 6c are not quantified. To quantify vesicle movement and obtain identity in consecutive frames the temporal resolution should be such that spatial overlap across frames exists, a standard in such trafficking studies, for example, of *Nrx* in murine neurons (Neupert et al., 2015, Klatt et al., 2021). Fourth, the authors discuss the possibility that transport could be Rab2 mediated but provide no evidence. The idea could easily be tested because several stock collections are available that express tagged Rab proteins and are suitable for live imaging, for example, those donated to BDSC by Hugo Bellen. Similarly, the authors claim that *Nrx-1*⊗PDZ-AT is retained in the ER or early endosomal compartments. Again, there is no data to support this notion, though it is straightforward to label either the ER or early endosomal compartments alongside Nanobody expression in the endogenous *Nrx-1-AT* background.

(4) An important control is missing. The authors tested whether functional NMJ parameters are rescued in the *Nrx*ΔPDZ-AT *BG380-Gal4*, *UAS-NB-PDZ* background. While this was the case, leading to the conclusion that no negative (steric) effects are inflicted by NB-PDZ binding, the same control experiments are not shown for the *Nrx-Gal4*>*UAS-Nrx-AT*; >*UAS-NB-mScarlet* background. Since the nanobody chimera is even bigger and the fluorophore might agglomerate, important protein-protein interactions at the *Nrx-1* C-terminus could be masked or might be the reason for the accumulation of mScarlet signal in live imaging experiments.

(5) There are additional unsubstantiated conclusions. Which data in the manuscript support the claims that “the PDZ binding motif mediates *Nrx-1* anchoring at presynaptic sites” and “ensures proper active zone assembly and function”? There is no experimental finding to support either of these claims.

Minor comments:

- Both the ALFA overexpression construct as well as the endogenously tagged *Nrx-1* locus are referred to as “*Nrx-1-AT*” throughout the manuscript which is confusing and may mislead readers about the nature of the experiments. Alleles of genes in *Drosophila* are usually indicated with additional information in superscript which could easily be done, for example, using *Nrx-1^{AT}* or *Nrx*ΔPDZ::*AT* for the endogenously tagged *Nrx-1*. Similarly, overexpression constructs could be identified with *UAS-XY*.
- Line 34: “assembles” does not seem to be the correct wording here.
- Line 121: “two (or even three)” reads oddly.
- In almost all figure panels, the driver line identity is omitted (except for Figures 6 c and d) but should be included for clarity.
- Figure S2: There is only one scale bar, but the legend reads “scale bars”
- Line 292: a word seems to be missing.
- Figure 2: Amplitude mv or mV?

- Evoked excitatory junction potentials are abbreviated as EJPs mostly, but as eEJP in figure 3.
- Figure 3: mv or mV?
- Figure 3: It isn't easy to match traces and their respective quantifications in this figure.
- Lines 245 & 250: "docked" sounds strange in this context.
- Line 315 to 317: It is difficult to follow the authors here. In addition, there is no quantification that allows deriving that the signal is linear. It is not clear what polyclonal secondaries are.
- Line 330: It would be interesting to know when the majority of flies die and what the fraction of escapers is. Are the following experiments done with escapers?
- Line 445: How is 100% penetrance defined?
- Line 580: "Supplemental S11" should likely be "S12".
- If Gaussian blurred images are subtracted for background reduction, it should be indicated when this method is used.
- No information about sequence alignment and secondary structure prediction methods is available but should be included.
- Line 622: What exactly do the authors mean by "compartment-specific expression"?
- Line 758: UAS-Nb-mCherry is apparently wrong.

Reviewer #3:

Remarks to the Author:

Key results: In this manuscript, Choudhury et al. investigate the subcellular distribution, C-terminal domain properties, and axonal trafficking of the *Drosophila* synaptic cell adhesion molecule neurexin-1 (Nrx-1). These qualities of mammalian neurexin-1 have been previously challenging to study due to its low abundance and its numerous splice variants that impede facile detection by conventional immunolabeling techniques. Moreover, tagging fly Nrx-1 with GFP at the C-terminus leads to aberrant subcellular localization and function. Choudhury et al. overcome these challenges via introduction of a small epitope tag (ALFA tag or AT) into an apparently tolerant region of the intracellular C-terminus of Nrx-1, enabling its detection via anti-AT nanobodies, either applied to fixed samples as immunolabels or expressed as genetically encoded intrabodies. They also use the AT/nanobody system to rescue a mutant form of Nrx-1-AT lacking a critical C-terminal PDZ binding motif (Nrx-1-AT- Δ PDZ) required for its presynaptic localization. By supplying the PDZ binding motif in trans via the anti-AT nanobody, they rescue Nrx-1-AT-PDZ mutant localization and presynaptic function.

Validity: Most conclusions and claims are supported, and the data and methodology presented in this paper appears to be valid overall. The methods include sufficient details to allow for the work to be reproduced. The data analysis is appropriate although quantitation of the immunolabeling results would enhance the authors statements as to the "remarkable" nature of labeling with the ALFATag nanobody system. The interpretation of some of the results are somewhat narrow and should be expanded to include other possibilities. There are numerous concerns as detailed below. Addressing these should markedly improve the manuscript.

Significance: The main strengths of this study are that 1) the authors demonstrate a method that enables more sensitive detection of a Nrx-1, allowing for a better understanding of the subcellular distribution of Nrx-1 and opening the door for other studies (e.g., immuno-EM analysis of synaptic Nrxn1-AT); and 2) the authors demonstrate that the C-terminal PDZ domain of Nrx-1 is necessary for its normal presynaptic distribution, and that defective Nrx-1-AT- Δ PDZ can be rescued via PDZ-domain-containing AT-nanobody binding to this mutant. Broadly speaking, this study adds to the demonstrated utility of using nanobodies as immunolabels and as intrabodies. Overall, these findings seem appropriate for *Nature Communications*, however substantial revisions and major corrections to the present manuscript would be required.

Clarity and context: The clarity of the manuscript could be improved. The writing is in many places a bit rough, with missing words and incorrect subject-verb agreement (singular/plural, tense) and

missing definite and indefinite articles, some of which are detailed below. The manuscript is also written in a manner that would be a bit challenging for those lacking specific expertise in the *Drosophila* NMJ model system, as to the authors not concisely presenting the rationale for certain experiments. While the authors appeared to provide relevant background information and references, understanding its significance to the present manuscript for the average reader may be helped by making the text more concise (e.g., limiting redundancy) and making more explicit statements about why an experiment/system was selected [for example, see comments below regarding the Nr_x-1-GAL4 CD4 experiments].

Specific comments and suggested improvements (Page/line)

Major:

6/173: The authors state they generated transgenes "to express equal levels of tagged and untagged Nr_x-1". Is this demonstrated at the protein level? For example does the tag alter Nr_x-1 protein stability/abundance compared to WT Nr_x-1? If not, the authors should state that this was their intention of these strains but that whether the outcome was achieved has not been confirmed at the protein level.

6/181: The authors state that strong neuronal expression of untagged Nr_x-1 "did not significantly enhance the Nr_x-1 signals in the neuropil". Is this quantified? How are the relative expression levels deduced? Were statistical analyses performed to justify the use of the word "significantly".

7/187: Here and in the other strains used in this section, simple quantification of the immunofluorescence signals in multiple samples is warranted.

Figure S2 should be incorporated into Figure 1. Most of the substantive evidence supporting the statements in the first results section (lines 185-199) seem to come from this figure. Figure 1b could be moved to supplementary data to make space for the info in S Figure 2.

10/300-302: The authors do not perform any labeling to determine whether the recruitment of other common presynaptic molecules is affected(E.g., synaptophysin), such that their interpretation of structure is primarily based on the signal from the protein (Nr_x-1) whose expression and properties they are manipulating.

12/362. Results section: The authors should provide the details of the structure of NbALFA-PDZ binding motif chimera as supplemental material, as per the other constructs. It is difficult to piece together what this comprises from the methods section. This information is much more critical than for example the information detailed in Supplemental Figure 5 which is quite obvious.

14/418: This section appears largely superfluous to the rest of the study. Moreover, it seems based on the premise that the subcellular localization of exogenous CD4 should mimic that of Nr_x-1. The proper experiment would be to examine Nr_x-1-AT labeling in these regions

14/435-445: The authors should make clear that they are not determining "Nr_x-1 expression" but a proxy reporter for this.

16/473: The authors state GFP cleaved off the chimera, and on lines 489-490 "Therefore, the spreading of the GFP signals suggests a physical separation of the GFP tag". This is speculation. The premise that GFP "falls off" NRX-1 needs to be addressed experimentally, for example by immunoblot. It seems just as or more likely that these signals are from separate pools of intact Nr_x-1-GFP proteins in distinct states and different properties including subcellular localization and immunoreactivity to anti-Nr_x-1 antibody used (which the authors already argue is problematic).

16/494-496: Masking is assumed. (also see line 243). Masking is a very specific molecular mechanism. I suggest using the more general model that GFP somehow "interferes" with the

phenotype. There are numerous examples of PDZ binding motifs that do not need to be C-terminally exposed to function, and the GFP could be exerting allosteric effects on the overall structure of the cytoplasmic domain. Would be better to emphasize the strong results that the PDZ domain is needed for presynaptic targeting of Nr1.

18/529: This claim should be demonstrated by knocking out the WT Nr1 expression in the context of Nr1-AT and Nb expression.

18/530: Are these biologically relevant structures or artifacts of the expression system? Do you see them with immunolabeling (either conventional ab or nb)?

20/593: Overall discussion is a bit long and should be made more concise.

20/598: The authors draw an equivalence between the inclusion of a C-term GFP with a PDZ masking mechanism. This is too much of an assumption.

23/677: The authors should discuss what their results tell us about the location of the PDZ binding motif within the structure of the Nr1 cytoplasmic domain. Their results suggest the PDZ binding motif can be "moved" quite a bit upstream (via Nb-PDZ binding to AT sequence) of the C-terminus and still function. Does this suggest that the PDZ binding motif could be knocked-in to the AT locus in Nr1-GFP to rescue the "masking" defect?

Minor:

2/53: citation format is incorrect

3/69: neurexins' (plural possessive?)

3/79: missing words – "not only"

3/89: abundance instead of abundant

4/100: intrabody capability

Page 4-5: These two paragraphs are more appropriate for and somewhat repetitive of the introduction. Perhaps can shorten this and move non redundant parts to intro

6/159-161: Insertion of AT within the extracellular domain? Clarify

6/165: have variable expansions

6/181: traffics

8/242: suggests rather than indicates

9/251: The authors suggest that the differences are due to expression levels, Can this be demonstrated/quantified? See earlier comments about lack of quantification.

Lines 253-256: "The increased mEJP frequency and reduced quantal content indicate primarily presynaptic deficits, probably due to C-terminal GFP obstructing Nr1 binding to interactors such as Scribble, Syd-1 or Spinophilin (Muhammad et al., 2015; Oswald et al., 2012; Rui et al., 2017)." This should be moved to the discussion section and alternative models should be provided.

10/288: Are N-terminal Nr1-11 antibodies available and if so would the authors expect that these would yield similar results?

11/311: Or existence of distinct Nr1 populations with different AT epitope accessibility? Please address

14/406: unsure what is meant by plastic

14/4400-416: this paragraph is more appropriate for discussion

14/420: "unprecedented ability": overstatement - wasn't the detection of Nr1-1 AT by NbALFA the underlying thesis of this work?

16/482-483 "we found that the AT but not GFP signals perfectly co-localized with the Nr1-1-positive immunoreactivities". The use of perfectly here is not scientifically valid.

17/507: "fully functional" is presumed, should be tempered

19/563: Could the differences in velocity measurements be due to the elevated temperature of mammalian cells (37 degrees C) versus flies?

19/570: Can you fix the imaged cell and immunolabel for Rab2 to confirm?

20/610: This claim should be tempered. Numerous factors can affect background signal in different preparations

21/622-623: not sure what is meant by this. do you just mean Nr1-1 AT KI samples? compartment experiment would be better with different Nb-fluorescent protein fusion

24/720: the beta-cell analogy is too specific and detailed given the scant evidence presented. The authors should temper language and generalize in a much more concise way to neuroendocrine cells, which use slightly different exocytotic machinery for DCV release and don't form synapses for fast NT release.

Reviewer #4:

Remarks to the Author:

The brief summary of my review is that I found this to be a very impressive research paper and recommend publishing it with the minor revisions listed below as well as my rationale for reaching this decision based on the given guidelines.

-Thomas Ravenscroft

1. What are the noteworthy results?

In this manuscript the authors use Nr1 as an example protein whiches function, localization and trafficking can be monitored using nanobodies.

Specifically they:

1. Generate a new, endogenously tagged Nr1-ALFA allele that can provide high-resolution images with the ALFA-nanobody with minimal background staining that can study the localization of the protein both in synapses and in axons.
2. They show that the absence/covering of the PDZ domain in Nr1 causes functional deficits in the NMJ and that this can be rescued with some beautiful work reconstituting the PDZ domain in a trans location, bound to an ALFA-nanobody.
3. Absence of the PDZ domain causes Nr1 mislocalization, likely leading to electrophysiological phenotypes.

4. Live trafficking of NrX1a occurs in both retero and anterograde in axons.

2. Will the work be of significance to the field and related fields? How does it compare to the established literature? If the work is not original, please provide relevant references.

This work is significant in the fields of developmental biology and neuroscience as well as provides a great example of using nanobodies to study structure-function relationships. This has great potential to inspire many others to use this nanobody approach across many areas of scientific research

3. Does the work support the conclusions and claims, or is additional evidence needed?

By and large, the manuscript results do support the claims. However, one area that is missing is quantifying the levels of ALFA-tagged protein compared to non-tagged. Many times (indicated below) the claims "equal levels of expression is used" without seeing a western or even qPCR of expression levels. This is of interest as we see NrX1a-AT at the NMJ and not the native antibody. Plus all the trafficking data in the last 2 figures carry more weight with this quantified. Additionally, the specifications for how the gaussian blur filter was used for Figure 5 and Supplemental Figure 10 are also not clear to me. I can see mScarlett spots in the non-ALFA condition, they are smaller and less intense however so the comment should be more on size rather than a presence vs absence statement. I think a like-for-like comparison for axonal transport is needed to show the expression-dependent effects i.e show close-up axon tracking for 6c and d like 6a and b are.

4. Are there any flaws in the data analysis, interpretation and conclusions? - Do these prohibit publication or require revision?

In the above answer (3) I outline the issues I have with analysis and interpretation. I think these require just further explanation, perhaps just to myself and not in the manuscript, to address those concerns. Additionally, the comments on expression/levels can be addressed with quantified data or the text in the paper rephrased. These are minor revisions that should not take long to address.

5. Is the methodology sound? Does the work meet the expected standards in your field?
Yes. It goes beyond those expectations.

6. Is there enough detail provided in the methods for the work to be reproduced?
Not quite, but with very straightforward clarification on some imaging procedures it will be.

Minor/Major comments below

- Minor - Fig 1a - Improve the key by having a separate key above the images
- Minor - Fig 1a - Annotate the glycosylated stalk on the diagram
- Minor - 135 - Active zones, limit NRX-1 activities - Not clear what this means
- Major - 174 - Claim that equal levels of transgene are expressed, RNA-seq/ a western blot will be needed to confirm this
- Minor- My opinion is that Supp figure 2 is more informative than Figures 1c-g. I recommend including this in the main figure along with the labelling in the control
- Major - Fig 2 - Overexpression increases bouton number by 10%, this is problematic when making comparisons, also, GFP tagged NrX-1 does the same. The increase in bouton number in the OE conditions indicates that the release per bouton is lower. I think this should be commented on.
- Minor - 235 - And a non-tagged transgene
- Major - qPCR/westerns are needed to compare levels with all the comments on comparable

expression

- Major - 277 - Indicate levels are the same, need some quantification of this (I know it is frustrating to ask this but with all the comparisons of levels etc. I think a western is needed to show differences in levels. Li et al show the Ab works well on adult head)
 - Minor - Figure 4 - The MIMIC diagram should label all components, both attR sites and both Minos repeats
 - Minor- Figure 4,S7,S8 - Are these late or early 3rd instar larvae as the expression seems to change depending on these stages in muscle.
 - Minor - We need to see channel specific images for the sup figure 7 and 8, hard to see details in the composites
 - Minor - For figure 4 please show the non-composite image for NrX-1 labelling
 - Minor- Supp figure 10 is more informative than figure 5a-h as you can see close-up details, I suggest swapping these or inserting S10 in the main figure
 - Major- I am not sure about the gaussian blur data showing the transport vesicles containing nrX1a. mScarlet is clearly transported along axons as it is present in the NMJ in s10 so we would expect to see it in axons as well. Furthermore the data in 5j' does show mScarlet signal in the axons. In addition it looks like the GFP positive spots in the axons of 5i are smaller and less numerous than in 5j. In the supplement we do see mscarlet spots in the non-AT condition but they appear drowned out by the higher signal in the cell body.
 - Further- how was that size (3) determined? I Assume it refers to pixel size? Looks like they are only selecting for larger vesicles, in which case they see larger vesicles with NrX1a-ALFA.
 - Minor - For figure 6 they show axon transport close up vs. low magnification view of the larval VNC. To show distinctly that there is no transport of the Nb-mScarlet in lieu of NrX-1A the same comparison needs to be made with the close up of axon transport with the gal80 system
 - Minor 644 - likely exit from the ER, without staining ER can't be sure, may be in Golgi so this needs to be rephrased
 - Minor 750 - How many UAS' were used in the constructs one or 20? This needs to be stated. Same for Nb-mCherry and Nb-PDZ
 - Minor 815 - Which samples used what fix?
 - Minor 825 - Little or no detergent? This is not helpful, what was used in this study?
 - Minor 816 - PBS or PBS Triton? Which samples had which and for how long? I know this is nitpicking but it needs to be repeatable
 - Minor 820 - How long were samples incubated in primary/secondary? I recommend adding a table for each figure and list each antibody used and add to supplement similar to table S1
 - Minor - 835 - "Imaris and ImageJ respectively" Respective to what?
 - Minor 838 - Should cite the filament algorithm reference (Flormann 2021 faseb)
 - Minor - 863/866 - Zoom 3x?
- Minor - 863 - Please report the frame rate

Response to Reviewers:

I would like to express my deep appreciation to the reviewers for their careful consideration of this manuscript and for their very constructive and comprehensive comments. Their suggestions have been incorporated in the revised manuscript, which was greatly improved by addressing all the issues raised.

Below are responses to specific comments and suggestions from reviewers.

Reviewer #1 (Remarks to the Author):

In this paper, the authors address the function of the C-terminal PDZ domain binding motif in *Drosophila* Neurexin-1 (Nrx-1) using a novel nanobody-based strategy. Inserting the recently developed ALFA-tag in a less conserved region of the cytoplasmic domain of Nrx-1, both in expression constructs and the endogenous nrx-1 locus, the authors first test their modified proteins and show that they are fully functional and recognized by an ALFA-tag-specific nanobody. Deletion of the C-terminal PDZ domain-binding motif revealed that it is important for ER/Golgi exit, Nrx-1 trafficking and synaptic stabilization at neuromuscular junctions (NMJs). Interestingly, functional differences were observed between complete deletion of this motif or just masking it by attaching the relatively large GFP protein to the C-terminus of Nrx-1. In addition, the authors observe that the ALFA system could be used as a split system, as the PDZ domain-binding motif could be delivered in "trans" to reconstitute Nrx-1 function. Furthermore, they find that Nrx-1 is also expressed in tissues outside the nervous system albeit at low levels.

The paper is well structured and the genetic tools are extensively characterized, thereby confirming previous findings. I have therefore only a few suggestions to improve the manuscript.

Major comments:

1) Previous studies (Li et al., *Neuron* 2007 and Chen et al *PlosOne* 2010), have examined the subcellular localization of Nrx-1 at presynaptic active zones. While Li et al. focused on larval NMJs using confocal microscopy, Chen et al. examined embryonic NMJs using immunogold labeling. The specificity of ALFA-tag in combination with STED microscopy offers a unique opportunity to further narrow down the subcellular localization of Nrx-1 at active or peri-active zones (Brp, FasII). It would thus be well worth adding super-resolution data for co-localization of endogenously tagged Nrx-1-AT and Nrx-1DeltaPDZ-AT with these markers.

We too had hoped that the ALFA-tag methodology would facilitate super-resolution imaging. But the intensity of "one-step detection" signals remains a serious limitation for low abundant protein such as Nrx-1. Such signals are proportional to the amount of protein of interest and are simply not bright enough for STED or other super-resolution methodologies.

We have also tried to (i) acquire high resolution details of the Nrx-1 synaptic localization relative to pre- and postsynaptic landmarks using the lightning process on a Leica Stellaris 8 platform, or to (ii) amplify the AT signals via conventional methods. But the low levels of endogenous Nrx-1 and the low signal/ noise ratio continued to remain a problem for super-resolution imaging.

2) Figure 6a and 6b show representative snapshots of vesicles trafficking along axons. However, only single vesicles are shown and a quantitative analysis summarizing the average

behavior is lacking. Although the authors estimate trafficking speed in the result section (lines 559-566), a more accurate quantitative analysis including n-numbers, particle sizes, speeds, types of movement would be more appropriate. Could the authors accumulate more Nb-Scarlet and Nr_x-1-AT positive vesicles by squashing the nerve or expressing these proteins in the background of transport mutations like kinesin? While it is a good idea that these are Rab2-positive vesicles, co-expression of Rab2 and Nr_x-1 is not shown in the paper. This is particularly important as Rab3 is transported along axons and has been shown to regulate Brp architecture at active zones (Graf et al., Neuron 2009).

As this reviewer and others recommended, we have tremendously expanded our live imaging analyses and accomplished a comprehensive analysis of the vesicles trafficking along the axons. The revised manuscript includes a thorough quantitative analysis of the n-numbers, particle speeds, and types of movement (please see revised Fig. 6 and associated text). The large pinhole used here for imaging precluded a precise estimation of the puncta size.

The revised manuscript also includes a new set of experiments demonstrating that 75% of the Nr_x-1-containing vesicles are Rab2-positive (Fig. 6 and the associated text).

Minor points:

a) Citations in line 66 (Taniguchi et al., 2007, Zhang et al., 2005) are not found in the reference list.

We thank the reviewer for catching this omission, which we corrected in the revised manuscript.

b) Figures 6C and 6D are difficult to comprehend and should be improved, as peripheral nerves are difficult to spot and unlabeled in Fig. 6D. Although the accompanying movie 3 clearly shows moving vesicles, most of them seem to be cytoplasmic particles but not axonal transport vesicles. This should be stressed in the corresponding legend. Alternatively, transported vesicles could be labeled in the movies or images and movies could be replaced entirely.

As the reviewer recommended, we have replaced these movies with clearer figures that focused on the axonal bundles exiting the neuropil. We have also specified in the corresponding legend that these particles reside close to the neuron soma. These data have been moved into the Supplementary S15 to make room in the main Fig. 6 for more relevant Rab2 co-localization studies. A completely new set of movies (1-6) has also been included in this revision.

c) The results section is quite long and could be shortened. It is also repetitive at places (e.g. lines 321-327) or contains paragraphs that would better fit to the discussion (e.g. lines 401-416).

As recommended by the reviewers, we have revised and shortened the results section, including lines 321-327.

Reviewer #2 (Remarks to the Author):

Serpe and colleagues report the first endogenously tagged Nr_x-1 mutant in *Drosophila*, using the ALFA epitope along with its high-affinity nanobody to localize the protein in vivo. They introduced the tag intramolecularly in the c-terminal sequences of the molecule by gene editing to avoid interfering with known functions, for example, of the PDZ-binding motif at the c-terminal end. Comparison with available antibodies to endogenous Nr_x-1 revealed superior sensitivity of the ALFA tag/nb. They then additionally deleted the PDZ recognition motif in the ALFA-tagged flies to study the known mislocalization of the protein (Nr_x-1^ΔPDZ) and functional impairments

in larval stages by morphological methods and electrophysiological recordings of NMJ parameters. Exploiting the high-affinity binding to the tag, they genetically expressed the ALFA nanobody fused to the missing PDZ-recognition motif in these mutants to demonstrate that it can reconstitute the function in the PDZ-lacking flies. They also extensively compared their endogenously tagged flies to various other overexpression and null mutants of *Drosophila* Nr_x-1 that were characterized previously, including a C-terminally GFP tagged Nr_x1 OE and a full-length untagged Nr_x-1 OE constructs, as well as a newly generated Nr_x-1 ALFA OE, construct that rescued the null-mutant phenotype. Other findings include trafficking data of Nr_x-1 along neurites or expression in enteroendocrine cells, mostly confirming results from previous studies in *Drosophila* or vertebrate systems. They conclude that the ALFA tag/ nanobody technology can be successfully applied in *Drosophila* even for low-abundance molecules such as Nr_x-1.

In my view, there are two general problems with this study that need to be resolved before publication:

(1) As it stands, the manuscript is confusingly structured and written, and, at least in the opinion of this reviewer, is completely underselling its most novel aspect which is the first demonstration of endogenous Nr_x in *Drosophila*! For example, the first two figures of the manuscript start defensively with a little impressive structural prediction on the tentative tag position and generation of an OE construct with the ALFA epitope that simply confirms or repeats what other OE approaches (tagged or wild-type Nr_x-1) have shown before in *Drosophila* and mice. Only in figure 3, the novel aspect of an endogenously-tagged Nr_x-1 using the ALFA system is introduced, and its high-affinity binding to the nanobody is exploited to target Nb-fusion protein with the PDZ rec motif to Nr_x-1⊗PDZ. Afterward, the study returns to overexpression constructs to confirm existing data on expression patterns and Nr_x trafficking along axons.

Our study showcases and challenges the versatility of a new tool, the ALFA system, towards detection and structure-function analyses of low abundant proteins localized within complex cellular structures. As such, a major concern is the proper functionality of the tagged protein. The extensive description of the rationale for selecting the ALFA-tag insertion site and the thorough assessment of ALFA-tagged transgene's activity address this major hurdle in the field.

Aside from the rigorous characterization of the tagged Nr_x-1 and its activities at both low/ endogenous and high/overexpression levels, our study highlights a myriad of applications that are now possible with a functional ALFA-tagged protein. They include structure-function analyses (Fig. 3), compartment specific detection (Figs. 4- 5) and live trafficking studies (Fig. 6). We have also improved this revised manuscript as suggested by the reviewers by expanding our trafficking analyses (revised Fig. 6 and associated supplementary figures).

An entire subsection has been reorganized under the heading "*Interfering with the PDZ binding motif alters Nr_x-1 synaptic function*" to improve the structure of the manuscript.

(2) Despite the successful application of the very sensitive ALFA tag/Nb system to an endogenous, low-abundant molecule such as Nr_x-1 in *Drosophila*, the actual results reported here are largely confirmatory and mostly repeat results from earlier studies. For example, various epitope-tagged Nr_x constructs were used in flies and mice to rescue the null-mutant alleles, and the significance of the PDZ recognition/binding motif at the C terminus of Nr_xn has been characterized extensively in vertebrate and invertebrate neurons before. Similarly, the data on the endogenous expression of Nr_x-1 also do not contain substantial new information on the whereabouts of Nr_x. For this, the authors used a Minos/TrojanGal4 line available from Bloomington. The insertion truncates the Nr_x-1 mRNA and renders flies homozygous lethal. Using heterozygous flies, they find pan-neuronal as well as body wall muscle expression - essentially the same pattern as shown in embryos and larvae (Chen et al. 2010, Sun et al.

2015). Also, expression was found in a subpopulation of gut cells but they fail to acknowledge that this was reported before (Sun et al. 2015). In contrast, it would be very interesting to see the ALFA system's usability for high-resolution imaging of NMJs using the endogenous Nr_x-1-AT.

We thank the reviewer for pointing out the missing reference, which we now incorporated in the revised manuscript.

The choice for Nr_x-1 as a case study for the versatility of this new tool was based on the availability of a substantial amount of knowledge for both vertebrate and invertebrate neuroligands. This allows for a comprehensive comparison between the *in vivo* utility of the new ALFA system versus the previous methodologies. Our goals were to first demonstrate the superiority of the ALFA system towards generating a functional tagged protein, then apply it to previously inaccessible *in vivo* questions, particularly to structure-function studies. To this end, we demonstrated that using a single ALFA tag, inserted at a carefully selected position, one could practically tackle every aspect of the cell biology and function of a complex protein.

Despite our best efforts, including using the advanced Leica Stellaris 8 platform (please also see response to reviewer #1), the low levels of endogenous Nr_x-1 and low signal/ noise ratio continued to hinder super-resolution imaging of the endogenous Nr_x-1^{AT} at the NMJ.

Additional major concerns:

(3) The trafficking data are preliminary at best. There is no information on how the speed and size of "vesicles" are quantified, neither on temporal resolution nor the time of a single scan. From the data presented in Figures 6 a and a', it is simply not possible to follow the conclusions.

- First, mScarlet accumulations are hypothesized to be physiologically relevant vesicles but this is not shown. A co-staining for known vesicular components would be necessary to make this point. In videos 3 and 4, accumulations of mScarlet appear to be very large and move alternately backward and forwards rather than unidirectionally, raising doubts about the nature of these "vesicles". Even if a vesicular structure was confirmed by decisive labeling, a larger number of vesicles (no information on the number of vesicles quantified is given in the current manuscript) need to be quantified to make sense of such observations.

As recommended by the reviewer, we have confirmed the vesicular nature of the mScarlet puncta. We accomplished this by exploiting an old observation by Lilly Jan, that anti-HRP stains all neuronal membranes in *Drosophila* (and other insects). We performed exhaustive analyses which demonstrated that the mScarlet-labeled puncta coincide with HRP-positive puncta. In these analyses we examined 1355 and 1343 puncta per genotype respectively. These new data are included in the revised Fig. 6 and Supplementary S16.

We have also analyzed 62 individual vesicles moving along the axons and included a thorough quantitative analysis of the particle numbers, speeds, and types of movement in the revised manuscript (please see revised Fig. 6 and associated text).

Second, anterograde or retrograde transport cannot be distinguished here, because Nr_x-1-Gal4 directed expression of UAS-Nr_x-AT is pan-neuronal, including sensory neuron axons which project through the abdominal nerves toward the VNC. Thus, the retrograde transport of large "vesicles" could correspond to anterograde transport in sensory neuron axons.

The reviewer is correct: *Nrx-1-Gal4* directs expression in both motor neurons and sensory neurons. In the revised manuscript we have corrected this part. We have also expanded the

number of vesicles imaged and captured fast and slow vesicles moving in both directions along the axon bundle. These data are statistically analyzed in the revised Fig. 6.

Third, the data collected with BG380-Gal4 in Figure 6c are not quantified. To quantify vesicle movement and obtain identity in consecutive frames the temporal resolution should be such that spatial overlap across frames exists, a standard in such trafficking studies, for example, of Nr1 in murine neurons (Neupert et al., 2015, Klatt et al., 2021).

As this reviewer and others recommended, we have replaced these figures with clearer ones that focused on the axonal bundles exiting the neuropil. These new data are presented in the revised Fig. 6, Supplementary S15 and the new set of Supplementary Movies.

Fourth, the authors discuss the possibility that transport could be Rab2 mediated but provide no evidence. The idea could easily be tested because several stock collections are available that express tagged Rab proteins and are suitable for live imaging, for example, those donated to BDSC by Hugo Bellen.

As the reviewer recommended, we have acquired the available Rab2^{CA}-YFP transgene (donated to BDSC by Hugo Bellen) and expanded our studies to test whether the Nr1-containing vesicles are Rab2-positive. We were unable to perform live imaging with three different UAS transgenes (*UAS-Nb-mScarlet*, *UAS-Rab2^{CA}-YFP* and *UAS-Nr1-AT*) or detect a consistent number of mScarlet puncta along the axon, probably due to Gal4 pool dividing among three different regulatory regions. Nonetheless, we conducted successful immunohistochemistry experiments on motor neuron soma, examined the nature of the Nr1/mScarlet-positive intracellular puncta, and found that 75% of the Nr1-containing vesicles are Rab2-positive.

These new data are included in the revised Fig. 6 and Supplementary S16.

Similarly, the authors claim that Nr1- \otimes PDZ-AT is retained in the ER or early endosomal compartments. Again, there is no data to support this notion, though it is straightforward to label either the ER or early endosomal compartments alongside Nanobody expression in the endogenous Nr1-AT background.

As requested, we have repeated these experiments using anti-Calnexin99A to label the ER compartment. The new data, presented in Supplementary S6, demonstrate that Nr1-dPDZ is indeed retained in the ER or early Golgi.

(4) An important control is missing. The authors tested whether functional NMJ parameters are rescued in the Nr1-dPDZ-AT GB380-Gal4, UAS-NB-PDZ background. While this was the case, leading to the conclusion that no negative (steric) effects are inflicted by NB-PDZ binding, the same control experiments are not shown for the Nr1-Gal4>USA-Nr1-AT; >UAS-NB-mScarlet background. Since the nanobody chimera is even bigger and the fluorophore might agglomerate, important protein-protein interactions at the Nr1 C-terminus could be masked or might be the reason for the accumulation of mScarlet signal in live imaging experiments.

The reviewer's concern is valid: the large Nb-mScarlet chimera may be prone to aggregation and therefore obstruct important protein interactions.

As recommended, we tested these possibilities using electrophysiology recordings from the following genotypes:

- *Nr1-Gal4>UAS-Nb-mScarlet*
- *Nr1-Gal4>USA-Nr1-AT, UAS-Nb-mScarlet*
- *Nr1-Gal4>USA-Nr1-AT*

These new results, included in the Supplementary S12, demonstrate that the addition of Nb-mScarlet chimera does not disrupt Nr_x-1 activities, nor impairs NMJ functionality.

(5) There are additional unsubstantiated conclusions. Which data in the manuscript support the claims that “the PDZ binding motif mediates Nr_x-1 anchoring at presynaptic sites” and “ensures proper active zone assembly and function”? There is no experimental finding to support either of these claims.

We have shortened and revised this paragraph and removed this sentence.

Minor comments:

- Both the ALFA overexpression construct as well as the endogenously tagged Nr_x-1 locus are referred to as “Nr_x-1-AT” throughout the manuscript which is confusing and may mislead readers about the nature of the experiments. Alleles of genes in *Drosophila* are usually indicated with additional information in superscript which could easily be done, for example, using Nr_x-1^{AT} or Nr_xΔPDZ::AT for the endogenously tagged Nr_x-1. Similarly, overexpression constructs could be identified with UAS-XY.

We thank the reviewer for this great suggestion and have changed the nomenclature in the revised manuscript.

- Line 34: “assembles” does not seem to be the correct wording here.

We have replaced this word with “structures”.

- Line 121: “two (or even three)” reads oddly.

We have corrected this sentence.

- In almost all figure panels, the driver line identity is omitted (except for Figures 6 c and d) but should be included for clarity.

As the reviewer requested, we have included the driver line identity in the figure panels.

- Figure S2: There is only one scale bar, but the legend reads “scale bars”

We have corrected this typo.

- Line 292: a word seems to be missing.

- Figure 2: Amplitude mv or mV?

- Evoked excitatory junction potentials are abbreviated as EJPs mostly, but as eEJP in figure 3.

- Figure 3: mv or mV?

We have corrected these typos.

- Figure 3: It isn't easy to match traces and their respective quantifications in this figure.

- Lines 245 & 250: “docked” sounds strange in this context.

- Line 315 to 317: It is difficult to follow the authors here. In addition, there is no quantification that allows deriving that the signal is linear. It is not clear what polyclonal secondaries are.

In the revised manuscript we have assembled a table listing each reagent and experimental condition used for every immunohistochemistry assay in each figure (please see Supplementary Table 2).

- Line 330: It would be interesting to know when the majority of flies die and what the fraction of escapers is. Are the following experiments done with escapers?

Our immunohistochemistry experiments have been performed on homozygous larvae. A detailed analysis of the developmental stage(s) when most of these mutants die is beyond the scope of this study.

- Line 445: How is 100% penetrance defined?

We have rephrased this sentence.

- Line 580: "Supplementary S11" should likely be "S12".

We have corrected this typo.

- If Gaussian blurred images are subtracted for background reduction, it should be indicated when this method is used.

As the reviewer requested, we have indicated in the corresponding figure panels (Fig. 5h and Supplementary S13) when we applied this method.

- No information about sequence alignment and secondary structure prediction methods is available but should be included.

MacVector sequence analysis application (version 18.5) was used for sequence alignment and secondary structure prediction.

This information has been added to the Methods and in the revised Figure legend.

- Line 622: What exactly do the authors mean by "compartment-specific expression"?

We clarified that we refer to pre- vs post-synaptic compartments, as discussed in the previous sentence.

- Line 758: UAS-Nb-mCherry is apparently wrong.

We corrected this error.

Reviewer #3 (Remarks to the Author):

Key results: In this manuscript, Choudhury et al. investigate the subcellular distribution, C-terminal domain properties, and axonal trafficking of the Drosophila synaptic cell adhesion molecule neurexin-1 (Nrx-1). These qualities of mammalian neurexin-1 have been previously challenging to study due to its low abundance and its numerous splice variants that impede facile detection by conventional immunolabeling techniques. Moreover, tagging fly Nrx-1 with GFP at the C-terminus leads to aberrant subcellular localization and function. Choudhury et al. overcome these challenges via introduction of a small epitope tag (ALFA tag or AT) into an apparently tolerant region of the intracellular C-terminus of Nrx-1, enabling its detection via anti-AT nanobodies, either applied to fixed samples as immunolabels or expressed as genetically encoded intrabodies. They also use the AT/nanobody system to rescue a mutant form of Nrx-1-AT lacking a critical C-terminal PDZ binding motif (Nrx-1-AT- Δ PDZ) required for its presynaptic localization. By supplying the PDZ binding motif in trans via the anti-AT nanobody, they rescue Nrx-1-AT-PDZ mutant localization and presynaptic function.

Validity: Most conclusions and claims are supported, and the data and methodology presented in this paper appears to be valid overall. The methods include sufficient details to allow for the work to be reproduced. The data analysis is appropriate although quantitation of the immunolabeling results would enhance the authors statements as to the "remarkable" nature of labeling with the ALFATag nanobody system. The interpretation of some of the results are somewhat narrow and should be expanded to include other possibilities. There are numerous concerns as detailed below. Addressing these should markedly improve the manuscript.

Significance: The main strengths of this study are that 1) the authors demonstrate a method that enables more sensitive detection of a Nr_x-1, allowing for a better understanding of the subcellular distribution of Nr_x-1 and opening the door for other studies (e.g., immuno-EM analysis of synaptic Nr_xn1-AT); and 2) the authors demonstrate that the C-terminal PDZ domain of Nr_x-1 is necessary for its normal presynaptic distribution, and that defective Nr_x-1-AT-ΔPDZ can be rescued via PDZ-domain-containing AT-nanobody binding to this mutant. Broadly speaking, this study adds to the demonstrated utility of using nanobodies as immunolabels and as intrabodies. Overall, these findings seem appropriate for Nature Communications, however substantial revisions and major corrections to the present manuscript would be required.

Clarity and context: The clarity of the manuscript could be improved. The writing is in many places a bit rough, with missing words and incorrect subject-verb agreement (singular/plural, tense) and missing definite and indefinite articles, some of which are detailed below. The manuscript is also written in a manner that would be a bit challenging for those lacking specific expertise in the *Drosophila* NMJ model system, as to the authors not concisely presenting the rationale for certain experiments. While the authors appeared to provide relevant background information and references, understanding its significance to the present manuscript for the average reader may be helped by making the text more concise (e.g., limiting redundancy) and making more explicit statements about why an experiment/system was selected [for example, see comments below regarding the Nr_x-1-GAL4 CD4 experiments].

Specific comments and suggested improvements (Page/line)

Major:

6/173: The authors state they generated transgenes “to express equal levels of tagged and untagged Nr_x-1”. Is this demonstrated at the protein level? For example does the tag alter Nr_x-1 protein stability/abundance compared to WT Nr_x-1? If not, the authors should state that this was their intention of these strains but that whether the outcome was achieved has not been confirmed at the protein level.

Indeed, our intention was to remove the position effects from our transgene analyses. We accomplished this goal by precise targeting of transgenic constructs to predetermined genomic sites in *Drosophila* using the φC31 integrase system in conjunction with recombinase-mediated cassette exchange. This methodology ensures equal transcription but not protein levels.

We have revised this sentence to remove any confusion.

6/181: The authors state that strong neuronal expression of untagged Nr_x-1 “did not significantly enhance the Nr_x-1 signals in the neuropil”. Is this quantified? How are the relative expression levels deduced? Were statistical analyses performed to justify the use of the word “significantly”.

We have rephrased this sentence in the revised manuscript.

7/187: Here and in the other strains used in this section, simple quantification of the immunofluorescence signals in multiple samples is warranted.

Figure S2 should be incorporated into Figure 1. Most of the substantive evidence supporting the statements in the first results section (lines 185-199) seem to come from this figure. Figure 1b could be moved to supplementary data to make space for the info in S Figure 2.

We followed this recommendation (also suggested by reviewer #4) and incorporated the NMJ data from the original Supplementary S2 in the revised Fig. 1.

10/300-302: The authors do not perform any labeling to determine whether the recruitment of other common presynaptic molecules is affected (E.g., synaptophysin), such that their interpretation of structure is primarily based on the signal from the protein (Nrx-1) whose expression and properties they are manipulating.

The reviewer is correct, our interpretation is solely based on the phenotypes (NMJ growth and electrophysiology recordings) of different Nrx-1 variants. We clarified this interpretation in the revised manuscript.

12/362. Results section: The authors should provide the details of the structure of NbALFA-PDZ binding motif chimera as supplementary material, as per the other constructs. It is difficult to piece together what this comprises from the methods section. This information is much more critical than for example the information detailed in Supplementary Figure 5 which is quite obvious.

We followed this recommendation and added the sequence of NbALFA-PDZ chimera to the revised Supplementary S5.

14/418: This section appears largely superfluous to the rest of the study. Moreover, it seems based on the premise that the subcellular localization of exogenous CD4 should mimic that of Nrx-1. The proper experiment would be to examine Nrx-1-AT labeling in these regions

Following this suggestion and the next three recommendations from this reviewer, we have re-organized this section under the heading "Interfering with the PDZ binding motif alters Nrx-1 synaptic function".

In the first part, we emphasized the strong results that the PDZ domain is needed for presynaptic targeting of Nrx1 (see below). In the second part, we used the *Nrx-1-Gal4* to guide us towards cells/tissues expressing Nrx-1. We then detected the endogenously tagged Nrx1-AT in some of these cells using tissue specific expression of Nb-mScarlet. These results showcase how ALFA system could facilitate studies on protein expression.

14/435-445: The authors should make clear that they are not determining "Nrx-1 expression" but a proxy reporter for this.

The reviewer has a good point and we have incorporated this clarification in the revised manuscript.

16/473: The authors state GFP cleaved off the chimera, and on lines 489-490 "Therefore, the spreading of the GFP signals suggests a physical separation of the GFP tag". This is speculation. The premise that GFP "falls off" NRX-1 needs to be addressed experimentally, for example by immunoblot. It seems just as or more likely that these signals are from separate pools of intact Nrx-1-GFP proteins in distinct states and different properties including subcellular localization and immunoreactivity to anti-Nrx-1 antibody used (which the authors already argue is problematic).

This paragraph has been revised as follows:

The GFP spreading may reflect a higher mobility and/or flexibility of presynaptic Nrx-1-GFP; alternatively, the GFP and Nrx-1 signals may represent distinct pools of Nrx-1-GFP proteins distributed to different subcellular locations.

16/494-496: Masking is assumed. (also see line 243). Masking is a very specific molecular mechanism. I suggest using the more general model that GFP somehow "interferes" with the phenotype. There are numerous examples of PDZ binding motifs that do not need to be C-terminally exposed to function, and the GFP could be exerting allosteric effects on the overall structure of the cytoplasmic domain. Would be better to emphasize the strong results that the PDZ domain is needed for presynaptic targeting of Nr1.

As discussed above, we followed the reviewer's recommendation and reorganized the entire subsection under the heading *"Interfering with the PDZ binding motif alters Nr1 synaptic function"*.

We also revised and simplified the interpretation of the Nr1-GFP phenotypes. We therefore refer to the "C-terminal GFP-induced interferences with the PDZ binding motif of Nr1" in the revised manuscript.

18/529: This claim should be demonstrated by knocking out the WT Nr1 expression in the context of Nr1-AT and Nb expression.

This sentence offered an alternative explanation and was removed from the revised manuscript. Probing into the underlying mechanism is beyond the scope of this study.

18/530: Are these biologically relevant structures or artifacts of the expression system? Do you see them with immunolabeling (either conventional ab or nb)?

Extensive new analyses of the puncta, including immunolabeling, have been included in the revised Fig. 6 and Supplementary S16.

20/593: Overall discussion is a bit long and should be made more concise.

In the revised manuscript we have reduced the length of the Discussion section.

20/598: The authors draw an equivalence between the inclusion of a C-term GFP with a PDZ masking mechanism. This is too much of an assumption.

As indicated above, we took this concern seriously and discuss about "interference" instead of "masking" in the revised manuscript.

23/677: The authors should discuss what their results tell us about the location of the PDZ binding motif within the structure of the Nr1 cytoplasmic domain. Their results suggest the PDZ binding motif can be "moved" quite a bit upstream (via Nb-PDZ binding to AT sequence) of the C-terminus and still function. Does this suggest that the PDZ binding motif could be knocked-in to the AT locus in Nr1-GFP to rescue the "masking" defect?

Our results do not suggest that the PDZ binding motif can be "moved" upstream. When provided in trans, via the Nb-ALFA-PDZ chimera, the PDZ-binding motif was still localized at the C-end of the chimeric molecule and was presumably free to interact with PDZ-containing proteins (please also see the revised Supplementary S5). In contrast, an "internally moved" PDZ binding motif would be sandwiched between other sequences that will limit its ability to interact.

Minor:

2/53: citation format is incorrect

3/69: neurexins' (plural possessive?)

3/79: missing words – “not only”
3/89: abundance instead of abundant
4/100: intrabody capability
We have corrected these typos.

Page 4-5: These two paragraphs are more appropriate for and somewhat repetitive of the introduction. Perhaps can shorten this and move non redundant parts to intro

6/159-161: Insertion of AT within the extracellular domain? Clarify

As recommended, we clarified this sentence in the revised manuscript:
"Since an additional AT helix could interfere with the folding of extracellular domains of Nr_x-1 or disrupt the rigidity of the stalk, we searched for a suitable AT insertion site within the intracellular domain of Nr_x-1."

6/165: have variable expansions
6/181: traffics
8/242: suggests rather than indicates
These typos have been corrected in the revised manuscript.

9/251: The authors suggest that the differences are due to expression levels, Can this be demonstrated/quantified? See earlier comments about lack of quantification.

We removed this sentence since our new qPCR data (included in the new Supplementary S2), indicated a ~40 fold increase between endogenous and overexpressed *Nrx-1 mRNA* levels.

Lines 253-256: “The increased mEJP frequency and reduced quantal content indicate primarily presynaptic deficits, probably due to C-terminal GFP obstructing Nr_x-1 binding to interactors such as Scribble, Syd-1 or Spinophilin (Muhammad et al., 2015; Oswald et al., 2012; Rui et al., 2017).” This should be moved to the discussion section and alternative models should be provided.

This sentence has been removed.

10/288: Are N-terminal Nr_x-1 antibodies available and if so would the authors expect that these would yield similar results?

Unfortunately, N-terminal Nr_x-1 antibodies are not commercially available.

11/311: Or existence of distinct Nr_x1 populations with different AT epitope accessibility? Please address.

We opted to keep the simplest explanation in the revised text.

14/406: unsure what is meant by plastic
Replaced by "flexible".

14/4400-416: this paragraph is more appropriate for discussion

We respectfully disagree, as this paragraph immediately addresses the accessibility of the AT epitope.

14/420: "unprecedented ability": overstatement - wasn't the detection of NrX-1 AT by NbALFA the underlying thesis of this work?

This entire paragraph has been modified during the reorganization of this subsection.

16/482-483 "we found that the AT but not GFP signals perfectly co-localized with the NrX-1-positive immunoreactivities". The use of perfectly here is not scientifically valid.

The word was removed.

17/507: "fully functional" is presumed, should be tempered

Replaced with "functional".

19/563: Could the differences in velocity measurements be due to the elevated temperature of mammalian cells (37 degrees C) versus flies?

This is indeed a possibility that we mentioned in the revised text.

19/570: Can you fix the imaged cell and immunolabel for Rab2 to confirm?

Yes- we have done this successfully. These new experiments are described in the revised Fig. 6 and Supplementary S16.

20/610: This claim should be tempered. Numerous factors can affect background signal in different preparations.

This sentence simply summarizes our experience and anticipation for similar experiments.

21/622-623: not sure what is meant by this. do you just mean NrX-1 AT KI samples? compartment experiment would be better with different Nb-fluorescent protein fusion

We have modified this sentence for clarity:

"To sort through these possibilities, one could start from the edited NrX-1-AT and express Nb-mScarlet in specific compartments to capture different pools of synaptic NrX-1."

24/720: the beta-cell analogy is too specific and detailed given the scant evidence presented. The authors should temper language and generalize in a much more concise way to neuroendocrine cells, which use slightly different exocytotic machinery for DCV release and don't form synapses for fast NT release.

This sentence has been revised and simplified as follows:

"Similar to other neuroendocrine cells (Mosedale *et al.*, 2012; Suckow *et al.*, 2008), fly enteroendocrine cells may use NrX-1 to facilitate secretory granule docking and hormone release."

Reviewer #4 (Remarks to the Author):

The brief summary of my review is that I found this to be a very impressive research paper and recommend publishing it with the minor revisions listed below as well as my rationale for reaching this decision based on the given guidelines.

-Thomas Ravenscroft

1. What are the noteworthy results?

In this manuscript the authors use Nr_x-1 as an example protein whiches function, localization and trafficking can be monitored using nanobodies.

Specifically they:

1. Generate a new, endogenously tagged Nr_x1-ALFA allele that can provide high-resolution images with the ALFA-nanobody with minimal background staining that can study the localization of the protein both in synapses and in axons.
2. They show that the absence/covering of the PDZ domain in Nr_x1 causes functional deficits in the NMJ and that this can be rescued with some beautiful work reconstituting the PDZ domain in a trans location, bound to an ALFA-nanobody.
3. Absence of the PDZ domain causes Nr_x1 mislocalization, likely leading to electrophysiological phenotypes.
4. Live trafficking of Nr_x1a occurs in both retero and anterograde in axons.

2. Will the work be of significance to the field and related fields? How does it compare to the established literature? If the work is not original, please provide relevant references.

This work is significant in the fields of developmental biology and neuroscience as well as provides a great example of using nanobodies to study structure-function relationships. This has great potential to inspire many others to use this nanobody approach across many areas of scientific research

3. Does the work support the conclusions and claims, or is additional evidence needed?

By and large, the manuscript results do support the claims. However, one area that is missing is quantifying the levels of ALFA-tagged protein compared to non-tagged. Many times (indicated below) the claims "equal levels of expression is used" without seeing a western or even qPCR of expression levels. This is of interest as we see Nr_x1a-AT at the NMJ and not the native antibody. Plus all the trafficking data in the last 2 figures carry more weight with this quantified.

We have addressed this concern (also raised by other reviewers) by analyzing the expression levels for non-tagged and ALFA-tagged Nr_x-1 relative to wild-type via qPCR. These new data are included in the Supplementary S2.

Additionally, the specifications for how the gaussian blur filter was used for Figure 5 and Supplemental Figure 10 are also not clear to me.

As recommended, more details and clarifications on how we utilized Gaussian blur filter have been added to the revised Materials and Methods.

I can see mScarlett spots in the non-ALFA condition, they are smaller and less intense however so the comment should be more on size rather than a presence vs absence statement. I think a like-for-like comparison for axonal transport is needed to show the expression-dependent effects i.e show close-up axon tracking for 6c and d like 6a and b are.

This section has been tremendously expanded and improved as detailed above, in the response to reviewers #1 and #2.

A new set of completely new and improved movies has also been added to the revised manuscript. They include close-ups on the proximal regions of motor neurons.

4. Are there any flaws in the data analysis, interpretation and conclusions? - Do these prohibit publication or require revision?

In the above answer (3) I outline the issues I have with analysis and interpretation. I think these require just further explanation, perhaps just to myself and not in the manuscript, to address those concerns. Additionally, the comments on expression/levels can be addressed with quantified data or the text in the paper rephrased. These are minor revisions that should not take long to address.

We have addressed all these recommendations in the revised manuscript.

5. Is the methodology sound? Does the work meet the expected standards in your field?
Yes. It goes beyond those expectations.

6. Is there enough detail provided in the methods for the work to be reproduced?
Not quite, but with very straightforward clarification on some imaging procedures it will be.

As indicated by the reviewer, we have included additional clarifications on imaging procedures in the revised manuscript.

In addition, we have assembled a supplementary table listing each reagent and experimental condition used for every immunohistochemistry assay performed here (please see Supplementary Table 2).

Minor/Major comments below

- Minor - Fig 1a - Improve the key by having a separate key above the images
- Minor - Fig 1a - Annotate the glycosylated stalk on the diagram

Figure 1 was revised following these two suggestions.

- Minor - 135 - Active zones, limit NRX-1 activities - Not clear what this means

This sentence was simplified for clarity.

- Major - 174 - Claim that equal levels of transgene are expressed, RNA-seq/ a western blot will be needed to confirm this

New qPCR data support this claim (as showed in Supplementary S2).

- Minor- My opinion is that Supp figure 2 is more informative than Figures 1c-g. I recommend including this in the main figure along with the labelling in the control

As the reviewer suggested, the data from the initial Supplementary S2 has been added to the revised Fig. 1.

- Major - Fig 2 - Overexpression increases bouton number by 10%, this is problematic when making comparisons, also, GFP tagged NrX-1 does the same. The increase in bouton number in the OE conditions indicates that the release per bouton is lower. I think this should be commented on.

Indeed, the overexpression phenotypes could be problematic, especially since the UAS-Nrx-1-GFP was randomly inserted in the genome and could be leaky or exhibit positional effect. Consequently, we refrained from further interpretation of these results.

- Minor - 235 - And a non-tagged transgene
We have corrected this typo.

- Major - qPCR/westerns are needed to compare levels with all the comments on comparable expression
- Major - 277 - Indicate levels are the same, need some quantification of this (I know it is frustrating to ask this but with all the comparisons of levels etc. I think a western is needed to show differences in levels. Li et al show the Ab works well on adult head)

Unfortunately, we could not run Western with this antibody. Therefore, we have rephrased the text describing the comparisons.

- Minor - Figure 4 - The MIMIC diagram should label all components, both attR sites and both Minos repeats
The missing components have been added to the revised diagram.

- Minor- Figure 4, S7, S8 - Are these late or early 3rd instar larvae as the expression seems to change depending on these stages in muscle.
The animals imaged in these experiments are early third instar larvae.

- Minor - We need to see channel specific images for the sup figure 7 and 8, hard to see details in the composites
We have included single channel panels as requested.

- Minor - For figure 4 please show the non-composite image for Nrx-1 labelling
We have added the separate Nrx-1 channel as recommended.

- Minor- Supp figure 10 is more informative than figure 5a-h as you can see close-up details, I suggest swapping these or inserting S10 in the main figure
As the reviewer recommended, we have included new data corresponding to the previous Supplementary S10 in the main Fig. 5.
Also, the associated Supplementary S10 includes many details and extensive quantification.

- Major- I am not sure about the gaussian blur data showing the transport vesicles containing nrx1a. mScarlet is clearly transported along axons as it is present in the NMJ in s10 so we would expect to see it in axons as well. Furthermore the data in 5j' does show mScarlet signal in the axons. In addition it looks like the GFP positive spots in the axons of 5i are smaller and less numerous than in 5j. In the supplement we do see mscarlet spots in the non-AT condition but they appear drowned out by the higher signal in the cell body.

The reviewer is correct, mScarlet should be transported throughout the cell, but it fills the cytoplasm diffusely. Only when Nrx-AT is present into the system mScarlet becomes associated with the neuronal membrane.

- Further- how was that size (3) determined? I Assume it refers to pixel size? Looks like they are only selecting for larger vesicles, in which case they see larger vesicles with Nrx1a-ALFA.

Indeed, 3 represent the pixel size, as we further clarified in the revised methods.

- Minor - For figure 6 they show axon transport close up vs. low magnification view of the larval VNC. To show distinctly that there is no transport of the Nb-mScarlet in lieu of NrX-1A the same comparison needs to be made with the close up of axon transport with the Gal80 system.

Unfortunately, we observed too few vesicles along the axons with the Gal80 system and were unable to perform such comparison. Nonetheless, the VNC images have been replaced with higher quality images and moved in Supplementary S15 to make room in the revised Fig. 6 for more relevant Rab2 experiments. (Please also see the new supplementary movies.)

- Minor 644 - likely exit from the ER, without staining ER can't be sure, may be in Golgi so this needs to be rephrased

This point has been clarified by an additional new experiment where we labeled the ER with anti-Calnexin 99A. The new data are included in the Supplementary S6.

- Minor 750 - How many UAS' were used in the constructs one or 20? This needs to be stated. Same for Nb-mScarlet and Nb-PDZ

In all cases we used 20x UAS. This information has been added to the Materials and Methods.

- Minor 815 - Which samples used what fix?
- Minor 825 - Little or no detergent? This is not helpful, what was used in this study?
- Minor 816 - PBS or PBS Triton? Which samples had which and for how long? I know this is nitpicking but it needs to be repeatable
- Minor 820 - How long were samples incubated in primary/secondary? I recommend adding a table for each figure and list each antibody used and add to supplement similar to table S1

As recommended by two reviewers, #2 and #4, in the revised manuscript we have assembled a table listing each reagent and experimental condition used for every immunohistochemistry assay in each figure (please see Supplementary Table 2).

- Minor - 835 - "Imaris and ImageJ respectively" Respective to what?

This has been corrected.

- Minor 838 - Should cite the filament algorithm reference (Flormann 2021 faseb)

Thank you for pointing that out. We added the missing reference in the revised manuscript.

- Minor - 863/866 - Zoom 3x?

These typos have been corrected.

Minor - 863 - Please report the frame rate

The frame rate and other details have been included in the revised Methods.

"... the frame size was 1024x1024 pixels with a scan speed of 2.6×10^6 pixels/second; each frame was captured every 5 seconds for a total of 10 minutes."

Reviewers' Comments:

Reviewer #1:

Remarks to the Author:

The authors have now thoughtfully revised their manuscript and added substantial amounts of new information and experiments. The authors introduce a new tagging system to visualize endogenous wild-type and truncated *Drosophila* Neurexin-1 (Nrx-1) in motor neurons and at NMJs. The authors introduce, and extensively characterize, new genetic tools to examine the function, subcellular localization and transport of this important synaptic receptor, revealing that the C-terminal PDZ-domain binding motif is required for membrane localization. In this solid piece of work, the authors largely confirm previous findings and but also provide new information by showing that Nrx-1 associates with Rab2-positive vesicles. I have only a few further comments.

Minor points:

a) Line 544-545: Dilution effects of a single Gal4 driver acting on several UAS-promoters should reduce signal intensity of the particles but should not have an effect on their numbers. The dilution effect might thus not fully explain the presence of only a few axonal puncta. As suggested previously, axonal puncta could be enriched experimentally, either by using mutants with axonal transport defects or by inducing a nerve crush.

b) Legend to Fig. 3l-o should be better described for readers unfamiliar with VNC anatomy. Since images in 3l and 3n have quite a different appearance it should be clarified if they represent the same Z-level. The inset and the term "subcellular distribution" should also be specified more precisely. What is the nature of the residual, non-overlapping Nrx-1 staining in the midline (Fig. 3l) or the lateral borders of the VNC (Fig. 3n)?

c) The purpose/benefit of Suppl. Fig. S16 should be better explained. It appears that the images in Fig. 6e-h and Fig. 6l-o are re-analyzed in Suppl. Fig. S16. If yes, it has to be clearly marked that these are the same images.

d) Figure 6e-f and 6l-m: The pattern of the segmented puncta (Fig. 6f and 6m) does not seem to reflect the intensity pattern of the HRP spots in the corresponding confocal images (Fig. 6e and 6l). Please modify.

e) Legend to Fig. 6, line 1309, reads "(d-f)" but should be "(d-j)". It might be important here to mention that the images show cytoplasmic particles in motor neurons, which should not be confused with particles in axons.

Reviewer #2:

Remarks to the Author:

The manuscript has been improved considerably by adding new data and information, and most of my concerns have been addressed in full or explained satisfactorily. The paper will show a variety of powerful applications of the ALFA tag system and also make a novel contribution to the field of synaptic cell adhesion molecules, perhaps most notably by demonstrating the trafficking and colocalization of endogenous *Drosophila* Nrx-1 with Rab2-positive vesicles.

Some suggestions for minor issues (sorted by text lines):

line 120 ff - much of this first section of Results seems more appropriate for the Introduction

237 ff - 'N' is used to abbreviate BG380-Gal4 in some figures, but 'N(BG380-Gal4)' or 'BG380-Gal4' is used, too. This should be consistent. In addition, 'N' is also the abbreviation for the gene Notch. Together with the authors' choice to label the Neurexin-Gal4 driver Nrx-Gal4 this is

confusing. Maybe BG380-Gal4 or something more descriptive, like 'motoneuron-Gal4', would work. The same issue is the case for the 'controls', which are sometimes indicated by genotype (w1118) and sometimes as 'control'.

252 - The study does not show that ALL functions are conserved.

333: What exactly does it mean that a distribution is consistent with electrophysiological recordings?

418: From the context, it is not entirely clear what experimental setting is referred to here. GFP is used as an nls::GFP construct in the previous section but in this section?!

421 - What does 'very high co-localization [...] within a distance close to the limit of STED microscopy' actually mean?

435 - The description of this experiment is difficult to follow. What exactly was done? Which reporters are referred to?

511 - This section is confusingly structured. Information on the speed of vesicles should likely not follow a sentence that states that there are no vesicles found in controls?!

528 - How can the authors distinguish motor and sensory neurons, when using the Nr_x-Gal4 driver?

547 ff - How can the authors be sure they have counted motor neuron somata - there are also interneuron somata in the larval VNC?!

785 - The authors state that BG380-Gal4, a very important line for their studies, was previously described, but I cannot find the original publication. Neither do they indicate an online resource to get more information about the stock. The same issue applies to their Nr_x-1(273) null allele.

Reviewer #3:

Remarks to the Author:

There remain some aspects of this manuscript that are of interest and that could potentially justify publication in Nature Communications. The reviews of the original submission listed a number of concerns related to the rigor of the results and constructive suggestions of how to address these to bring the study to a level consistent with publication. Although the authors include a fair amount of new data and have revised the language in the revised manuscript, many of the concerns previously noted were not addressed. It is disappointing that these suggestions were disregarded, as the authors presumably already have collected the data (e.g., measurement and comparison of immunofluorescence intensity in images from Nr_x1 in WT and edited samples). The authors also make claims that are still unsupported by the presented evidence, most notably the role of the Nr_x1 PDZ binding motif in the synaptic targeting/function of Nr_x1. These issues should be addressed experimentally. Alternatively the authors will need to further revise the manuscript to identify and clarify these major caveats. Due to these continuing issues the manuscript is not suitable for publication in its present form.

(page/line)

Major:

8/239-240: The authors state that "Since the Nr_x-1-GFP transgene effectively rescued the morphological defects of Nr_x-1 null mutants, the net levels of synaptic Nr_x-1-GFP cannot explain its inability to rescue the NMJ function". This is not necessarily the case, especially as stated by the

authors themselves on the next page (lines 266-268): the NMJ electrophysiological parameters are very sensitive to the dose of Nr_x-1: loss of one copy of Nr_x-1 as well as neuronal overexpression of Nr_x-1 lead to impaired evoked synaptic transmission and reduced quantal content (Li et al., 2007).

10/279-280; 291-292: The authors state that they cannot "capture" (label) Nr_x1-AT at larval NMJs using C-terminal antibodies "at comparable confocal settings" used to detect labeling with the ALFA tag. However, previous studies have shown that unedited WT Nr_x1 can be detected at NMJs with these same antibodies (e.g., as in Li et al. 2007). The degree of immunolabeling should be quantified as previously recommended as a proxy measure of protein abundance, or via western blot as performed by Li et al 2007. It is unclear from the presented data whether the edited Nr_x1 AT protein displays similar abundance as the endogenous Nr_x1 protein in WT flies. The altered morphological qualities of NMJs in the edited flies phenocopies the Nr_x1 overexpression flies, perhaps implying that Nr_x1 AT is more stable/abundant than WT Nr_x1. As such, claims of superior sensitivity of the AT-Nb approach should be tempered with the caveat that it is unknown whether similar levels of target (ie., Nr_x1-AT vs Nr_x1 WT) were present in the imaged samples.

13/389-392: The data presented support that Nb-PDZ allows for mutant Nr_x1 deltaPDZ-AT to exit the ER and traffic to synapses, but is synaptically targeted Nr_x1 deltaPDZ-AT still engaged with Nb-PDZ? At least a subset of synaptic Nr_x1dPDZ-AT must have dissociated from Nb to present the Nb-accessible epitope labeled in fixed cells. At a minimum, in the absence of additional data the caveat should be noted.

13/394: The themes in this section require further development, and overall this section still feels disconnected from the rest of the manuscript. It does not add much to the rest of the study as this section's premise, that the presence of GFP results in disrupted PDZ binding motif function, is still not rigorously tested. The information about the Gal4 system could be retained for the subsequent section ("Compartment-specific detection..")

14/429-433: The way this section is written is confusing and overly speculative. It appears the authors draw a conceptual equivalency between Nr_x1-GFP and impaired PDZ binding motif function, such that "interfering with PDZ binding motif" is synonymous with Nr_x1-GFP. While the data show that Nr_x1-GFP displays altered synaptic distribution and impaired synaptic function, the assertion that these defects are due to faulty PDZ binding motif interactions of synaptic Nr_x1-GFP is not rigorously tested, for example in direct binding assays to an established PDZ domain-containing binding partner. One alternative hypothesis is that the Nr_x1 PDZ binding motif is dispensable for synaptic function, but required for ER exit (as the manuscript shows). Why is Nb-engaged synaptic Nr_x1 functional, but Nr_x1-GFP is not, if position of PDZ binding motif is "flexible"? While presence of the GFP molecule may cause, for example, steric hindrance of interactions, including PDZ binding motif-dependent interactions, it is interesting that nanobodies are approximately the same overall dimensions as GFP, and Nb-PDZ is presumably engaged with synaptic Nr_x1 dPDZ-AT. To support their conclusions, the authors should test the necessity of a C-terminally exposed PDZ binding motif in Nr_x1 by making a Nr_x1-GFP construct with the PDZ binding motif on the C-terminus, rather than on the N-terminus of GFP. Otherwise the claims here should be greatly tempered or removed in the absence of demonstrating that the addition of GFP to Nr_x1-GFP results in interference with the PDZ binding motif and loss of binding to established PDZ domain-containing binding partner.

19/563-564: See comments on lines 429-433. "Interference with PDZ binding motif" (i.e., Nr_x1-GFP?) is asserted but not tested. The authors demonstrate that the PDZ binding motif is necessary for Nr_x1 trafficking, but its necessity for synaptic function based on Nr_x1-GFP functional qualities is not compelling.

21/639-641: This is a testable hypothesis. Moreover, nanobodies are about the same size as GFP. Why isn't Nb-PDZ causing the same issue as Ct GFP? How much synapse-targeted Nr_x1-AT is still Nb engaged?

Minor:

6/163: "...Nrx-1 proteins has..." appears to be a subject/verb error

13/382-384: I am not quite sure what is meant by "flexible" - are you saying that the position of the PDZ bm in the Nrx1 Ct is flexible? This should be clarified.

19/562: "endogenous levels of" : no quantitation of endogenous WT Nrx1 or edited Nrx1 AT protein has been performed as requested. I suggest removing any reference to "levels."

20/608: Necessity of PDZ binding motif for Nrx1 synaptic stabilization is not clear from the present data.

20-21/614-622: As written this seems conceptually disconnected from the discussion of the PDZ binding motif.

21/627: By "flexibility" do the authors mean absolute position of this motif in the Nrx1 C-terminus?

21/634: homogeneously

Reviewer #4:

Remarks to the Author:

The authors have addressed all my prior concerns. I think this is a good study that warrants publication, however, I do have concerns remaining regarding the new colocalization experiments with rab2 in Figure 6 that should be addressed before acceptance.

1. The selected regions for performing the colocalization appear arbitrary and in Figure 6g you can see in the top left and bottom left of the highlighted region Nrx-1 positive punctae with no rab2 signal. Can the authors explain why these regions were chosen? I think the % correlation between Nrx-1 and Rab2 will be lower should these cells be included (unless Rab2 signal is being masked by the Nrx-1 which leads to comment 2)

2. For Figure 6g, it is important to show the rab2 and Nrx-1 panels individually to see if those cells do indeed express Rab2, more so than what is in 6k-r which is not super informative as the scarlet signal is uniform in the absence of the ALFA tag.

3. The authors should acknowledge that there appears to be a cell-type-specific expression of Rab2 and therefore transport of Nrx1 can not be solely dependent on Rab2. This is also true as 25% of Nrx1 vesicles are not in the vicinity of Rab2.

4. The authors claim that Nrx-1 comigrates with Rab2 to synaptic terminals without showing any colocalization in synaptic terminals (Line 554). This should either be changed to colocalization in vesicles, or NMJ stainings showing colocalization should be shown.

Response to Reviewers:

I would like to express my genuine appreciation to the reviewers for their time and careful consideration of this manuscript and for the constructive criticism. All of the reviewers' recommendations have been incorporated in the revised manuscript.

The most significant change is an additional set of experiments, suggested by reviewer #3, in which we tested side by side how Nb-PDZ and Nb-PDZ-GFP chimera rescue the morphological and physiological deficits of *Nrx*^{ΔPDZ-AT} animals. We found that larva with reconstituted Nrx-1-GFP (*Nrx*^{ΔPDZ-AT}/Nb-PDZ-GFP) phenocopy *Nrx-1*^{null} mutants rescued by neuronal expression of Nrx-1-GFP. These results (included in the revised Fig. 4 and Supplementary Fig. 9) demonstrate that the hindrance of the PDZ binding motif has minimal (if any) influence onto Nrx-1 surface expression and synaptic trafficking and instead impacts Nrx-1 functioning at presynaptic sites.

Below is the point-by-point response to reviewers.

Reviewer #1 (Remarks to the Author):

The authors have now thoughtfully revised their manuscript and added substantial amounts of new information and experiments. The authors introduce a new tagging system to visualize endogenous wild-type and truncated Drosophila Neurexin-1 (Nrx-1) in motor neurons and at NMJs. The authors introduce, and extensively characterize, new genetic tools to examine the function, subcellular localization and transport of this important synaptic receptor, revealing that the C-terminal PDZ-domain binding motif is required for membrane localization. In this solid piece of work, the authors largely confirm previous findings and but also provide new information by showing that Nrx-1 associates with Rab2-positive vesicles. I have only a few further comments.

We thank the reviewer for recognizing our efforts (new information and experiments) to improve this manuscript. For this re-revised version, we generated even more reagents and expanded the experiments to dissect additional functions for the C-terminal PDZ-domain binding motif.

Minor points:

a) Line 544-545: Dilution effects of a single Gal4 driver acting on several UAS-promoters should reduce signal intensity of the particles but should not have an effect on their numbers. The dilution effect might thus not fully explain the presence of only a few axonal puncta. As suggested previously, axonal puncta could be enriched experimentally, either by using mutants with axonal transport defects or by inducing a nerve crush.

As the reviewer mentioned, we realize that crushing the nerve or using additional mutants with axonal transport defects would be great experimental ways to enrich the axonal puncta. However, such extensive trafficking studies are beyond the scope of this manuscript. Here we wished to showcase the feasibility of such *in vivo* studies using the ALFA system. In addition, we captured a significant association of the Nrx-1 with Rab2-positive vesicles. Nonetheless, we are hopeful that our work will inspire scientists who study synaptic trafficking (and know how to crush nerves) to include the ALFA system in their toolbox.

b) Legend to Fig. 3l-o should be better described for readers unfamiliar with VNC anatomy. Since images in 3l and 3n have quite a different appearance it should be clarified if they

represent the same Z-level. The inset and the term "subcellular distribution" should also be specified more precisely.

As the reviewer requested, we included clarifications on the subcellular distribution of Nr_x1 in both the Fig. 3 legend as well as the associated main text, as follows:

Main text:

Editing the *Nrx-1* locus also preserved the normal subcellular localization of Nr_x-1 (Fig. 3 f-i). First, both the wild-type Nr_x-1 and the endogenously tagged Nr_x-1^{AT} concentrated at the nerve cord neuropil, a synapse-rich region comprised of dendrites, axons and glial cell processes.

Figure 3. Detection and *in vivo* reconstitution of endogenously edited ALFA-tagged Nr_x-1.

(a-b) Representative traces of mEJPs and EJPs in control and endogenously tagged *Nrx-1*^{AT} third instar larvae.

(c-e) Quantification of mEJP amplitude, EJP amplitude and Quantal Content in the indicated genotypes.

(f-i) Confocal images of VNCs or muscle 6/7 NMJs of third instar larvae of the indicated genotypes labeled for ALFA tag (green) and Nr_x-1 or HRP (magenta). The images were acquired with the same confocal microscopy settings and scaled equally for direct visual comparison. The edited Nr_x-1^{AT} shows the expected Nr_x-1 accumulation in neurites.

(j-k) Representative traces of mEJPs and EJPs recorded in *Nrx*^{ΔPDZ-AT} third instar larvae alone (**j**) or in the presence of neuronally expressed Nb-PDZ chimera (**k**). Deletion of the PDZ binding motif of Nr_x-1 causes loss-of-function phenotypes that resemble the *Nrx-1*^{null} mutant deficits. These phenotypes are completely rescued by neuronal expression of the Nb-PDZ chimera.

(l-o) Confocal images of VNCs or muscle 6/7 NMJs of third instar larvae of the indicated genotypes labeled for ALFA tag (green) and Nr_x-1 or HRP (magenta). The absence of a PDZ binding motif shifts the distribution of Nr_x^{ΔPDZ-AT} from the neurites to the motor neuron soma, more specifically in the ER. Addition of the PDZ binding motif *in trans* restores the cell surface location and normal distribution of reconstituted Nr_x-1 (*Nrx*^{ΔPDZ-AT}/Nb-PDZ) to the neuropil.

What is the nature of the residual, non-overlapping Nr_x-1 staining in the midline (Fig. 3l) or the lateral borders of the VNC (Fig. 3n)?

The lateral VNC cells are motor neurons labeled by the *BG380-Gal4* driver.

The residual midline staining was only seen in *Nrx*^{ΔPDZ-AT} VNCs. We don't understand what its nature.

c) The purpose/benefit of Suppl. Fig. S16 should be better explained. It appears that the images in Fig. 6e-h and Fig. 6l-o are re-analyzed in Suppl. Fig. S16. If yes, it has to be clearly marked that these are the same images.

e) Legend to Fig. 6, line 1309, reads "(d-f)" but should be "(d-j)". It might be important here to mention that the images show cytoplasmic particles in motor neurons, which should not be confused with particles in axons.

We have completely reorganized Fig. 6 and the associated Suppl. Fig. S17 as requested by this reviewer and by reviewer #4 (please see below). We have also clearly stated that S17 is an extension of Fig. 6 and stressed that the analyses cover the soma of several motor neurons.

d) Figure 6e-f and 6l-m: The pattern of the segmented puncta (Fig. 6f and 6m) does not seem to reflect the intensity pattern of the HRP spots in the corresponding confocal images (Fig. 6e and 6l). Please modify.

Thank you for pointing this out - we modified the HRP panels.

Reviewer #2 (Remarks to the Author):

The manuscript has been improved considerably by adding new data and information, and most of my concerns have been addressed in full or explained satisfactorily. The paper will show a variety of powerful applications of the ALFA tag system and also make a novel contribution to the field of synaptic cell adhesion molecules, perhaps most notably by demonstrating the trafficking and colocalization of endogenous *Drosophila* Nr_x-1 with Rab2-positive vesicles.

We thank the reviewer for the appreciative comments. We too hope that our work will prompt the field to take full advantage of the ALFA system's versatility.

Some suggestions for minor issues (sorted by text lines):

line 120 ff - much of this first section of Results seems more appropriate for the Introduction

We too pondered over the best place for this description of the Nr_x1 topology and interactions and realized that the reader would benefit most from it in the context of searching for a suitable site for the ALFA tag insertion.

237 ff - 'N' is used to abbreviate BG380-Gal4 in some figures, but 'N(BG380-Gal4)' or 'BG380-Gal4' is used, too. This should be consistent. In addition, 'N' is also the abbreviation for the gene Notch. Together with the authors' choice to label the Neurexin-Gal4 driver Nr_x-Gal4 this is confusing. Maybe BG380-Gal4 or something more descriptive, like 'motoneuron-Gal4', would work. The same issue is the case for the 'controls', which are sometimes indicated by genotype (w1118) and sometimes as 'control'.

The reviewer is correct, trying to simplify the labeling while keeping the detailed information for the aficionados created unnecessary complications. Consequently, in the main text we used a descriptive labeling "MN-Gal4", that we defined up front as short-hand for motor neuron-specific Gal4, *BG380-Gal4*. We trust this will remove any remaining ambiguities.

We also revised the labels for controls for consistency.

252 - The study does not show that ALL functions are conserved.

As recommended by the reviewer, we toned down this sentence as follows:

In contrast, insertion of the compact AT in a non-conserved cytoplasmic region has a minimal fingerprint and it preserved both NMJ morphology and synaptic transmission functions of Nr_x-1.

333: What exactly does it mean that a distribution is consistent with electrophysiological recordings?

Thank you for pointing out the need for clarification. This sentence now reads as follows:

This distribution is consistent with the deficits captured by electrophysiological recordings...

418: From the context, it is not entirely clear what experimental setting is referred to here. GFP is used as an nls::GFP construct in the previous section but in this section?!

421 - What does 'very high co-localization [...] within a distance close to the limit of STED microscopy' actually mean?

We clarified these points by rephrasing as follows:

We compared the distribution of *Nrx-1-Gal4* expressed *Nrx-1-AT* and *Nrx-1-GFP* chimera at synaptic terminals using stimulated emission depletion (STED) microscopy (Fig. 4d-g). Scatter plots of *Nrx-1* and *AT* or *GFP* signals revealed co-localization with a Pearson's coefficient of 0.93 for *Nrx-1* and *AT*, within a distance of 15.80 ± 9.60 nm, nearing the resolution limit achievable by STED microscopy (Supplementary Fig. 8).

435 - The description of this experiment is difficult to follow. What exactly was done? Which reporters are referred to?

This entire section has been reorganized to address the suggestions from this reviewer and from reviewer #3 (please see below). We performed additional experiments to show that obscuring the PDZ binding motif (through the addition of a C-terminal GFP tag) primarily impacts *Nrx-1* functioning at presynaptic sites. These new data are included in the revised Figure 4 and in the new Supplementary Fig. 9.

511 - This section is confusingly structured. Information on the speed of vesicles should likely not follow a sentence that states that there are no vesicles found in controls?!

We reorganized the section as suggested by the reviewer.

528 - How can the authors distinguish motor and sensory neurons, when using the *Nrx-Gal4* driver?

We cannot. This is why we refer to the vesicles' movement collectively "towards" or "away from" the VNC.

547 ff - How can the authors be sure they have counted motor neuron somata - there are also interneuron somata in the larval VNC?!

The location of motor neuron cell bodies within each segment is highly stereotyped and easily recognizable. In addition, motor neuron soma are primarily located dorsally, making them readily identifiable under confocal microscopy. Their distinct localization and stereotyped pattern as well as our extensive work on larval motor neurons give us confidence that we are isolating and examining motor neurons in these experiments.

785 - The authors state that BG380-Gal4, a very important line for their studies, was previously described, but I cannot find the original publication. Neither do they indicate an online resource to get more information about the stock. The same issue applies to their *Nrx-1(273)* null allele.

The BG380-Gal4 line was described by Vivian Budnik (Budnik et al. 1996). The *Nrx-1[273]* allele was generated in Manzoor Bhat laboratory (Li et al. 2007). We have added these references to the "Molecular constructs and fly stocks" section in Materials and Methods.

Reviewer #3 (Remarks to the Author):

There remain some aspects of this manuscript that are of interest and that could potentially justify publication in Nature Communications. The reviews of the original submission listed a number of concerns related to the rigor of the results and constructive suggestions of how to address these to bring the study to a level consistent with publication. Although the authors include a fair amount of new data and have revised the language in the revised manuscript, many of the concerns previously noted were not addressed. It is disappointing that these suggestions were disregarded, as the authors presumably already have collected the data (e.g., measurement and comparison of immunofluorescence intensity in images from Nr_x1 in WT and edited samples). The authors also make claims that are still unsupported by the presented evidence, most notably the role of the Nr_x1 PDZ binding motif in the synaptic targeting/function of Nr_x1. These issues should be addressed experimentally. Alternatively the authors will need to further revise the manuscript to identify and clarify these major caveats. Due to these continuing issues the manuscript is not suitable for publication in its present form.

We thank the reviewer for the thorough review of this study and for the constructive criticism. We followed these recommendations and performed additional experiments, as described below.

(page/line)

Major:

8/239-240: The authors state that “Since the Nr_x-1-GFP transgene effectively rescued the morphological defects of Nr_x-1 null mutants, the net levels of synaptic Nr_x-1-GFP cannot explain its inability to rescue the NMJ function”. This is not necessarily the case, especially as stated by the authors themselves on the next page (lines 266-268): the NMJ electrophysiological parameters are very sensitive to the dose of Nr_x-1: loss of one copy of Nr_x-1 as well as neuronal overexpression of Nr_x-1 lead to impaired evoked synaptic transmission and reduced quantal content (Li et al., 2007).

We thank the reviewer for catching this mistake: loss of one copy of Nr_x-1 has an effect on NMJ morphology, not on NMJ electrophysiology. We removed this sentence (lines 266-268) from the revised manuscript.

The statement

“Since the Nr_x-1-GFP transgene effectively rescued the morphological defects of Nr_x-1 null mutants, the net levels of synaptic Nr_x-1-GFP cannot explain its inability to rescue the NMJ function.”

remains and is further reinforced by additional experiments (below).

10/279-280; 291-292: The authors state that they cannot “capture” (label) Nr_x1-AT at larval NMJs using C-terminal antibodies “at comparable confocal settings” used to detect labeling with the ALFA tag. However, previous studies have shown that unedited WT Nr_x1 can be detected at NMJs with these same antibodies (e.g., as in Li et al. 2007). The degree of immunolabeling should be quantified as previously recommended as a proxy measure of protein abundance, or via western blot as performed by Li et al 2007. It is unclear from the presented data whether the edited Nr_x1 AT protein displays similar abundance as the endogenous Nr_x1 protein in WT flies. The altered morphological qualities of NMJs in the edited flies phenocopies the Nr_x1 overexpression flies, perhaps implying that Nr_x1 AT is more stable/abundant than WT Nr_x1. As such, claims of superior sensitivity of the AT-Nb approach should be tempered with the caveat

that it is unknown whether similar levels of target (ie., Nr_x1-AT vs Nr_x1 WT) were present in the imaged samples.

We apologize for the misunderstanding. Here we used a rabbit polyclonal antibody generated by the late Dave Featherstone against a synthetic peptide, ELRLPAQRTSTSAFESPDLR. The Li et al study used a guinea pig polyclonal antibody generated in Manzoor Bhat laboratory against a recombinant protein containing the cytoplasmic region of Nr_x-1 fused with a His6 tag at the N terminus. As expected, the two antibodies appear to have different properties. Also, the guinea pig, but not the rabbit polyclonal antibody, could be used for Western blots.

As such, the need for a different solution for detection of synaptic proteins, a solution that would solve both the variability of immunohistochemistry results and the limited availability of reagents, cannot be overstated.

To address the reviewer's concerns regarding the Nr_x-1 levels, we imaged, quantified and compared the levels of Nr_x-1 in wild-type, *Nrx-1^{AT}* edited and *Nrx-1* mutant VNCs. These new data, included in Suppl. Fig. 5, demonstrate that the insertion of the AT does not significantly change the level of net Nr_x-1, as detected with the anti-Nr_x-1 rabbit polyclonal antibody. The slight increases in the number of boutons and the NMJ length observed in *Nrx-1^{AT}* are within the range of background variations and may reflect our use of *w¹¹¹⁸* as control instead of the *w¹¹¹⁸; [nos-Cas9]attP40/CyO* line used for germline transformation.

13/389-392: The data presented support that Nb-PDZ allows for mutant Nr_x1 deltaPDZ-AT to exit the ER and traffic to synapses, but is synaptically targeted Nr_x1 deltaPDZ-AT still engaged with Nb-PDZ? At least a subset of synaptic Nr_x1 deltaPDZ-AT must have dissociated from Nb to present the Nb-accessible epitope labeled in fixed cells. At a minimum, in the absence of additional data the caveat should be noted.

We thank this reviewer for the thoughtful comment, which is an important note for the reader. As recommended, we revised these sentences to discuss the implication of our results, as follows:

Finally, the high affinity of the ALFA system was not a barrier in the detection of reconstituted proteins in fixed tissues: We have successfully used fluorescently labeled Atto-NbALFA for detecting reconstituted Nr_x-1 (Nr_x^{DPDZ-AT}/Nb-PDZ). This indicates that at least a subset of synaptic Nr_x^{DPDZ-AT} must have dissociated from Nb-PDZ to become accessible for detection.

13/394: The themes in this section require further development, and overall this section still feels disconnected from the rest of the manuscript. It does not add much to the rest of the study as this section's premise, that the presence of GFP results in disrupted PDZ binding motif function, is still not rigorously tested. The information about the Gal4 system could be retained for the subsequent section ("Compartment-specific detection..")

This section has been recommended by this reviewer in the previous round of reviews; we were grateful for this suggestion as it greatly enhanced the overall structure of the manuscript, especially in the re-revised format.

The reviewer is correct, the previous revision of this manuscript did not directly test whether the presence of GFP results in disrupted PDZ binding motif function. A rigorous test required additional experiments that we performed and included in this re-revised manuscript (as described below).

We also follow the reviewer's recommendation and moved the *Nrx-1-Gal4* expression data in the subsequent section.

14/429-433: The way this section is written is confusing and overly speculative. It appears the authors draw a conceptual equivalency between *Nrx1-GFP* and impaired PDZ binding motif function, such that "interfering with PDZ binding motif" is synonymous with *Nrx1-GFP*. While the data show that *Nrx1-GFP* displays altered synaptic distribution and impaired synaptic function, the assertion that these defects are due to faulty PDZ binding motif interactions of synaptic *Nrx1-GFP* is not rigorously tested, for example in direct binding assays to an established PDZ domain-containing binding partner. One alternative hypothesis is that the *Nrx1* PDZ binding motif is dispensable for synaptic function, but required for ER exit (as the manuscript shows). Why is Nb-engaged synaptic *Nrx1* functional, but *Nrx1-GFP* is not, if position of PDZ binding motif is "flexible"? While presence of the GFP molecule may cause, for example, steric hindrance of interactions, including PDZ binding motif-dependent interactions, it is interesting that nanobodies are approximately the same overall dimensions as GFP, and Nb-PDZ is presumably engaged with synaptic *Nrx1* dPDZ-AT. To support their conclusions, the authors should test the necessity of a C-terminally exposed PDZ binding motif in *Nrx1* by making a *Nrx1-GFP* construct with the PDZ binding motif on the C-terminus, rather than on the N-terminus of GFP. Otherwise the claims here should be greatly tempered or removed in the absence of demonstrating that the addition of GFP to *Nrx1-GFP* results in interference with the PDZ binding motif and loss of binding to established PDZ domain-containing binding partner.

We appreciate the efforts of this reviewer to suggest direct experimental test that will prove or disprove whether the presence of C-terminal GFP results in disrupted PDZ binding motif function. We also thought a lot about a more rigorous demonstration. We first considered the reviewer suggestion to modify the relative position between the GFP and the PDZ binding domain. However, our AlphaFold modeling studies indicate that in this configuration the PDZ domain would be kept relatively far away from the rest of the *Nrx-1* intracellular domain (and would introduce a beta-barrel in between).

An alternative strategy, applicable also to probing the function of any other PDZ binding motifs, would be to expand our "in trans" rescue experiments to include an Nb-PDZ-GFP chimera. Specifically, we wished to compare the "in trans" activity of a C-terminally exposed PDZ binding motif vs. one that is followed by a GFP tag, which may hinder some of the functionally relevant interactions.

This experiment required several steps: (a) generate a chimera with three modules, Nb-ALFA, PDZ-binding motif followed by GFP, (b) introduce it along the Nb-PDZ at the same docking site in the fly genome, and (c) test the resulting transgenes side by side in the *Nrx^{ΔPDZ-AT}* background for functional reconstitution *in vivo*. Albeit risky, this experiment had the potential to provide a great deal of understanding of the role of the PDZ binding motif in our case study protein, *Nrx-1*, and, if successful, could tremendously benefit our readers and their studies on other PDZ-binding motifs.

This new experiment and the results are included in Fig. 4 and Suppl. Fig. 9. In brief, we found that the reconstituted *Nrx-1-GFP* (*Nrx^{ΔPDZ-AT}/Nb-PDZ-GFP*) can exit the ER/early Golgi compartment and traffic to the cell surface. Also, these animals show normal NMJ morphology, indicating that reconstituted *Nrx-1-GFP* reached the synaptic terminals. However, larvae with reconstituted *Nrx-1-GFP* (*Nrx^{ΔPDZ-AT}/Nb-PDZ-GFP*) exhibit electrophysiological deficits similar to *Nrx-1* null mutants rescued by *Nrx-1-GFP*. Together these results demonstrate that the

hindrance of the PDZ binding motif has minimal (if any) influence on Nr-x-1 surface expression and synaptic trafficking and instead impacts the Nr-x-1 function at presynaptic sites.

19/563-564: See comments on lines 429-433. "Interference with PDZ binding motif" (i.e., Nr-x1-GFP?) is asserted but not tested. The authors demonstrate that the PDZ binding motif is necessary for Nr-x1 trafficking, but its necessity for synaptic function based on Nr-x1-GFP functional qualities is not compelling.

This comment has been addressed above.

21/639-641: This is a testable hypothesis. Moreover, nanobodies are about the same size as GFP. Why isn't Nb-PDZ causing the same issue as Ct GFP? How much synapse-targeted Nr-x1-AT is still Nb engaged?

This issue has been addressed above.

Our new data indicate that the PDZ binding motif must have a free C-end for engaging in the appropriate interactions at synaptic terminals. Since using the pair of docked *UAS-Nb-PDZ* and *UAS-Nb-PDZ-GFP* transgenes we cannot detect the AT at the synaptic terminals, we speculate that most of the synaptic targeted, reconstituted Nr-x1-AT(-GFP) is still Nb engaged.

Minor:

6/163: "...Nr-x-1 proteins has..." appears to be a subject/verb error

Thank you for catching this typo, which we have corrected.

13/382-384: I am not quite sure what is meant by "flexible" - are you saying that the position of the PDZ bm in the Nr-x1 Ct is flexible? This should be clarified.

We simplified the statement as follows:

Of note, the presence of the PDZ binding motif both *in cis*, in Nr-x-1(AT), or *in trans*, in reconstituted Nr-x-1 (Nr-x^{ΔPDZ-AT}/Nb-PDZ), ensures that Nr-x-1 traffics normally and reaches the synaptic terminals.

19/562: "endogenous levels of" : no quantitation of endogenous WT Nr-x1 or edited Nr-x1 AT protein has been performed as requested. I suggest removing any reference to "levels."

As the reviewer requested, this revised manuscript includes rigorous quantification of the endogenous levels of wild-type Nr-x-1 and edited Nr-x-1-AT. These data have been included in the new Suppl Fig. 5.

20/608: Necessity of PDZ binding motif for Nr-x1 synaptic stabilization is not clear from the present data.

We replaced this sentence with "function at synaptic terminals".

20-21/614-622: As written this seems conceptually disconnected from the discussion of the PDZ binding motif.

This paragraph has been simplified as follows:

Mammalian neuexins contain a highly conserved basic motif implicated in ER retention (Gokce and Sudhof 2013), whereas *Drosophila* Nr_x-1 contains a typical di-leucine ER retention motif [reviewed in (Bonifacino 2014)]. In addition, other sequences may be involved in trafficking of Nr_x-1 since neuronal expression of a Nr_x-1 variant with no intracellular residues can rescue the NMJ growth of *Nrx-1^{null}*, presumably by reaching the synaptic terminals and establishing productive interactions with Nlg1 (Banovic et al. 2010; Banerjee et al. 2017). Likewise, most of the cytoplasmic tails of mammalian neuexins are dispensable for neuexin-neuroligin interactions and formation of heterologous synapses in cultured cells (Scheiffele et al. 2000; Fairless et al. 2008; Gokce and Sudhof 2013).

21/627: By “flexibility” do the authors mean absolute position of this motif in the Nr_x1 C-terminus?

This point of the discussion is not only about absolute position, but also about different PDZ binding motifs.

21/634: homogeneously

We thank the reviewer for catching this typo which was corrected.

Reviewer #4 (Remarks to the Author):

The authors have addressed all my prior concerns. I think this is a good study that warrants publication, however, I do have concerns remaining regarding the new colocalization experiments with rab2 in Figure 6 that should be addressed before acceptance.

We thank this reviewer and appreciate very much the constructive feedback from both previous and current comments.

1. The selected regions for performing the colocalization appear arbitrary and in Figure 6g you can see in the top left and bottom left of the highlighted region Nr_x-1 positive punctae with no rab2 signal. Can the authors explain why these regions were chosen? I think the % correlation between Nr_x-1 and Rab2 will be lower should these cells be included (unless Rab2 signal is being masked by the Nr_x-1 which leads to comment 2)

We chose the regions highlighted and analyzed in Figure 6g to include type Ib motor neurons but exclude type Is. This is because, even though *Nrx1-Gal4* drive expression of various reporters in both type Ib and Is motor neurons, for reasons that we do not understand, *Nrx1>Rab2^{CA}YFP* cannot be detected in neither the soma nor the synaptic terminals of type Is motor neurons.

We included the following explanatory sentence in the Results section:

We limited our analyses to type Ib motor neurons and excluded the type Is, which showed expected pattern of expression for Nb-mScarlet but had very limited Rab2^{CA}-YFP expression for reasons that we do not understand.

2. For Figure 6g, it is important to show the rab2 and Nr_x-1 panels individually to see if those

cells do indeed express Rab2, more so than what is in 6k-r which is not super informative as the scarlet signal is uniform in the absence of the ALFA tag.

As recommended, we reorganized this figure and its associated supplemental to retain the most informative data in the main figure, including the panels requested by the reviewer. The more technical aspects of the data quantification have been included in the related supplemental figure (currently the new Supplementary Fig. 17).

3. The authors should acknowledge that there appears to be a cell-type-specific expression of Rab2 and therefore transport of Nr1 cannot be solely dependent on Rab2. This is also true as 25% of Nr1 vesicles are not in the vicinity of Rab2.

Indeed, 25% of the Nr1-AT/Nb-mScarlet containing vesicles are Rab2 negative, suggestive of additional Nr1 trafficking mechanisms. As indicated above, we also noted the cell type-specific expression of Rab2.

As the reviewer recommended, we modified the main text to reflect these possibilities. The revised manuscript reads as follows:

The absence of Rab2 signals in the remaining 25% of the vesicles together with the type Ib-specific expression of Rab2 noted above suggest that additional mechanism(s) may control Nr1 trafficking at synaptic terminals.

4. The authors claim that Nr1 comigrates with Rab2 to synaptic terminals without showing any colocalization in synaptic terminals (Line 554). This should either be changed to colocalization in vesicles, or NMJ stainings showing colocalization should be shown.

The reviewer is correct. Consequently, we changed the text as follows:

Together these results indicate that a large fraction of Nr1 co-localizes with Rab2-positive vesicles and may migrate together to synaptic terminals.

References:

- Banerjee S, Venkatesan A, Bhat MA. 2017. Neurexin, Neuroligin and Wishful Thinking coordinate synaptic cytoarchitecture and growth at neuromuscular junctions. *Molecular and cellular neurosciences* **78**: 9-24.
- Banovic D, Khorramshahi O, Oswald D, Wichmann C, Riedt T, Fouquet W, Tian R, Sigrist SJ, Aberle H. 2010. Drosophila neuroligin 1 promotes growth and postsynaptic differentiation at glutamatergic neuromuscular junctions. *Neuron* **66**: 724-738.
- Bonifacino JS. 2014. Adaptor proteins involved in polarized sorting. *J Cell Biol* **204**: 7-17.
- Budnik V, Koh YH, Guan B, Hartmann B, Hough C, Woods D, Gorczyca M. 1996. Regulation of synapse structure and function by the Drosophila tumor suppressor gene dlg. *Neuron* **17**: 627-640.
- Fairless R, Masius H, Rohlmann A, Heupel K, Ahmad M, Reissner C, Dresbach T, Missler M. 2008. Polarized targeting of neurexins to synapses is regulated by their C-terminal sequences. *J Neurosci* **28**: 12969-12981.
- Gokce O, Sudhof TC. 2013. Membrane-tethered monomeric neurexin LNS-domain triggers synapse formation. *J Neurosci* **33**: 14617-14628.

- Li J, Ashley J, Budnik V, Bhat MA. 2007. Crucial role of *Drosophila* neurexin in proper active zone apposition to postsynaptic densities, synaptic growth, and synaptic transmission. *Neuron* **55**: 741-755.
- Scheiffele P, Fan J, Choih J, Fetter R, Serafini T. 2000. Neuroligin expressed in nonneuronal cells triggers presynaptic development in contacting axons. *Cell* **101**: 657-669.

Reviewers' Comments:

Reviewer #1:

Remarks to the Author:

In their revised manuscript, Vicidomini et al. have now further improved their manuscript by exploring the function of the C-terminal PDZ-domain binding motif in *Drosophila* Neurexin-1 (Nrx-1), finding that adding the C-terminal PDZ-binding motif does not interfere with trafficking of Nrx-1 to NMJs but with proper insertion at presynaptic sites and synaptic function. The paper is a detailed structure-function analysis that largely confirms previous observations but also provides new information, e.g. with respect to Nrx-1 trafficking or the function of the C-terminal domain. I have only a few comments.

1) Suppl Fig. 9: Most of the microscopic images in this figure contain suspicious annotations, probably leftovers of the quantifications. However, these line drawings are not indicated in the legend and could easily be interpreted as true tissue signals. It would be best to remove them. In addition, it is absolutely necessary to remove dozens of conspicuous Greek letters written on the neuropile in the first image of the third row from the top.

2) Line 364-365 and 1213: "This insertion disrupts Nrx-1 cDNA..." and "The T2A-Gal4 in-frame insertion disrupts the expression of Nrx-1...", respectively. The authors refer to molecular consequences for Nrx-1 due to the insertion of the T2A-Gal4 construct at the Mi10178 locus. However, the molecular characterization and experimental verification is not described in Materials and Methods. Relying on indirect read-outs like homozygotic lethality or database annotations seems to be inappropriate for the first description of the line.

3) Line 445: The sentence "The Nrx-1 expression in larval enteroendocrine cells..." should be corrected to "Expression of the Nrx-1-Gal4 driver in larval enteroendocrine cells...", as expression of endogenous Nrx-1 protein was not shown in these cells. Please check the entire manuscript for similar occurrences.

4) Fig. 6: The issue with transported Nrx-1 particles in axons has not been resolved. Most particles seem to wiggle around in the soma rather than being actively transported in a specific direction. Quite a few particles are even transported retrogradely (Suppl. Mov3). While it is obvious that no such aggregates form in untagged Nrx-1, what is the evidence that these particles are not just unspecific aggregates due to increased Nrx-1-AT expression?

5) Fig. 6: Why is the association with Rab2-positive transport vesicles shown in the soma and not in peripheral axons, where the spatial resolution should be greatly increased? In addition, co-localization in the soma is still not recognizable at the depicted resolution. While HRP- and mScarlet-staining form a recognizable and similar pattern (Fig. 6d-6e), this is not seen in the Rab2-CA-YFP images (Fig. 6f). Highlighting co-localizing vesicle with arrows or showing single confocal layers rather than projections (or in addition to projections) might help. The patterns of puncta in the microscopic images (Fig. 6d-6f) seem to be also vastly different from those derived by machine learning algorithms (Fig. 6d''-f''). Due to these contradictions, I recommend to show the association using a second independent method.

Reviewer #2:

Remarks to the Author:

The authors have addressed most of my prior concerns. This is a good study demonstrating the usefulness of the ALFA-tag + nbAT-ATTOxxx system in vivo from a method point of view, making it suitable for publication. I remain less excited about the novelty of Nrx-1 data as they mostly repeat earlier findings, also from the Oswald et al {Oswald et al. (2012) 22864612} & Ramesh et al {Ramesh et al. (2021) 33651992} studies using a well-characterized *Drosophila* Nrx-1::GFP. The

Nrx-1AT + nbAT-GFP system variation appears rather cumbersome (without or with split as in Nrx-1AT Δ PDZ + nbAT-PDZ).

Unfortunately, a mistake was introduced in Figure 6C during the last revision: As shown in the attachment below (red circles), two data points vanished from their prior location and two new ones appeared at odd places in this box-whisker plot. I suspect that the two data points simply got "displaced" during the reorganization of Figure 6 but the authors might want to check all their data distribution diagrams for consistency carefully.

Reviewer #3:

Remarks to the Author:

The authors have done a commendable job in addressing the issues noted in previous iterations of this manuscript. In particular, the new Nb-PDZ-GFP experiments are a valuable addition to the study. These data reinforce the critical role of Nrx-1's PDZ binding motif in organizing nascent synapses and supporting normal neurotransmission.

While greatly improved, the manuscript would still benefit from a small amount of editing to reduce redundancy. For instance, the discussion of the qualities and potential uses of the ALFA tag/Nb system could be condensed, as these points are reiterated multiple times throughout the manuscript.

In addition to this, these comments should be addressed:

Line 168/ Fig. 1: It might be helpful to specify in the figure or legend that the "GFP" channel is showing us the background signal in flies not expressing GFP. It took a while to deduce that this is what seems to be depicted.

Lines 233-235: Please comment on the event frequency.

Reviewer #4:

Remarks to the Author:

Dear Authors,

I appreciate the work done to address the comments of myself and the other reviewers. The work done in this manuscript highlights new structure/function information for a key neurodevelopment/neurotransmission protein in Nrx-1 that will have value beyond the Drosophila research community. The extensive work in the manuscript is sufficient for me to view their claims as correct and done with suitable controls. The methodology used in the manuscript nicely highlights the use of the ALFA tag system for structure/function analysis and is done elegantly. The new experiments with the nb-PDZ-GFP expression add to this usefulness and provide more insight into the role of the PDZ domain.

Reviewer #5:

None

Reviewer #1 (Remarks to the Author):

In their revised manuscript, Vicidomini et al. have now further improved their manuscript by exploring the function of the C-terminal PDZ-domain binding motif in *Drosophila* Neurexin-1 (Nrx-1), finding that adding the C-terminal PDZ-binding motif does not interfere with trafficking of Nrx-1 to NMJs but with proper insertion at presynaptic sites and synaptic function. The paper is a detailed structure-function analysis that largely confirms previous observations but also provides new information, e.g. with respect to Nrx-1 trafficking or the function of the C-terminal domain. I have only a few comments.

1) Suppl Fig. 9: Most of the microscopic images in this figure contain suspicious annotations, probably leftovers of the quantifications. However, these line drawings are not indicated in the legend and could easily be interpreted as true tissue signals. It would be best to remove them. In addition, it is absolutely necessary to remove dozens of conspicuous Greek letters written on the neuropile in the first image of the third row from the top.

We thank the reviewer for pointing out these errors. We have removed the strange Greek letters as well as the drawings.

2) Line 364-365 and 1213: "This insertion disrupts Nrx-1 cDNA..." and "The T2A-Gal4 in-frame insertion disrupts the expression of Nrx-1...", respectively. The authors refer to molecular consequences for Nrx-1 due to the insertion of the T2A-Gal4 construct at the Mi10178 locus. However, the molecular characterization and experimental verification is not described in Materials and Methods. Relying on indirect read-outs like homozygotic lethality or database annotations seems to be inappropriate for the first description of the line.

As reviewer #2 pointed out in the first round of reviews, we were not the first to use this T2A-Gal4 line. We have already included the appropriate reference (Sun et al 2015) in the revised manuscript.

3) Line 445: The sentence "The Nrx-1 expression in larval enteroendocrine cells..." should be corrected to "Expression of the Nrx-1-Gal4 driver in larval enteroendocrine cells...", as expression of endogenous Nrx-1 protein was not shown in these cells. Please check the entire manuscript for similar occurrences.

We modified this sentence as recommended.

4) Fig. 6: The issue with transported Nrx-1 particles in axons has not been resolved. Most particles seem to wiggle around in the soma rather than being actively transported in a specific direction. Quite a few particles are even transported retrogradely (Suppl. Mov3). While it is obvious that no such aggregates form in untagged Nrx-1, what is the evidence that these particles are not just unspecific aggregates due to increased Nrx-1-AT expression?

In fact, we observed similar aggregates in animals with increased Nrx-1-AT expression, as determined by immunohistochemistry of fixed specimen (Fig. 1g). Importantly, these aggregates are only visible at the periphery of the neuropil and not in the neuron soma. Therefore, the simplest explanation is that these aggregates correspond to particles that are transported outside the neuron soma.

As the reviewer pointed out, the aggregates move in both directions. This is because the driver used here (*BG480-Gal4*) labels both motor neurons as well as sensory neurons, which send their axons from the periphery to the ventral ganglion.

5) Fig. 6: Why is the association with Rab2-positive transport vesicles shown in the soma and not in peripheral axons, where the spatial resolution should be greatly increased? In addition, co-localization in the soma is still not recognizable at the depicted resolution. While HRP- and mScarlet-staining form a recognizable and similar pattern (Fig. 6d-6e), this is not seen in the Rab2-CA-YFP images (Fig. 6f). Highlighting co-localizing vesicle with arrows or showing single confocal layers rather than projections (or in addition to projections) might help. The patterns of puncta in the microscopic images (Fig. 6d-6f) seem to be also vastly different from those derived by machine learning algorithms (Fig. 6d''-f''). Due to these contradictions, I recommend to show the association using a second independent method.

As the reviewer indicated, the resolution of our images (compressed to generate a manageable PDF file) may not do justice to the super-resolution analyses performed in our study. The high-resolution images provided in this revised manuscript clearly highlight the co-localization.

Also, as the reviewer recommended, we have extended the associated supplemental figure (S17) and show four single confocal layers with (mScarlet and Rab2) co-localizing vesicles marked by arrows.

In addition, we would like to restate our previous response to this reviewer on this topic.

As the reviewer mentioned, we realize that crushing the nerve or using additional mutants with axonal transport defects would be great experimental ways to enrich the axonal puncta. However, such extensive trafficking studies are beyond the scope of this manuscript. Here we wished to showcase the feasibility of such *in vivo* studies using the ALFA system. In addition, we captured a significant association of the Nr_x-1 with Rab2-positive vesicles. Nonetheless, we are hopeful that our work will inspire scientists who study synaptic trafficking (and know how to crush nerves) to include the ALFA system in their toolbox.

Reviewer #2 (Remarks to the Author):

The authors have addressed most of my prior concerns. This is a good study demonstrating the usefulness of the ALFA-tag + nbAT-ATTOxxx system *in vivo* from a method point of view, making it suitable for publication. I remain less excited about the novelty of Nr_x-1 data as they mostly repeat earlier findings, also from the Oswald et al {Oswald et al. (2012) 22864612} & Ramesh et al {Ramesh et al. (2021) 33651992} studies using a well-characterized *Drosophila* Nr_x-1::GFP. The Nr_x-1AT + nbAT-GFP system variation appears rather cumbersome (without or with split as in Nr_x-1ATΔPDZ + nbAT-PDZ).

Unfortunately, a mistake was introduced in Figure 6C during the last revision: As shown in the attachment below (red circles), two data points vanished from their prior location and two new ones appeared at odd places in this box-whisker plot. I suspect that the two data points simply got "displaced" during the reorganization of Figure 6 but the authors might want to check all their data distribution diagrams for consistency carefully.

We thank the reviewer for praising our efforts to improve this manuscript. We also appreciate catching the mistake in Fig. 6. We corrected it and revisited all the files to ensure accuracy.

Reviewer #3 (Remarks to the Author):

The authors have done a commendable job in addressing the issues noted in previous iterations of this manuscript. In particular, the new Nb-PDZ-GFP experiments are a valuable addition to the study. These data reinforce the critical role of Nr_x-1's PDZ binding motif in organizing nascent synapses and supporting normal neurotransmission.

We thank the reviewer for the appreciative comments.

While greatly improved, the manuscript would still benefit from a small amount of editing to reduce redundancy. For instance, the discussion of the qualities and potential uses of the ALFA tag/Nb system could be condensed, as these points are reiterated multiple times throughout the manuscript.

In addition to this, these comments should be addressed:

Line 168/ Fig. 1: It might be helpful to specify in the figure or legend that the "GFP" channel is showing us the background signal in flies not expressing GFP. It took a while to deduce that this is what seems to be depicted.

As recommended, we have added an explanatory sentence to the Fig. 1 legend as follows:
The GFP and ALFA tag signals/channels are shown for all genotypes to illustrate the background signal.

Lines 233-235: Please comment on the event frequency.

We have included mEJP frequency results in Fig. 2 but not in the subsequent figures. Consequently, we refrained from discussing or commenting on data not shown.

Reviewer #4 (Remarks to the Author):

Dear Authors,

I appreciate the work done to address the comments of myself and the other reviewers. The work done in this manuscript highlights new structure/function information for a key neurodevelopment/neurotransmission protein in Nr_x-1 that will have value beyond the Drosophila research community. The extensive work in the manuscript is sufficient for me to view their claims as correct and done with suitable controls. The methodology used in the manuscript nicely highlights the use of the ALFA tag system for structure/function analysis and is done elegantly. The new experiments with the nb-PDZ-GFP expression add to this usefulness and provide more insight into the role of the PDZ domain.

We thank the reviewer for the appreciative comments. We too hope that our work will encourage the field to take full advantage of the versatility of the ALFA system.